# Heterogeneous Agent Q-weighted Policy Optimization

**Bor Jiun, Lin**
Department of Computer Science
and Information Engineering,
National Taiwan University
`crlc112358@gmail.com`

**Chun-Yi Lee**
Department of Computer Science
and Information Engineering &
Artificial Intelligence Center
of Research Excellence,
National Taiwan University
`cylee@csie.ntu.edu.tw`

## Abstract

Multi-agent reinforcement learning (MARL) confronts a fundamental tension between stability and expressiveness. Stability requires avoiding divergence under non-stationary updates, while expressiveness demands capturing multimodal strategies for heterogeneous coordination. Existing methods sacrifice one for the other: value-decomposition and trust-region approaches ensure stability but assume restrictive unimodal policies, while expressive generative models lack optimization guarantees. To address this challenge, we introduce **H**eterogeneous **A**gent **Q**-weighted Policy **O**ptimization (HAQO), a framework unifying sequential advantage-aware updates, Q-weighted variational surrogates, and entropy regularization. Our analysis establishes monotone improvement guarantees under bounded critic bias, extending trust-region theory to diffusion-based policies with intractable log-likelihoods. HAQO achieves superior returns and reduced variance compared to policy-gradient baselines across diverse benchmarks. The ablation studies confirm sequential updates ensure stability, expressive policies enable multimodality, and entropy regularization prevents collapse. HAQO reconciles stability and expressiveness in MARL with theoretical rigor and practical effectiveness.

## 1 Introduction

Multi-Agent Reinforcement Learning (MARL) has emerged as a central paradigm for enabling intelligent agents to cooperate effectively in complex, dynamic environments. Its importance spans across domains such as robotics (Chen et al., 2025; Qin et al., 2025; Cai et al., 2024; Feng et al., 2025), autonomous driving (Shai et al., 2016; Wu et al., 2025; Taghavifar et al., 2025), and smart factory applications (Waseem & Chang, 2025; Leet et al., 2025; Movahed et al., 2025). Despite advances in these realms, designing MARL algorithms that achieve both stability and expressiveness remains a fundamental challenge. The complexity of MARL arises from four main factors: (i) the joint action space grows exponentially with the number of agents, which renders exhaustive exploration and optimization computationally infeasible, (ii) the learning process is inherently non-stationary, as each agent's policy update alters the environment dynamics observed by other agents, (iii) agents are often heterogeneous, possessing distinct dynamics, observation modalities, or action repertoires, and (iv) decentralized coordination presents additional obstacles where communication constraints and partial observability prevent centralized control in practical scenarios. Heterogeneity particularly exacerbates all previous challenges, as it complicates learning when agents operate with different observation spaces, action spaces, or functional roles, which usually necessitate specialized approaches that frequently produce brittle policies (Zhong et al., 2024), unlike homogeneous settings where parameter sharing mechanisms

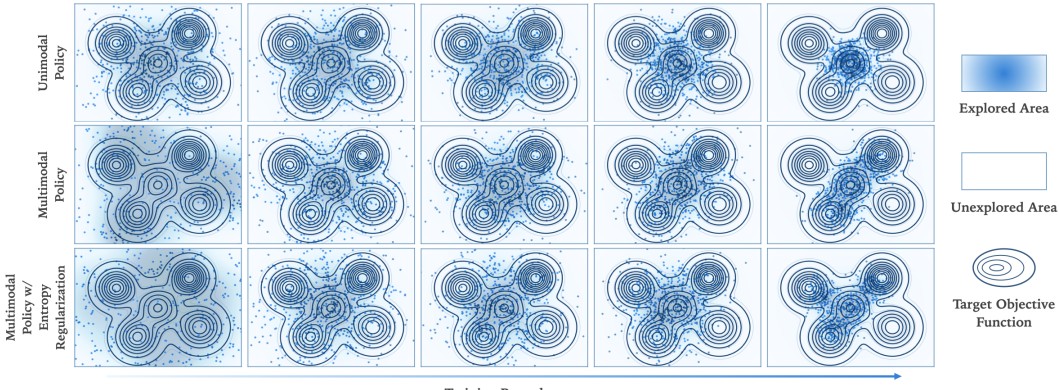

Figure 1: **Policy exploration comparison across different algorithmic components.** This multi-modal coordination task requires agents to discover and maintain coverage across four distinct target regions (circular contours) rather than collapsing to suboptimal single-mode strategies. We visualize agent policies through dimensionality reduction to demonstrate learned representations. **Top:** Gaussian policies exhibit rapid mode collapse, failing to capture optimal multimodal distributions. **Middle:** Multimodal policies improve coverage but suffer from exploration instability and mode imbalance. **Bottom:** HAQO combining multimodal policies with entropy regularization achieves stable, balanced exploration that effectively tracks all target modes throughout training.

function effectively. These challenges collectively render coordination and convergence substantially more challenging than in homogeneous MARL scenarios.

Among these challenges, heterogeneous multi-agent systems confront additional expressiveness limitations that compound the coordination difficulties. Expressiveness in MARL refers to the ability of policy representations to capture the full spectrum of behaviors and coordination patterns that agents may need to exhibit in complex multi-agent environments. When agents possess different capabilities, observation modalities, or roles, the coordination problem becomes even more complex as each agent must learn strategies that account for the diverse behavioral repertoires of its heterogeneous partners. Conventional unimodal policy classes, such as Gaussian policies (Tan, 1997), impose severe constraints on the types of strategies agents can learn and execute. These limitations become particularly problematic in heterogeneous scenarios where optimal coordination requires agents to switch between distinct behavioral modes or employ complex, multimodal action distributions that adapt to their partners' diverse capabilities. As a result, unimodal policy representations cannot adequately represent the rich strategic repertoire necessary for effective multi-agent coordination in heterogeneous systems, and often result in mode collapse where agents converge to suboptimal behaviors that represent only a limited subset of the available strategy space. Furthermore, simultaneous policy updates across all heterogeneous agents create destabilizing non-stationarity that impedes convergence to meaningful coordination solutions (Zhong et al., 2024). The limitation of unimodal policies is further illustrated in Fig. 1, where Gaussian policies collapse to a single strategy and fail to represent the multimodal behaviors needed for coordination. However, multimodal policies alone prove insufficient, as demonstrated by their unstable exploration patterns. The inclusion of entropy regularization in the proposed HAQO framework enhances the multimodal policy's exploration capabilities, effectively enabling more balanced coverage across all strategic modes.

Beyond these fundamental challenges, three critical issues emerge as particularly significant barriers to practical MARL deployment in heterogeneous environments. First, coordinating learning updates presents substantial difficulties, as simultaneous policy updates across multiple heterogeneous agents create conflicting gradients that cause training instability and hinder convergence to meaningful solutions. This challenge becomes exponentially

more complex when agents possess different learning capabilities, update frequencies, and policy representations, making traditional homogeneous coordination mechanisms inadequate. Second, policy expressiveness limitations severely constrain the strategic diversity that heterogeneous agents require for effective coordination. Conventional unimodal policy representations cannot capture the rich multimodal behaviors necessary for sophisticated coordination strategies, particularly when heterogeneous agents must adapt their decision-making processes to accommodate diverse partner capabilities, observation modalities, and action repertoires. Although multimodal RL objectives can be cast as variational lower bounds, extending these guarantees to heterogeneous MARL is non-trivial. The key obstacle is ensuring that expressive policies still yield improvement when agents differ in roles and update sequentially. Third, the integration of expressive policy representations with stable learning algorithms remains a fundamental challenge, as more sophisticated policy classes introduce additional variance and computational complexity that can destabilize the already fragile multi-agent learning dynamics in heterogeneous settings. These interconnected challenges highlight the critical need for algorithmic frameworks that can simultaneously address coordination stability, policy expressiveness, and heterogeneous agent capabilities while maintaining theoretical guarantees and practical applicability.

To overcome these challenges, we propose **Heterogeneous-Agent Q-weighted Optimization (HAQO)**, a framework that addresses the demanding requirements of stability and expressiveness in heterogeneous MARL. HAQO employs sequential advantage-aware updates to mitigate non-stationarity, a Q-weighted variational surrogate to align expressive policies with return maximization, and an entropy surrogate to sustain exploration. These components address the limitations of prior methods, and enable heterogeneous agents to maintain diverse behaviors while ensuring stability, providing theoretical guarantees and practical effectiveness. Our contributions are threefold:

1. Introduce a MARL framework that couples sequential advantage-based updates with expressive policy models, enabling coordination in heterogeneous agents with flexibility.

2. Generalize Q-weighted policy optimization to the multi-agent setting, extending advantage-weighted objectives with explicit importance-sampling corrections for sequential updates.

3. Derive an exact bound on return differences under sequential multi-agent updates, yielding tight constants that account for heterogeneous dynamics and non-stationary environments.

## 2 Related Works

Early MARL research adopted independent learning frameworks (Tan, 1997) where agents treated others as environmental components and applied single-agent algorithms. This approach demonstrated fundamental limitations due to non-stationarity from each agent's perspective, creating unstable learning dynamics and poor scalability. Centralized MARL methods (Wen et al., 2022; Pedro et al., 2024) addressed these limitations through information sharing and joint optimization, mitigating non-stationarity while improving coordination. However, these methods required full observability and centralized execution, proving impractical for real-world systems with quadratic computational scaling. The centralized training and decentralized execution (CTDE) paradigm (Sunehag et al., 2018) emerged as a breakthrough, exploiting global state information during training while relying on decentralized local policies for execution. CTDE enabled two dominant algorithmic families: value-based and policy-based approaches. Value decomposition methods (Rashid et al., 2018; Yang et al., 2020a; Rashid et al., 2020; Yang et al., 2020b; Iqbal et al., 2020; Zhou et al., 2020; Li et al., 2022; Shen et al., 2022; Sun et al., 2021; Liu et al., 2019; Hu et al., 2021; Wang et al., 2020; Hao et al., 2023; Lin & Lee, 2024) factorize joint value functions into per-agent components to address credit assignment. Policy-based approaches (Chao

et al., 2022; Papoudakis et al., 2021; Ackermann et al., 2019; Son et al., 2019; Wang et al., 2021) extend single-agent policy gradient theorems to multi-agent domains, often stabilized through trust-region or proximal objectives.

Recent research has addressed coordination challenges through sequential update schemes that update agents individually while conditioning on others' recent policies, providing formal guarantees for monotone joint return improvement and Nash equilibrium convergence (Zhong et al., 2024). Stabilization strategies include trust-region methods (Kuba et al., 2022c) that bound policy changes and coordination graphs (Kok & Vlassis, 2006) that approximate inter-agent dependencies. For policy expressiveness, normalizing flows (Ma et al., 2024) enable flexible distributions through invertible transformations, while diffusion models (Janner et al., 2022) demonstrate exceptional capability for generating diverse behaviors in sequential decision-making. Advantage-weighted objectives have emerged as promising approaches for biasing updates toward high-value actions while reducing variance, with methods like RWR (Štrupl et al., 2022) and AWR (Peng et al., 2021) casting optimization as weighted supervised learning. Recent extensions (Ding et al., 2024) connect these concepts with diffusion models through Q-weighted variational bounds approximating policy gradients. However, these approaches remain limited to homogeneous settings and perform poorly with differing observation or action spaces (Liu et al., 2024). Coordination mechanisms either rely on homogeneity assumptions or impose structural constraints limiting applicability in heterogeneous environments, while expressive policy representations introduce computational costs and variance, with stable integration into cooperative online multi-agent settings remaining largely unexplored.

## 3 PRELIMINARIES

### 3.1 HETEROGENEOUS-AGENT MIRROR LEARNING (HAML)

Zhong et al. (2024) generalizes the mirror-learning methodology (Kuba et al., 2022b) to multi-agent settings with heterogeneous agent configurations. In contrast to homogeneous MARL algorithms that rely on parameter sharing, HAML explicitly allows each agent to maintain a distinct policy tailored to its specific capabilities and constraints. HAML operates within the CTDE paradigm (Sunehag et al., 2018), and its primary innovation lies in extending trust-region style convergence guarantees to heterogeneous multi-agent settings through the integration of mirror-descent update rules for each individual agent. Specifically, for each agent $i$, the policy update takes the following form:

$$\pi^i_{\text{new}} = \arg \min_{\pi^i \in \Pi^i} \left[ D_{KL}(\pi^i \| \pi^i_{\text{old}}) - \eta \mathbb{E}_{s,a \sim \pi}[A^i(s,a)] \right] \tag{1}$$

where $\pi^i \in \Pi^i$ represents the policy for agent $i$ within its designated policy class $\Pi^i$, $D_{KL}(\pi^i | \pi^i_{\text{old}})$ denotes the Kullback-Leibler divergence between the new and old policies serving as a regularization term, $\eta$ is a scaling parameter, and $A^i(s,a)$ is the multi-agent advantage function that evaluates the relative value of actions taken by agent $i$ in the context of the joint state-action space. This formulation ensures that each agent's policy update balances exploitation of advantageous actions with regularization to prevent destabilizing changes that could disrupt coordination with others.

### 3.2 SEQUENTIAL UPDATE DECOMPOSITION

A central challenge in MARL is that simultaneous updates can create instability due to conflicting gradient directions. HAML and related methods (Zhong et al. (2024)) address this by updating agents sequentially in a randomized order $i_{1:n}$. Let $\pi_{old}$ and $\pi_{new}$ denote the joint policies before and after one round of updates. The return difference can be decomposed into per-agent contributions:

$$J(\pi_{\text{new}}) - J(\pi_{\text{old}}) = \sum_{m=1}^{n} \left[ J(\pi^{i_{1:m}}_{\text{new}}, \pi^{i_{m+1:n}}_{\text{old}}) - J(\pi^{i_{1:m-1}}_{\text{new}}, \pi^{i_{m:n}}_{\text{old}}) \right]. \tag{2}$$

This decomposition expresses the global return improvement as a summation of marginal improvements contributed by each agent, conditioned on the latest updates of its predecessors. As shown in (Zhong et al., 2024), this decomposition principle ensures monotonic improvement guarantees for the overall joint policy, provided that each individual agent's update yields local improvement.

### 3.3 Diffusion Policy

Diffusion policies represent a paradigm shift in reinforcement learning. They move beyond restrictive unimodal parametrizations to capture complex, multimodal action distributions essential for high-dimensional control. While foundational works by Zhong et al. (2024) and subsequent applications in single-agent domains have demonstrated their efficacy in robotic manipulation, their application to heterogeneous multi-agent systems remains underexplored. Our work builds upon the theoretical formalisms established by Chi et al. (2023) and Kuba et al. (2022a), and extends them by utilizing CTDE paradigm to address the specific challenges of heterogeneous coordination. Unlike prior works that rely on parameter sharing, our approach operates in a standard, purely cooperative MARL setting where agents optimize a shared objective $J(\theta) = \mathbb{E}\tau \sim p_\theta[\sum_t \gamma^t r_t]$. The integration of diffusion models serves not merely to enhance expressiveness, but to resolve the stability-expressiveness tension by enabling agents to model complex joint distributions without sacrificing the monotonic improvement guarantees provided by our sequential update framework.

### 3.4 Q-weighted Variational Policy Optimization (QVPO)

While sequential updates provide stability, policy expressiveness is equally critical for capturing multimodal strategies. Diffusion models Song et al. (2021) offer high-capacity generative policies, however, their variational lower bound (VLB) training objective lacks direct alignment with reinforcement learning. To address this, Ding et al. (2024) introduces a value-weighted training objective connecting VLB to policy gradient theory through the Q-weighted variational loss, formulated as:

$$\mathcal{L}_{\text{QVPO}}(\theta) = \mathbb{E}_{s,a\sim\pi}\mathbb{E}_{t\sim\text{Uniform}(1,T)}\mathbb{E}_{\epsilon\sim\mathcal{N}(0,I)}\left[Q(s,a)\cdot\left\|\epsilon - \epsilon_\theta\left(\sqrt{\bar{\alpha}_t}a + \sqrt{1-\bar{\alpha}_t}\epsilon, s, t\right)\right\|^2\right]. \tag{3}$$

where $\epsilon_\theta$ is the denoising network parameterized by $\theta$, $\bar{\alpha}_t$ is the schedule at $t$, and $T$ is the total timesteps. This objective forms a tight lower bound of the policy gradient objective under appropriate weighting transformations. This result bridges generative modeling and reinforcement learning, allowing diffusion policies to be trained online while prioritizing actions with higher Q-values.

### 4 Motivational Experiment

To motivate our approach, we design a occupation game that agents must cover all targets to maximize the visitation density across multiple targets, which constitutes a multimodal objective that requires coordinated coverage. Fig. 2 compares Gaussian policies with HAQO. Baseline methods exhibit mode collapse and concentrate on target subsets while neglecting others. In contrast, HAQO achieves diverse coverage across most targets with consistent performance.

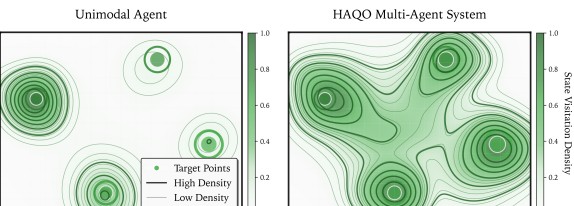

Figure 2: Comparison between a conventional MARL baseline employing unimodal Gaussian policies (left) and the proposed HAQO framework (right). The contours represent the state visitation density distribution.

This improvement stems from two key designs: expressive multimodal policies that represent diverse strategic behaviors, and sequential advantage-aware updates that mitigate the instability of simultaneous updates, confirming sequential coordination's importance.

## 5 METHODOLOGY

In this section, we present the **Heterogeneous-Agent Q-weighted Optimization (HAQO)** framework, which addresses the three critical challenges identified in heterogeneous MARL. It integrates: (i) sequential advantage-aware updates that mitigate non-stationarity and provide monotone improvement guarantees in Section 5.1; (ii) a Q-weighted lower bound for multi-agent setting in Section 5.2; and (iii) an entropy regularization term which bridges MARL with generative modeling and ensure robust exploration in Section 5.3. We synthesize these theoretical foundations and derive the monotone improvement bound for HAQO, together with the practical algorithm in Section 5.4.

### 5.1 TRUST-REGION MULTI-AGENT POLICY IMPROVEMENT VIA SEQUENTIAL UPDATES

In MARL, simultaneous policy updates can precipitate performance collapse since advantage functions are defined relative to joint policies, which causes concurrent policy changes to invalidate advantage estimates and eliminate improvement guarantees. To address this challenge, we adopt sequential update mechanisms to mitigate this non-stationarity by permuting the update order $i_{1:n}$ and define *per-agent surrogate objectives* $\mathcal{L}^i(\pi^i) = \mathbb{E}_{s \sim \rho^{\pi_{old}}, a^{-i} \sim \pi_{new}^{-i}} \mathbb{E}_{a^i \sim \pi^i}[A^i(s, a^i)]$ that condition on current policies of other agents $a^{-i} \sim \pi_{new}^{-i}$ while retaining the baseline state visitation distribution $\rho^{\pi_{old}}$ for analytical tractability. $\mathcal{L}^i(\pi^i)$ measures agent $i$'s advantage under updated peer policies while averaging over the baseline state distribution, enabling clean telescoping arguments in performance difference analysis as in Lemma A.8. Since $\mathcal{L}^i(\pi^i)$ relies on $s \sim \rho^{\pi_{old}}$, we must constrain policy update magnitudes to control the distributional mismatch between $(\rho^{\pi_{old}}, \rho^{\pi_{new}})$.

We impose per-agent trust-region constraints $\mathbb{E}_{s \sim \rho^{\pi_{old}}} \mathrm{KL}(\pi_{old}^i(\cdot|s) \| \pi^i(\cdot|s)) \leq \delta_i$ where $\delta_i$ is a small update radius. This heterogeneous-agent analogue of the TRPO KL-divergence constraint ensures that distributional drift across sequential updates remains quadratically small, which leads to the tighter bound in Proposition 5.1 following Zhong et al. (2024). This proposition establishes that sequential updates guarantee monotone joint return improvement up to quadratic KL drift penalties, with bounds scaling linearly rather than exponentially with agent count because since sequential execution avoids simultaneous update instability. We provide a structural explanation for why sequential updates deliver stable practical improvement, with proof derivation in Appendix A.6.

**Proposition 5.1** (Heterogeneous Sequential Improvement Bound). *Let agents $i \in \{1, 2, \ldots, n\}$ update in a random permutation order $i_{1:n}$. Suppose rewards are bounded (Assumption A.1), policies are regular (Assumption A.3), and trust-region constraints hold with radius $\delta_i$ for each agent $i$ (Assumption A.5). Then there exist constants $C_i = O\left(\frac{\gamma R_{\max}}{(1-\gamma)^2}\right)$ for each agent $i$ such that:*

$$J(\pi_{\text{new}}) - J(\pi_{\text{old}}) \geq \sum_{i=1}^{n} \left( \mathcal{L}^i(\pi_{\text{new}}^i) - \mathcal{L}^i(\pi_{\text{old}}^i) \right) - \sum_{i=1}^{n} C_i \, \delta_i^2, \quad \textit{(proof at Appendix A.6)} \quad (4)$$

### 5.2 HETEROGENEOUS AGENT Q-WEIGHTED VARIATIONAL (QV) DIFFUSION OBJECTIVE

Standard policy-gradient methods rely on gradients of $\log \pi^i(a^i|s)$. However, in diffusion-based actors, the likelihood of a raw action is intractable: the policy is defined implicitly through a reverse SDE, and $\log \pi^i$ cannot be evaluated in closed form. This limitation precludes the use of classical likelihood-ratio surrogates. To resolve this, following Ding et al. (2024), we leverage a Q-weighted denoising objective, which is both (i) implementable for diffusion models via score matching, and (ii) theoretically aligned with return maximization through a VLB. For each agent $i$, conditioning on the already-updated policies of other agents, we define the multi-agent QV surrogate as follows:

$$\mathcal{J}_i^{\text{QV}}(\theta^i) = \mathbb{E}_{s \sim \rho^{\pi_{old}}} \mathbb{E}_{a^{-i} \sim \pi_{new}^{-i}} \mathbb{E}_{a^i \sim \pi_{old}^i, \epsilon, t}\left[ \omega^i(s, a^i, a^{-i}) \left\| \epsilon - \epsilon_{\theta^i}(\sqrt{\bar{\alpha}_t} \, a^i + \sqrt{1 - \bar{\alpha}_t} \, \epsilon, s, t) \right\|^2 \right], \quad (5)$$

where $\epsilon_{\theta^i}$ is the denoiser, and $\omega^i(s, a^i, a^{-i})$ is a non-negative weight function derived from the conditional advantage. Intuitively, $\mathcal{J}^{\mathrm{QV}}$ biases denoising updates toward actions that exhibit high advantage under the current critic. In principle, one could set $\omega^i(s, a^i, a^{-i}) = A^i(s, a^i)$ directly. However, since $A^i$ can be negative, this would yield a sign-indefinite objective that no longer lower-bounds the improvement surrogate. To preserve positivity and hence variational tightness, we adopt transformations that retain advantage alignment while discarding negative contributions as follows:ransformations that retain advantage alignment while discarding negative contributions:

$$\mathsf{qadv}: \quad \omega^i(s, a^i, a^{-i}) = \max\Big\{0, \ A^i(s, a^i) - \mathbb{E}_{\tilde{a}^i \sim \pi^i_{old}} A^i(s, \tilde{a}^i)\Big\}, \tag{6}$$

$$\mathsf{qcut}: \quad \omega^i(s, a^i, a^{-i}) = \max\{0, \ A^i(s, a^i)\} \cdot \mathbf{1}\{A^i(s, a^i) \geq \varepsilon\}. \tag{7}$$

The $\mathsf{qadv}$ transformation centers advantages before rectification to maintain weight calibration around the baseline policy, while $\mathsf{qcut}$ imposes thresholds to focus on high-advantage samples and filter out noise. These transformations create a practical objective that remains non-negative while capturing relative advantage scaling and ensuring the denoising loss serves as a valid policy-gradient surrogate. Proposition 5.2 demonstrates that $\mathcal{J}^{\mathrm{QV}}_i(\theta^i)$ serves as a robust policy improvement surrogate in heterogeneous settings, maintaining tight bounds with exact critics and degrading gracefully under approximation. The positivity constraint preserves monotonicity by preventing negative-advantage interference, while $\mathsf{qadv}/\mathsf{qcut}$ balances smooth calibration with selective precision, enabling expressive diffusion policies to achieve the same theoretical guarantees as classical policy gradients without restrictive unimodal assumptions. The detailed proof is offered in Appendix A.7.

**Proposition 5.2** (QV Lower Bound)**.** *Let $\omega^i$ be constructed via $\mathsf{qadv}$ or $\mathsf{qcut}$ such that $\omega^i \geq 0$ and is monotone in the conditional advantage. For any feasible diffusion policy $\pi^i$ with denoiser $\epsilon_{\theta^i}$,*

$$\mathcal{L}^i(\pi^i) \ \geq \ \mathcal{J}^{\mathrm{QV}}_i(\theta^i) \ - \ \xi_i, \qquad \xi_i = O(\epsilon_Q), \quad \text{(proof at Appendix A.7)} \tag{8}$$

*where $\epsilon_Q$ is the critic bias from Assumption A.4. The implicit constants in $O(\epsilon_Q)$ scale with the diffusion forward-noise schedule. Equality holds in the zero-bias, well-specified reverse-process limit.*

## 5.3 Entropy Regularization for Diffusion-Based Policies

Entropy regularization is essential in modern RL (Liu et al., 2024) by preventing premature convergence and promoting exploration, typically computed via $\log \pi^i(a^i|s)$ for tractable densities. However, diffusion-based actors define policies implicitly, rendering $\log \pi^i$ computationally intractable. To resolve this, we introduce an entropy surrogate that injects uniformly sampled actions into the denoising objective, broadening policy support and preventing collapse. Proposition 5.3 shows this surrogate maintains positive spectral floors in action covariance, providing entropy lower bounds that enable diffusion policies to retain diverse strategies without explicit likelihood computation.

Proposition 5.3 shows that diffusion policies can achieve entropy regularization without explicit log-likelihood computation. The surrogate employs uniform reconstruction as a state-agnostic baseline, which guarantees positive variance floors that prevent mode collapse while remaining compatible with denoising formulations. This surrogate balances with the Q-weighted objective, where $\mathcal{J}^{\mathrm{QV}}$ drives probability mass toward high-advantage actions, while $\mathcal{J}^{\mathrm{ent}}$ prevents pathological concentration around a narrow set of actions. This effectively extends maximum-entropy RL principles to intractable settings. Although one-sided in preventing collapse rather than upper-bounding entropy, this surrogate enables expressive diffusion policies to maintain exploration guarantees equivalent to those of classical maximum-entropy methods, while preserving HAQO's sequential update stability.

**Proposition 5.3** (Entropy Surrogate for Diffusion Actors)**.** *Let $\pi^i$ be a diffusion-based policy whose density $\pi^i(a^i \mid s)$ is implicit. Define the surrogate objective:*

**Table 1. Performance Comparison on Multi-Agent MuJoCo Environments**

| Environment | HAA2C | MAPPO | HATRPO | HAPPO | HAQO |
|---|---|---|---|---|---|
| Ant-v2 4x2 | 5637 (86) | 5874 (32) | 5013 (432) | 5793 (59) | **6014 (201)** |
| HalfCheetah-v2 2x3 | 4231 (1069) | 6984 (132) | 5369 (247) | **7024 (103)** | 6873 (137) |
| Hopper-v2 3x1 | 1832 (923) | 3612 (57) | 3733 (102) | 3481 (173) | **3884 (81)** |
| Walker2d-v2 2x3 | 1124 (94) | 5013 (483) | 3744 (373) | 5523 (214) | **5681 (301)** |
| Walker2d-v2 6x1 | 1923 (234) | **4693 (247)** | 2109 (223) | 4317 (401) | 4789 (293) |
| Humanoid-v2 17x1 | − | 732 (13) | − | 6739 (201) | **7013 (311)** |

Values represent mean returns with standard deviations. **Blue** indicates other algorithms' best performance, **Red** for HAQO's.

$$\mathcal{J}_i^{\text{ent}}(\theta^i) \;=\; \mathbb{E}_s \, \mathbb{E}_{\tilde{a}^i \sim U(\mathcal{A}^i),\, \epsilon,\, t} \left[ \alpha_i \left\| \epsilon - \epsilon_{\theta^i}(\sqrt{\bar{\alpha}_t}\, \tilde{a}^i + \sqrt{1 - \bar{\alpha}_t}\, \epsilon,\, s,\, t) \right\|^2 \right], \textit{(proof at Appendix A.8)} \tag{9}$$

*where $\alpha_i > 0$ is a scaling coefficient and $\epsilon_{\theta^i}$ is the diffusion denoiser. Then:*

1. *$\mathcal{J}_i^{\text{ent}}(\theta^i) \geq 0$ for all $\theta^i$, ensuring that it provides only non-negative exploration pressure.*

2. *Minimizing $\mathcal{J}_i^{\text{ent}}$ enforces a spectral floor $\lambda_{\min}(\Sigma_s^i) \geq \sigma_{\min}^2 > 0$, guaranteeing the bound*

$$H(A^i \mid s) \;\geq\; \frac{d_i}{2} \log\!\big(2\pi e\, \sigma_{\min}^2\big), \quad \textit{where } \Sigma_s^i = \text{Var}(A^i \mid s) \textit{ and } d_i = \dim \mathcal{A}^i. \tag{10}$$

### 5.4 Monotone Improvement Guarantees for HAQO

HAQO integrates three essential components: **(i)** a drift penalty via KL divergence or PPO-style clipping that constrains updates locally around the baseline policy, **(ii)** the Q-weighted variational surrogate, and **(iii)** the entropy surrogate. These define a tractable optimization problem whose solution guarantees theoretical improvement and practical stability under sequential heterogeneous updates. For agent $i$ in the random permutation order, the HAQO objective is defined as follows:

$$J_i^{\text{HAQO}}(\theta^i) = \max_{\pi^i \in T^i(\pi_{\text{old}}^i)} \underbrace{J_i^{\text{QV}}(\theta^i)}_{\text{QV lower bound surrogate}} + \underbrace{J_i^{\text{ent}}(\theta^i)}_{\text{entropy surrogate}} - \underbrace{D^i(\pi^i \| \pi_{\text{old}}^i)}_{\text{drift penalty (KL or PPO-clip)}} \tag{11}$$

where $T^i(\pi_{\text{old}}^i)$ is the feasible trust region around the baseline policy, and $D^i$ denotes either an expected KL divergence or a PPO clipping that enforces bounded deviation from the current policy.

This per-agent objective synthesizes three interlocking components: $J_i^{\text{QV}}(\theta^i)$ drives return-improving updates, $J_i^{\text{ent}}(\theta^i)$ injects controlled exploration without tractable log-densities, and the drift penalty constrains updates within safe baseline policy neighborhoods. Theorem 5.4 establishes monotone return improvement under quadratic drift penalties and bounded critic error. Algorithm 1 integrates these theoretical components with $K$-candidate variance reduction into a single iterative procedure with centralized critic updates followed by randomized per-agent policy optimization, translating theoretical guarantees into a scalable, stable algorithm for practical implementation.

**Theorem 5.4** (Monotone Improvement of HAQO)**.** *Assume bounded rewards (Assumption A.1), regular policies (Assumption A.3), critic bias $\epsilon_Q$ (Assumption A.4), and sequential updates with feasible neighborhoods constrained by $D^i$. Then the HAQO update satisfies (proof at Appendix A.9.):*

$$J(\pi_{\text{new}}) - J(\pi_{\text{old}}) \;\geq\; \sum_{i=1}^{n} \left( \mathcal{J}_i^{\text{QV}}(\theta_{\text{new}}^i) + \mathcal{J}_i^{\text{ent}}(\theta_{\text{new}}^i) \right) - \sum_{i=1}^{n} C_i\, \delta_i^2 - O(\epsilon_Q), \quad C_i = O\!\Big(\tfrac{\gamma R_{\max}}{(1-\gamma)^2}\Big). \tag{12}$$

## 6 Experimental Results

### 6.1 Experimental Setups

**Environments.** We evaluate HAQO on established benchmarks encompassing key coordination challenges: Multi-Agent Particle Environment (MPE) (Mordatch & Abbeel, 2017) for discrete and continuous control, StarCraft Multi-Agent Challenge (SMAC/SMACv2) (Vinyals et al., 2017; Ellis et al., 2022) for discrete coordination under partial observability, Google Research Football (GRF) (Kurach et al., 2020) for high-dimensional strategic coordination requiring multimodal policies, Multi-MuJoCo (Kuba et al., 2022a) for continuous control, and Bi-DexterousHands (Bi-D) (Zhong et al., 2024) for bimanual manipulation coordination. This suite systematically varies action dimensionality, observability constraints, and inter-agent coupling to provide rigorous evaluation across diverse coordination paradigms. The complete specifications are in Appendix A.11.1.

**Baselines.** We compare against state-of-the-art (SoTA) value-based and policy-based cooperative MARL methods. From value decomposition, we include QMIX (Rashid et al., 2018), which demonstrates strong cooperative performance but relies on monotonic factorization that limits heterogeneity handling. Among policy-gradient approaches, we evaluate MAPPO (Chao et al., 2022) with centralized training and parameter sharing, MADDPG (Lowe et al., 2017) with individual actor-critic pairs that faces scalability challenges, and recent heterogeneous methods HAPPO and HATRPO (Zhong et al., 2024) with sequential updates under trust-region constraints. While these approaches improve stability over simultaneous updates, they remain limited by unimodal Gaussian policies. All results use three random seeds with consistent hyperparameters for fair comparison.

### 6.2 Comparison to Policy-based Approaches

In this subsection, we compare HAQO against MAPPO, MADDPG, HAPPO, and HATRPO (i.e., the SoTA policy-based MARL method that combines trust-region updates with advantage-weighted objectives). In Multi-MuJoCo, where agents jointly control continuous body parts, HAQO consistently outperforms policy-gradient baselines. Table 1 shows HAQO achieves higher asymptotic returns and more stable convergence in continuous control tasks, demonstrating effective handling of heterogeneous action spaces. These results confirm that HAQO inherits trust-region stability while leveraging expressive diffusion policies to capture strategies that unimodal baselines miss.

Specifically, the ***Humanoid*** task illustrates this advantage: successful locomotion requires multimodal behaviors including alternating leg phases, torso stabilization, and arm coordination. Gaussian policies in MAPPO and HAPPO collapse to homogeneous behaviors causing unstable locomotion, while HAQO's diffusion policies capture multimodal gait cycles for robust walking. Our additional evaluation results for MPE, SMAC/SMACv2, GRF, and Bi-D in Appendix A.11.2 further confirm HAQO's superior stability, diverse strategies, and robust coordination across all domains.

### 6.3 Ablation Studies

#### 6.3.1 Sequential Update

Ablation studies in Bi-D reveal the importance of sequential updates. Without sequential updates, conflicting actions cause dropped objects and unstable grasps, with Fig. 3 showing concurrent updates yield poor performance and frequent failures. Sequential updates allow agents to adapt to partners' recent policies, stabilizing coordination and establishing HAQO's sequential update foundation.

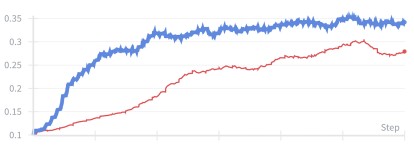

Figure 3: Average step rewards with sequential updates versus simultaneous updates. Sequential updates yield faster learning and higher performance.

### 6.3.2 EXPRESSIVE POLICY

We compare diffusion-based policies with unimodal Gaussian policies using average returns in Table 4. Diffusion policies consistently achieve higher returns with lower deviations, exemplified in 3 vs 1 with keeper where Gaussian agents obtain $66.82 \pm 9.12$ while diffusion agents achieve $97.27 \pm 1.07$.

Figure 4: Gaussian vs Diffusion Policy on GRF.

| Environment | Agent w/ Gaussian Policy | Agent w/ Diffusion Policy |
|---|---|---|
| PS | 76.24 (8.23) | **92.14 (2.13)** |
| RPS | 43.27 (10.31) | **80.39 (3.87)** |
| 3 vs 1 with keeper | 66.82 (9.12) | **97.27 (1.07)** |

In complex tasks like Pass and shoot with keeper (PS) and Run pass and shoot with keeper (RPS), Gaussian agents converge to brittle single-tactic behaviors while diffusion policies sustain diverse tactical modes. The reduced deviation confirms that diffusion policies provide both higher and more stable performance, demonstrating that expressive policies are essential in GRF's multimodal coordination scenarios.

### 6.3.3 ENTROPY REGULARIZATION

We evaluate entropy regularization in a customized MPE environment with two agents covering three landmarks requiring distributed coverage maximization. Without entropy regularization (Fig. 5 left), agents collapse to covering only two landmarks, reflecting value-driven objectives' over-exploitation. In contrast, with entropy regularization (Fig. 5 right), agents maintain balanced coverage

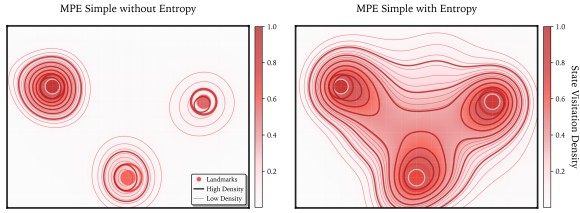

Figure 5: The effect of entropy regularization for MPE. State visitation density of two agents across three landmarks without entropy (left) and with entropy (right).

across all landmarks. These results confirm Proposition 5.3: the entropy surrogate prevents collapse even without explicit log-likelihoods, and can result in robust and generalizable coordination policies.

## 7 CONCLUSIONS

We proposed HAQO, a unified framework for heterogeneous-agent RL integrating sequential updates, Q-weighted surrogates, and entropy regularization. Our theoretical analysis established monotone improvement guarantees for expressive policies, and empirical results showed that HAQO consistently outperforms policy-based baselines. Ablation studies confirmed each component's necessity: sequential updates mitigate non-stationarity, diffusion policies capture multimodality, and entropy regularization prevents collapse. These contributions show that stability and expressiveness can be reconciled within a framework, enabling scalable coordination in complex MARL systems.

## 8 ACKNOWLEDGE

The authors gratefully acknowledge the support from the National Science and Technology Council (NSTC) in Taiwan under grant numbers NSTC 114-2221-E-002-069-MY3, NSTC 113-2221-E-002-212-MY3, and NSTC 114-2218-E-A49-026, as well as the support from the Academia Sinica Scholar Award (ASSA) under grant number AS-ASSA-115-02. The authors would also like to express their appreciation for the donation of the GPUs from NVIDIA Corporation and NVIDIA AI Technology Center (NVAITC) used in this work. Furthermore, the authors extend their gratitude to the National Center for High-Performance Computing (NCHC) for providing computational and storage resources. The authors also thank the NVIDIA Taipei-1 supercomputer for providing essential computing resources.

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

# A APPENDIX

## A.1 NOTATIONS

We summarize the notations used throughout this paper. These notations span policy representations, value functions, diffusion-based objectives, and auxiliary constants introduced in our theoretical analysis. Table 2 provides a comprehensive overview.

**Table 2. Summary of notations used in the HAQO framework.**

| Notation | Description |
|---|---|
| $\pi_i, \pi_i^{\text{old}}, \pi_i^{\text{new}}$ | Policy of agent $i$; old and updated versions in sequential optimization. |
| $\Pi_i$ | Policy class for agent $i$. |
| $J(\pi)$ | Expected return under joint policy $\pi$. |
| $Q^i(s, a)$ | Action-value function (critic) for agent $i$. |
| $V^i(s)$ | State-value function for agent $i$. |
| $A^i(s, a)$ | Advantage function for agent $i$: $A^i(s, a) = Q^i(s, a) - V^i(s)$. |
| $\rho_\pi$ | State visitation distribution under policy $\pi$. |
| $D_{\text{KL}}(\pi_i \| \pi_i')$ | Kullback-Leibler divergence between two policies of agent $i$. |
| $\delta_i$ | Trust-region radius (KL constraint) for agent $i$. |
| $\mathcal{N}$ | The number of agents in the environment. |
| $\epsilon$ | PPO clipping parameter for ratio constraints. |
| $R_{\max}$ | Upper bound on per-step reward. |
| $\gamma$ | Discount factor. |
| $C_i$ | Constant in sequential improvement bound, $C_i = \mathcal{O}\left(\frac{\gamma R_{\max}}{(1-\gamma)^2}\right)$. |
| $\theta_i$ | Parameters of agent $i$'s diffusion policy (denoising network). |
| $\epsilon, \epsilon_\theta$ | Gaussian noise and denoising network output in diffusion objectives. |
| $\alpha_t, \bar{\alpha}_t$ | Diffusion noise schedule parameters at timestep $t$. |
| $\omega_i(s, a_i, a_{-i})$ | Advantage-based weight function (e.g., $q_{\text{adv}}, q_{\text{cut}}$). |
| $q_{\text{adv}}$ | Transformation: $\max\{0, A^i(s, a_i) - \mathbb{E}_{\tilde{a}_i \sim \pi_i^{\text{old}}} A^i(s, \tilde{a}_i)\}$. |
| $q_{\text{cut}}$ | Transformation: $\max\{0, A^i(s, a_i)\} \cdot \mathbf{1}\{A^i(s, a_i) \geq \varepsilon\}$. |
| $\alpha_i$ | Scaling coefficient for entropy regularization. |
| $\Sigma_s^i$ | Covariance of action distribution for agent $i$ at state $s$. |
| $d_i$ | Action dimension of agent $i$. |
| $\xi_i$ | Slack term due to critic bias: $\xi_i = \mathcal{O}(\epsilon_Q)$. |
| $\epsilon_Q$ | Maximum bias in critic approximation. |
| $\Delta_m(s, a_{i_{1:m}})$ | Incremental contribution in sequential advantage decomposition. |
| $i_{1:n}$ | Random permutation order of agents for sequential updates. |
| $\mathcal{T}^i(\pi_i^{\text{old}})$ | Feasible trust-region neighborhood around baseline policy. |
| $D_i(\pi_i \| \pi_i^{\text{old}})$ | Drift penalty term (KL divergence or PPO-style clipping). |
| $K_b, K_t$ | Candidate counts: $K_b$ for behavior sampling, $K_t$ for target critic updates. |

A.2 HAQO ALGORITHM

This section provides the complete HAQO training procedure in Algorithm 1. While the manuscript highlights key theoretical components, i.e., sequential updates, Q-weighted surrogates, and entropy regularization, this algorithm consolidates them into a practical pipeline specifying critic updates, sequential agent permutation, and variance reduction through candidate sampling. This allows readers to directly connect theoretical analysis with implementation, and ensures reproducibility.

---

**Algorithm 1** HAQO (Sequential CTDE with Diffusion Actors)

---

1: **Inputs:** Replay buffer $\mathcal{D}$; Actor policys $\{\pi_{\theta^i}\}_{i=1}^n$; Critic policy $\{Q_\omega^i\}$; Trust radii $\{\delta_i\}$ / PPO clip $\epsilon$; Entropy weights $\{\alpha_i\}$; Candidate counts $K_b$ (behavior), $K_t$ (target).
2: **for** iteration $k = 0, 1, \ldots$ **do**
   ———————————— **Data Collection** ————————————
3:     **Sample candidates:** For each state $s$ and agent $i$, draw $K_b$ candidates $a_j^i \sim \pi_{\theta^i}(\cdot|s)$.
4:     **Score actions:** Evaluate each candidate using critic: $Q_\omega^i(s, a_j^i, a^{-i})$.
5:     **Execute best:** Select joint action $a = (a^{1,\star}, \ldots, a^{n,\star})$ with $a^{i,\star} \in \arg\max_j Q_\omega^i$.
6:     **Store transition:** Append $(s, a, r, s')$ to replay buffer $\mathcal{D}$.
   ———————————— **Critic Update** ————————————
7:     **Targets:** Train $Q_\omega^i$ by minimizing TD loss with $K_t < K_b$:

$$y_t^i = r_t + \gamma \, \mathbb{E}_{a' \sim \pi_{K_t}^i(\cdot|s_{t+1})}\big[\, Q_\omega^i(s_{t+1}, a', a_{t+1}^{-i}) \,\big].$$

   ———————————— **Actor Update** ————————————
8:     **Permutation:** Sample a random order $i_{1:n}$.
9:     **for** $m = 1$ to $n$ **do**
10:         **Sample batch:** Draw mini-batch $\mathcal{B} = (s, a, r, s')$ from $\mathcal{D}$.
11:         **Advantage est.:** Compute $A^{i_m}(s, a^{i_m})$ (e.g., GAE or TD($\lambda$)).
12:         **Weight computation:** Set weights $\omega^{i_m}(s, a^{i_m}, a^{-i_m})$ using Eq. (6):

$$\text{qadv:} \quad \omega^{i_m} \leftarrow \max\Big\{0, \, A^{i_m}(s, a^{i_m}) - \mathbb{E}_{\tilde{a}^{i_m} \sim \pi_{\text{old}}^{i_m}}\big[A^{i_m}(s, \tilde{a}^{i_m})\big]\Big\}$$

$$\text{qcut:} \quad \omega^{i_m} \leftarrow \max\{0, \, A^{i_m}(s, a^{i_m})\} \cdot \mathbf{1}\{A^{i_m}(s, a^{i_m}) \geq \varepsilon\}$$

13:         **Diffusion target:** Add noise $a_t^{i_m} = \sqrt{\bar{\alpha}_t}\, a^{i_m} + \sqrt{1 - \bar{\alpha}_t}\, \epsilon$ with $\epsilon \sim \mathcal{N}(0, I)$.
14:         **QV surrogate:** Construct the Q-weighted variational objective as in Eq. (5):

$$\mathcal{J}_{i_m}^{\text{QV}}(\theta^{i_m}) = \mathbb{E}_{\mathcal{B}, t, \epsilon}\Big[\omega^{i_m}(s, a^{i_m}, a^{-i_m}) \cdot \big\|\epsilon - \epsilon_{\theta^{i_m}}(a_t^{i_m}, t, s)\big\|^2\Big]$$

15:         **Entropy regularization:** Add entropy surrogate from Eq. (9):

$$\mathcal{J}_{i_m}^{\text{ent}}(\theta^{i_m}) = \alpha_{i_m} \, \mathbb{E}_{s, \, \tilde{a}^{i_m} \sim U(\mathcal{A}^{i_m}), \, t}\Big[-\big\|\tilde{a}^{i_m} - \mu_{\theta^{i_m}}(\tilde{a}_t^{i_m}, t, s)\big\|^2\Big].$$

16:         **Policy gradient:** $g^{i_m} \leftarrow \nabla_{\theta^{i_m}}\big(\mathcal{J}_{i_m}^{\text{QV}} + \mathcal{J}_{i_m}^{\text{ent}}\big)$ from Eq. (11).
17:         **PPO:** Define importance ratio $r^{i_m}(\theta) = \frac{\pi_{\theta^{i_m}}(a^{i_m}|s)}{\pi_{\theta_{\text{old}}^{i_m}}(a^{i_m}|s)}$ and the clipped surrogate:
$\mathcal{L}_{i_m}^{\text{clip}}(\theta) = \mathbb{E}_{(s,a) \sim \mathcal{B}}\Big[\min\big(r^{i_m}(\theta)\, A^{i_m}(s, a^{i_m}), \, \text{clip}(r^{i_m}(\theta), 1 - \epsilon, 1 + \epsilon)\, A^{i_m}(s, a^{i_m})\big)\Big]$

$- \beta_{\text{KL}} \, \mathbb{E}_s \, \text{KL}\big(\pi_{\theta_{\text{old}}^{i_m}}^{i_m}(\cdot \mid s) \,\|\, \pi_{\theta^{i_m}}^{i_m}(\cdot \mid s)\big).$

18:         **Drift Penalty:** Take a gradient step on $\mathcal{L}_{i_m}^{\text{clip}}(\theta)$.
19:         **Parameter update:** $\theta^{i_m} \leftarrow \theta^{i_m} + \eta \Delta\theta^{i_m}$.

---

A.3  STANDING ASSUMPTIONS

Our analysis builds upon a collection of assumptions that are fundamental in the reinforcement learning literature. We emphasize their particular significance in the heterogeneous-agent, diffusion-based setting of HAQO. Each assumption represents both a technical and modeling choice that identifies the precise theoretical regime in which our monotonicity and variance guarantees hold.

**Assumption A.1** (Bounded rewards)**.** *There exists a finite constant $R_{\max} > 0$ such that*

$$|r(s,a)| \leq R_{\max}, \quad \forall(s,a) \in \mathcal{S} \times \mathcal{A}. \tag{13}$$

*As a consequence:*

$$|Q^i(s,a)| \leq \frac{R_{\max}}{1-\gamma}, \qquad \forall i \in \mathcal{N}, \tag{14}$$

$$|V^i(s)| \leq \frac{R_{\max}}{1-\gamma}, \qquad |A^i(s,a)| \leq \frac{2R_{\max}}{1-\gamma}. \tag{15}$$

*Discussion.* This is the classical bounded-reward assumption used in performance difference bounds (e.g., Schulman et al. (2015)). It ensures that all value functions are uniformly bounded by a geometric series argument:

$$\left|\mathbb{E}\Big[\sum_{t=0}^{\infty} \gamma^t r_t\Big]\right| \leq \sum_{t=0}^{\infty} \gamma^t R_{\max} = \frac{R_{\max}}{1-\gamma}. \tag{16}$$

Several important consequences follow:

- **Finite surrogates.** Every surrogate objective we optimize, namely advantage-weighted, Q-weighted, or entropy-regularized objectives, remains finite. In particular, $\mathcal{L}_{QV}^i(\pi^i) = \mathbb{E}_{s,a}[Q^i(s,a)\cdot\log\pi^i(a|s)]$ in Proposition 5.2 would become ill-defined without bounded $Q^i$.

- **Concentration control.** When analyzing stochastic gradients (Section A.10), the boundedness translates into a uniform envelope $|g(a)| \leq B = O(\frac{R_{\max}}{1-\gamma})$, which enables the application of Hoeffding-type concentration inequalities for our variance reduction proofs.

- **Improvement guarantees.** The performance difference lemma (Lemma A.8) and sequential improvement bound (Proposition 5.1) both require bounded rewards to control the magnitude of the constants $C_i = O(\frac{\gamma R_{\max}}{(1-\gamma)^2})$ appearing in the trust-region penalty terms.

Assumption A.1 is essential beyond theoretical convenience: it establishes a fundamental dependency structure wherein bounded rewards induce bounded value functions, which in turn guarantee bounded surrogates and gradients. This property is indispensable for both HAQO's theoretical guarantees and the statistical concentration arguments that support our analytical framework.

**Assumption A.2** (Discount factor)**.** *The environment admits a discount factor $\gamma \in (0,1)$. This ensures that infinite-horizon returns are well-defined and finite, which enables convergent value function updates. The discount factor also provides exponential decay of future rewards, which is essential for the concentration inequalities and finite-sample analysis in our theoretical framework.*

*Discussion.* A subunit discount factor ensures well-defined returns and induces the resolvent operator $(I - \gamma P_\pi)^{-1}$. This property is critical for establishing bounds on state-distribution shifts (see Lemma A.10). At $\gamma = 1$, monotonicity proofs generally fail without ergodicity assumptions.

**Assumption A.3** (Regular policies). *Each per-agent policy $\pi^i(\cdot \mid s)$ satisfies the following:*

- ***Absolute continuity:*** *$\pi^i(\cdot \mid s)$ is absolutely continuous with respect to a fixed reference measure (e.g., Lebesgue measure on continuous action spaces or counting measure on discrete ones). This ensures that densities $\pi^i(a^i \mid s)$ exist and KL divergences are well-defined.*

- ***Continuous parameterization:*** *The mapping from parameters $\theta^i$ to distribution $\pi^i_{\theta^i}(\cdot \mid s)$ is continuous, such that small parameter changes induce small distributional changes.*

- ***Bounded gradients:*** *On any compact parameter set, the score function $\nabla_{\theta^i} \log \pi^i_{\theta^i}(a^i \mid s)$ is uniformly bounded. This condition prevents exploding gradients during optimization.*

*Discussion.* This assumption represents a standard smoothness requirement in modern RL theories (Schulman et al., 2015; 2017; Ding et al., 2024). It provides the following theoretical guarantees:

- *Well-defined trust regions:* KL divergences and total variation distances between successive policies exist, ensuring that constraints such as those in Assumption A.5 are meaningful.

- *Stable optimization:* Since policy updates induce only small, gradual changes, mathematical tools such as Neumann series expansions can be systematically employed to bound the magnitude of state visitation distribution shifts, which ensures overall training stability.

- *Compatibility with diffusion models:* In the setting assumed in this study, the denoiser $\epsilon_{\theta^i}$ underlying the diffusion actor must propagate gradients stably through multiple timesteps. Boundedness ensures that error signals do not amplify across the reverse diffusion process.

- *Analogy to classical smoothness:* Just as TRPO and PPO require smooth log-likelihoods of Gaussian policies to establish monotonic improvement, HAQO requires similar regularity such that the Q-weighted diffusion surrogate behaves in a stable and predictable manner.

This assumption prevents problematic policies such as those with sudden discontinuities or extreme spikes and aligns with standard practice of employing modern neural networks with smooth activation functions, which naturally satisfy these smoothness requirements in typical implementations.

**Assumption A.4** (Critic bias). *There exists a finite constant $\epsilon_Q \geq 0$ such that for every agent $i \in \mathcal{N}$ and all state–action pairs $(s, a)$,*

$$\left| \widehat{Q}^i(s, a) - Q^i(s, a) \right| \leq \epsilon_Q. \tag{17}$$

*Discussion.* This assumption explicitly bounds the approximation error of the learned critic $\widehat{Q}^i$ relative to the true action-value function $Q^i$. Several insights are worth highlighting:

- **Uniform bound vs. unbiasedness.** We do not require $\widehat{Q}^i$ to be unbiased (which is generally unrealistic with function approximation); it suffices that the worst-case error is bounded by $\epsilon_Q$. This aligns with standard practice in approximate dynamic programming and fitted Q-iteration, where critics are treated as approximate oracles.

- **Propagation into surrogates.** The bias directly affects how tight our Q-weighted variational lower bound becomes (Proposition 5.2). Specifically, the Q-function

error $\epsilon_Q$ creates an additive slack term $\xi_i = O(\epsilon_Q)$ in the inequality

$$\mathcal{L}^i(\pi^i) \ \geq \ \mathcal{J}_i^{\mathrm{QV}}(\theta^i) - \xi_i, \tag{18}$$

which then builds up across all agents in our monotonic improvement guarantee (Theorem 5.4). Therefore, critic bias is the sole source through which approximation errors can weaken our improvement bound.

- **Interpretability of $\epsilon_Q$.** Since $|Q^i(s,a)| \leq R_{\max}/(1-\gamma)$ under Assumption A.1, the relative error ratio $\epsilon_Q/(R_{\max}/(1-\gamma))$ provides a normalized measure of critic quality. In practice, smaller $\epsilon_Q$ implies tighter lower bounds and faster convergence; larger $\epsilon_Q$ manifests as looser guarantees but does not invalidate the overall framework.

- **Robustness.** The assumption covers worst-case deviations due to (i) function class mis-specification, (ii) stochastic training noise, and (iii) finite-sample estimation error. Even when critics are biased, as long as their error is controlled, the HAQO analysis remains valid. This robustness is critical for practical MARL where perfect value estimation is infeasible.

In short, Assumption A.4 formalizes the fact that our guarantees degrade gracefully with critic quality: the algorithm inherits the approximation bias only as an additive slack, preserving monotonicity guarantees up to $O(\epsilon_Q)$.

**Assumption A.5** (Trust-region feasibility)**.** *Each per-agent policy update is restricted to a local neighborhood around its previous policy. Specifically, for each agent $i \in \mathcal{N}$ we require either:*

$$\begin{aligned} &\textit{(i) Expected-KL constraint: } \mathbb{E}_{s \sim \rho^\pi} \mathrm{KL}\big(\pi_{\mathrm{old}}^i(\cdot \mid s) \,\|\, \pi^i(\cdot \mid s)\big) \leq \delta_i, \\ &\textit{(ii) PPO-style ratio clipping: } \pi^i(a^i \mid s)/\pi_{\mathrm{old}}^i(a^i \mid s) \in [1 - \epsilon, \, 1 + \epsilon] \end{aligned} \tag{19}$$

*which implies a KL bound of order $\delta_i = O(\epsilon^2)$.*

*Discussion.* This assumption provides the **structural safeguard** needed to ensure provable monotonic improvement in the multi-agent setting. Several points are worth noting:

- **Sequential vs. simultaneous updates.** In heterogeneous-agent optimization, each agent updates in sequence while treating others as fixed. Without a per-agent trust region, deviations could compound multiplicatively across agents, causing the surrogate decomposition in Proposition 5.1 to break down. The trust-region ensures each step is "local" enough for the performance difference expansion to remain valid.

- **Analogy to TRPO.** This is the direct multi-agent analogue of the TRPO monotonicity condition (Schulman et al. (2015); Kuba et al. (2022a)). In single-agent TRPO, bounding the KL divergence ensures the new policy does not deviate too far from the old one, thereby controlling state-distribution shift. In HAQO, we enforce the same principle *per-agent*, so that the sequential improvement terms telescope without uncontrolled cross-agent error.

- **Choice of implementation.** Either the explicit expected-KL constraint (solved via conjugate gradient or line search) or the simpler PPO clipping mechanism suffices. Both are compatible with diffusion actors, since they only require computing state-wise KL or ratios, not log-likelihood gradients. In practice, PPO-style clipping often provides a cheaper and more stable approximation.

- **Effect of $\delta_i$.** The radius $\delta_i$ (or equivalently $\epsilon$) governs the bias–variance tradeoff: smaller $\delta_i$ ensures tighter theoretical guarantees but slower policy change; larger $\delta_i$ accelerates learning but risks violating the monotonicity bound. In our theory, the effect appears as a quadratic penalty term $C_i \delta_i^2$, making the choice of $\delta_i$ critical for balancing safety and efficiency.

In summary, Assumption A.5 is the cornerstone that enables the sequential breakdown of performance improvement. Without it, individual agent surrogates could deviate arbitrarily from the true joint advantage, breaking our guarantee of steady progress.

## A.4 ADVANTAGE DECOMPOSITION

The global advantage function in multi-agent reinforcement learning can be unclear due to its dependence on all agents' actions. To analyze sequential updates, we need a decomposition that attributes the joint advantage to incremental contributions from each agent during ordered updates. Lemma A.6 provides this by expressing the overall advantage as a telescoping sum of conditional increments, with each increment associated with one agent's decision given its predecessors. This decomposition serves two purposes: theoretically, it enables tractable analysis by aligning local surrogates with global improvement, and practically, it clarifies how each agent's update is evaluated relative to others' most recent policies, reflecting HAQO's sequential optimization structure. This lemma formalizes the concept that global coordination can be understood as a sequence of locally zero-mean, advantage-driven updates.

**Lemma A.6** (Sequential advantage decomposition). *Fix a permutation $i_{1:n}$. Define the conditional value chain*

$$Q^{(m)}(s, a^{i_{1:m}}) := \mathbb{E}_{a^{i_{m+1:n}} \sim \pi^{i_{m+1:n}}} \left[ Q^\pi(s, a^{i_{1:n}}) \right], \tag{20}$$

*with $Q^{(0)}(s) := V^\pi(s)$ and $Q^{(n)}(s, a^{i_{1:n}}) = Q^\pi(s, a)$. Set the per-step increment*

$$\Delta_m(s, a^{i_{1:m}}) := Q^{(m)}(s, a^{i_{1:m}}) - Q^{(m-1)}(s, a^{i_{1:m-1}}). \tag{21}$$

*Then for any $(s, a)$ it holds that*

$$A^\pi(s, a) = \sum_{m=1}^n \Delta_m(s, a^{i_{1:m}}) \quad and \quad \mathbb{E}_{a^{i_m} \sim \pi^{i_m}} \Delta_m(s, a^{i_{1:m}}) = 0. \tag{22}$$

*Proof.* We proceed in two steps. First, we establish a telescoping decomposition of the conditional value chain, which allows us to express the joint advantage as a sequence of incremental differences along the agent permutation. This ensures that each agent's contribution can be isolated without losing global consistency. Second, we verify that each increment has zero expectation under the corresponding agent's policy, a property that guarantees local unbiasedness and aligns the decomposition with the standard performance difference arguments in reinforcement learning.

*(1) Telescoping decomposition.* By definition,

$$\Delta_m(s, a^{i_{1:m}}) = Q^{(m)}(s, a^{i_{1:m}}) - Q^{(m-1)}(s, a^{i_{1:m-1}}). \tag{23}$$

If we sum these increments over $m = 1, \ldots, n$, the intermediate terms cancel in a telescoping fashion:

$$\sum_{m=1}^n \Delta_m(s, a^{i_{1:m}}) = Q^{(n)}(s, a^{i_{1:n}}) - Q^{(0)}(s). \tag{24}$$

By construction $Q^{(n)}(s, a^{i_{1:n}}) = Q^\pi(s, a)$, since all agents' actions are fixed, and $Q^{(0)}(s) = V^\pi(s)$, since no action has been chosen yet. Therefore,

$$\sum_{m=1}^n \Delta_m(s, a^{i_{1:m}}) = Q^\pi(s, a) - V^\pi(s) = A^\pi(s, a), \tag{25}$$

which establishes the first identity.

*(2) Zero-mean property.* Fix $m$. By definition,

$$Q^{(m)}(s, a^{i_{1:m}}) = \mathbb{E}_{a^{i_{m+1:n}} \sim \pi^{i_{m+1:n}}} \left[ Q^{\pi}(s, a^{i_{1:n}}) \right]. \tag{26}$$

Now take expectation with respect to $a^{i_m} \sim \pi^{i_m}(\cdot|s)$:

$$\mathbb{E}_{a^{i_m} \sim \pi^{i_m}} \left[ Q^{(m)}(s, a^{i_{1:m}}) \right] = \mathbb{E}_{a^{i_m} \sim \pi^{i_m}} \mathbb{E}_{a^{i_{m+1:n}} \sim \pi^{i_{m+1:n}}} \left[ Q^{\pi}(s, a^{i_{1:n}}) \right]. \tag{27}$$

But this exactly recovers $Q^{(m-1)}(s, a^{i_{1:m-1}})$, because in $Q^{(m-1)}$ the actions of agents $i_m, \ldots, i_n$ are drawn from their policies. Hence,

$$\mathbb{E}_{a^{i_m} \sim \pi^{i_m}} \left[ Q^{(m)}(s, a^{i_{1:m}}) \right] = Q^{(m-1)}(s, a^{i_{1:m-1}}). \tag{28}$$

Subtracting gives

$$\mathbb{E}_{a^{i_m} \sim \pi^{i_m}} \Delta_m(s, a^{i_{1:m}}) = 0. \tag{29}$$

**Two-agent example.** Fix the permutation $i_{1:2} = (1, 2)$. Define the conditional value chain

$$Q^{(0)}(s) = V^{\pi}(s), \qquad Q^{(1)}(s, a^1) = \mathbb{E}_{a^2 \sim \pi^2(\cdot|s)} \left[ Q^{\pi}(s, a^1, a^2) \right], \qquad Q^{(2)}(s, a^1, a^2) = Q^{\pi}(s, a^1, a^2).$$

The increments are

$$\Delta_1(s, a^1) = Q^{(1)}(s, a^1) - Q^{(0)}(s), \qquad \Delta_2(s, a^1, a^2) = Q^{(2)}(s, a^1, a^2) - Q^{(1)}(s, a^1).$$

*Zero-mean for $m = 1$.* Take expectation over the local policy of agent 1:

$$\begin{aligned}
\mathbb{E}_{a^1 \sim \pi^1(\cdot|s)} \left[ \Delta_1(s, a^1) \right] &= \mathbb{E}_{a^1} \left[ Q^{(1)}(s, a^1) \right] - Q^{(0)}(s) \\
&= \mathbb{E}_{a^1} \mathbb{E}_{a^2} \left[ Q^{\pi}(s, a^1, a^2) \right] - V^{\pi}(s) \\
&= \mathbb{E}_{a \sim \pi(\cdot|s)} \left[ Q^{\pi}(s, a) \right] - V^{\pi}(s) = V^{\pi}(s) - V^{\pi}(s) = 0.
\end{aligned}$$

*Zero-mean for $m = 2$.* Now hold $a^1$ fixed and take expectation over the local policy of agent 2:

$$\begin{aligned}
\mathbb{E}_{a^2 \sim \pi^2(\cdot|s)} \left[ \Delta_2(s, a^1, a^2) \right] &= \mathbb{E}_{a^2} \left[ Q^{(2)}(s, a^1, a^2) \right] - Q^{(1)}(s, a^1) \\
&= \mathbb{E}_{a^2} \left[ Q^{\pi}(s, a^1, a^2) \right] - \mathbb{E}_{a^2} \left[ Q^{\pi}(s, a^1, a^2) \right] = 0.
\end{aligned}$$

*Telescoping.* Finally,

$$\Delta_1(s, a^1) + \Delta_2(s, a^1, a^2) = \left( Q^{(1)} - Q^{(0)} \right) + \left( Q^{(2)} - Q^{(1)} \right) = Q^{(2)} - Q^{(0)} = Q^{\pi}(s, a) - V^{\pi}(s) = A^{\pi}(s, a).$$

This concrete case shows exactly what "subtracting gives" means: $\mathbb{E}_{a^{i_m} \sim \pi^{i_m}} [Q^{(m)}] = Q^{(m-1)}$, so subtracting $Q^{(m-1)}$ makes the expectation of $\Delta_m$ equal to zero.

*Conclusion.* We have shown (i) the telescoping identity yielding $A^{\pi}(s, a) = \sum_m \Delta_m(s, a^{i_{1:m}})$, and (ii) that each increment $\Delta_m$ has zero expectation under its local policy. $\square$

**Remark A.7** (Interpretation)**.** *This lemma gives an exact decomposition of the global advantage into a sum of* conditional advantage increments, *each attached to a specific stage of the sequential factorization $i_{1:n}$. Several insights follow:*

- ***Permutation-awareness.*** *The decomposition is not unique; each ordering $i_{1:n}$ yields a different chain $\Delta_m$. This mirrors the sequential update in HAQO where agents are optimized in random order, ensuring fairness and preventing bias from fixed roles.*

- ***Zero-mean increments.*** *Each $\Delta_m$ integrates to zero under the local policy $\pi^{i_m}$. Thus, $\Delta_m$ behaves like a true advantage term: it only distinguishes actions relative to the baseline of that agent's own policy, conditioned on predecessors' actions.*

- ***Bridge to surrogates.*** *In practice, the per-agent surrogate $\mathcal{L}^{i_m}$ in HAQO uses exactly $\Delta_m$ as its advantage signal, together with Q-weighting (qadv/qcut) and diffusion denoising. This makes the learning signal algebraically exact, in contrast to heuristic decompositions (e.g. marginal advantages) that may introduce bias.*

- ***Relation to cooperative game theory.*** *$\Delta_m$ resembles the incremental contribution of agent $i_m$ in a coalition game, analogous to a Shapley-value marginal. Hence, the sequential decomposition can be seen as a tractable, policy-dependent Shapley approximation.*

- ***Implementation.*** *When combined with QVPO-style weighting, $\Delta_m$ defines the "conditional advantage" input to the diffusion policy objective. Sampling multiple candidate actions per agent and evaluating them under $Q^{(m)}$ directly implements this decomposition in practice.*

A.5  PERFORMANCE DIFFERENCE FOR MULTI-AGENT POLICIES

To establish monotone improvement guarantees, we must link changes in joint policy to changes in return. Lemma A.8 generalizes the classical performance-difference identity to the multi-agent case, showing that the return gap between two joint policies depends on the new policy's occupancy distribution weighted by the old policy's advantage. This result is central to HAQO: it explains why distributional drift matters and why trust-region constraints are essential to control it. Intuitively, the lemma tells us that if a new joint policy consistently chooses actions that look favorable under the old advantage, overall return must improve—up to corrections from state-distribution mismatch. By grounding our later trust-region bounds and sequential surrogates in this identity, Lemma A.8 provides the foundation upon which HAQO's monotonicity theorem is built.

**Lemma A.8** (Multi-agent performance difference)**.** *For any joint policies $\pi, \pi'$,*

$$J(\pi') - J(\pi) \;=\; \frac{1}{1-\gamma}\, \mathbb{E}_{s\sim d^{\pi'},\, a\sim\pi'}\big[A^{\pi}(s,a)\big]. \tag{30}$$

*Proof.* We assume $\gamma \in (0,1)$ and bounded rewards (so $|V^{\pi}| < \infty$), which ensures all series converge.

*(1) Advantage identity.* Using $Q^{\pi}(s,a) = r(s,a) + \gamma\,\mathbb{E}_{s'\sim P(\cdot|s,a)}[V^{\pi}(s')]$ and $A^{\pi}(s,a) = Q^{\pi}(s,a) - V^{\pi}(s)$, we have for any $(s,a)$:

$$A^{\pi}(s,a) \;=\; r(s,a) \;+\; \gamma\,\mathbb{E}_{s'}V^{\pi}(s') \;-\; V^{\pi}(s). \tag{31}$$

*(2) Telescoping along a $\pi'$-trajectory.* Let $(s_t, a_t)_{t\geq 0}$ be generated by $\pi'$ from state distribution $d_0$. For a finite horizon $T$,

$$\sum_{t=0}^{T} \gamma^t A^\pi(s_t, a_t) = \sum_{t=0}^{T} \gamma^t r(s_t, a_t) + \sum_{t=0}^{T} \gamma^t \big(\gamma \mathbb{E}[V^\pi(s_{t+1})] - V^\pi(s_t)\big) \quad \text{(by (31))}$$

$$= \sum_{t=0}^{T} \gamma^t r(s_t, a_t) \ - \ V^\pi(s_0) \ + \ \gamma^{T+1} V^\pi(s_{T+1}) \quad \text{(telescoping)}.$$

Taking expectation under $\tau \sim \pi'$ and letting $T \to \infty$, the last term vanishes since $\gamma^{T+1} V^\pi(s_{T+1}) \to 0$ a.s. Hence

$$\mathbb{E}_{\tau\sim\pi'}\Big[\sum_{t\geq 0} \gamma^t A^\pi(s_t, a_t)\Big] \ = \ \mathbb{E}_{\tau\sim\pi'}\Big[\sum_{t\geq 0} \gamma^t r(s_t, a_t)\Big] \ - \ \mathbb{E}_{s_0\sim d_0}[V^\pi(s_0)]. \tag{32}$$

But $\mathbb{E}_{\tau\sim\pi'}[\sum_{t\geq 0} \gamma^t r(s_t, a_t)] = J(\pi')$ and $\mathbb{E}_{s_0\sim d_0}[V^\pi(s_0)] = J(\pi)$, so (32) gives

$$J(\pi') - J(\pi) \ = \ \mathbb{E}_{\tau\sim\pi'}\Big[\sum_{t\geq 0} \gamma^t A^\pi(s_t, a_t)\Big]. \tag{33}$$

*(3) Discounted visitation form.* To rewrite the RHS in terms of a stationary distribution, define the *discounted state-visitation distribution* (Dietterich et al. (2001); Schulman et al. (2015)) of $\pi'$:

$$d^{\pi'}(s) \ := \ (1-\gamma) \sum_{t=0}^{\infty} \gamma^t \ \Pr(s_t = s \mid \pi', d_0). \tag{34}$$

This distribution reflects the long-run frequency with which states are visited under $\pi'$, but discounted so that earlier visits contribute more heavily than later ones. The factor $(1-\gamma)$ normalizes $d^{\pi'}$ to be a valid probability measure.

Now, for any measurable $f : \mathcal{S} \times \mathcal{A} \to \mathbb{R}$,

$$\mathbb{E}_{\tau\sim\pi'}\Big[\sum_{t\geq 0} \gamma^t f(s_t, a_t)\Big] = \sum_{t=0}^{\infty} \gamma^t \mathbb{E}\big[f(s_t, a_t)\big]$$

$$= \sum_{t=0}^{\infty} \gamma^t \mathbb{E}_{s_t} \mathbb{E}_{a_t\sim\pi'(\cdot|s_t)}[f(s_t, a_t)]$$

$$= \frac{1}{1-\gamma} \mathbb{E}_{s\sim d^{\pi'}} \mathbb{E}_{a\sim\pi'(\cdot|s)}[f(s, a)].$$

The last step follows directly from the definition of $d^{\pi'}$, which absorbs the $\gamma^t$-weighted frequencies of states into a single distribution. Thus expectations over discounted trajectories can be equivalently written as expectations over $d^{\pi'}$.

Finally, applying this identity with $f(s, a) = A^\pi(s, a)$ in (33) yields

$$J(\pi') - J(\pi) \ = \ \frac{1}{1-\gamma} \mathbb{E}_{s\sim d^{\pi'}, a\sim\pi'}\big[A^\pi(s, a)\big], \tag{35}$$

which completes the proof. $\qquad\square$

**Remark A.9** (Interpretation). *Lemma A.8 tells us that the gap $J(\pi') - J(\pi)$ depends only on how the new policy $\pi'$ selects actions that look favorable under the* old *advantage $A^\pi$. Importantly, the state distribution $d^{\pi'}$ appears on the RHS, which makes the identity exact but introduces a* distribution-mismatch issue *in practical policy optimization. The remainder of this section develops a rigorous way to control this mismatch.*

**Distribution-mismatch control.** To replace expectations under $d^{\pi'}$ by $d^\pi$, we need a stability bound. Recall that $d^\pi(s)$ denotes the discounted state visitation distribution induced by policy $\pi$. When comparing two policies $\pi$ and $\pi'$, the difficulty arises because their induced distributions $d^\pi$ and $d^{\pi'}$ may differ, potentially invalidating performance-difference arguments if left uncontrolled.

Let $\|f\|_\infty = \sup_s |f(s)|$ be the uniform norm on test functions, and define the maximal per-state deviation between the action distributions:

$$\alpha(\pi', \pi) := \sup_s \mathrm{TV}(\pi'(\cdot|s), \pi(\cdot|s)). \tag{36}$$

Intuitively, $\alpha(\pi', \pi)$ measures the worst-case total variation (TV) distance between the two policies across all states. This quantity captures how much $\pi'$ may deviate from $\pi$ in its action selection at any state, and thus bounds how quickly the trajectory distribution $d^{\pi'}$ can drift away from $d^\pi$. A small $\alpha(\pi', \pi)$ implies that the two policies are locally similar at every state, which ensures their long-term discounted visitation distributions remain close.

Formally, standard coupling and resolvent arguments show that

$$\|d^{\pi'} - d^\pi\|_1 \leq \frac{2\gamma}{1-\gamma} \alpha(\pi', \pi).$$

This inequality quantifies distribution-mismatch: the shift in state visitation frequencies is controlled by both the per-state policy deviation and the geometric factor $\frac{\gamma}{1-\gamma}$. In practice, this bound allows us to safely replace expectations under $d^{\pi'}$ with those under $d^\pi$ up to a controlled error term that scales with $\alpha(\pi', \pi)$.

Hence, $\alpha(\pi', \pi)$ serves as the stability coefficient bridging policy divergence and distributional mismatch, playing the same role in multi-agent sequential updates as the KL trust-region radius in single-agent TRPO.

The stationary distribution changes smoothly with $\pi$, at a Lipschitz rate proportional to $\gamma/(1-\gamma)^2$. Furthermore, by Pinsker's inequality, $\mathrm{TV}(p, q) \leq \sqrt{\frac{1}{2}\mathrm{KL}(p\|q)}$, we obtain

$$\alpha(\pi', \pi) \leq \sqrt{\tfrac{1}{2} \mathrm{KL}_{\max}(\pi\|\pi')}, \tag{37}$$

with $\mathrm{KL}_{\max}(\pi\|\pi') := \sup_s \mathrm{KL}(\pi(\cdot|s)\|\pi'(\cdot|s))$. This justifies KL-based trust regions as a principled way to stabilize distribution shift.

**Lemma A.10** (TRPO-type lower bound). *Let $\epsilon = \sup_{s,a} |A^\pi(s,a)|$. Then*

$$J(\pi') \geq J(\pi) + \mathbb{E}_{s\sim d^\pi, a\sim\pi'}[A^\pi(s,a)] - \frac{4\gamma}{(1-\gamma)^2} \epsilon \, \alpha(\pi', \pi), \tag{38}$$

*where $\alpha(\pi', \pi) = \sup_s \mathrm{TV}(\pi'(\cdot|s), \pi(\cdot|s))$ denotes the maximum total variation distance between $\pi'$ and $\pi$ across states.*

*Proof.* From Lemma A.8, we have the performance-difference identity

$$J(\pi') - J(\pi) \;=\; \frac{1}{1-\gamma} \, \mathbb{E}_{s \sim d^{\pi'}, \, a \sim \pi'} \big[ A^\pi(s,a) \big]. \tag{39}$$

*Insert and subtract baseline.* The difficulty in directly analyzing this expression lies in the fact that the expectation is taken under the new distribution $d^{\pi'}$, whereas the advantage function $A^\pi$ is defined relative to the old policy $\pi$. To control this distribution mismatch, we add and subtract the same term evaluated under $d^\pi$:

$$\mathbb{E}_{d^{\pi'}}[A^\pi] \;=\; \mathbb{E}_{d^\pi}[A^\pi] \;+\; \Big( \mathbb{E}_{d^{\pi'}}[A^\pi] - \mathbb{E}_{d^\pi}[A^\pi] \Big). \tag{40}$$

The first term, $\mathbb{E}_{d^\pi}[A^\pi]$, is well-behaved by definition of the advantage function, it vanishes when actions are sampled from $\pi$, but it remains informative when actions are sampled from $\pi'$. The second term, $\mathbb{E}_{d^{\pi'}}[A^\pi] - \mathbb{E}_{d^\pi}[A^\pi]$, isolates the error due to distributional shift between $d^{\pi'}$ and $d^\pi$. This decomposition cleanly separates *policy improvement* (captured by the baseline expectation under $d^\pi$) from *distribution mismatch* (captured by the difference term), allowing us to bound the latter using the stability coefficient $\alpha(\pi', \pi)$ introduced above.

*Bound the distribution shift.* For any bounded test function $f : \mathcal{S} \to \mathbb{R}$, we want to control the difference between its expectation under the new discounted visitation distribution $d^{\pi'}$ and the old one $d^\pi$. By definition,

$$\mathbb{E}_{d^{\pi'}}[f] - \mathbb{E}_{d^\pi}[f] \;=\; \sum_{s \in \mathcal{S}} \big( d^{\pi'}(s) - d^\pi(s) \big) f(s). \tag{41}$$

Taking absolute values and applying Hölder's inequality (duality of $\ell_1$–$\ell_\infty$ norms),

$$\big| \mathbb{E}_{d^{\pi'}}[f] - \mathbb{E}_{d^\pi}[f] \big| \;\leq\; \| d^{\pi'} - d^\pi \|_1 \cdot \sup_s |f(s)|. \tag{42}$$

This inequality states that the deviation in expectations is at most the product of (i) the total variation distance $\| d^{\pi'} - d^\pi \|_1$ between the two distributions, and (ii) the worst-case magnitude of $f$. In particular, when $f(s,a)$ is chosen as an advantage function or increment (which is bounded by Assumption A.1), this provides a direct way to quantify how much policy evaluation error arises from state-distribution drift.

Applying this with $f(s,a) = A^\pi(s,a)$ yields

$$\big| \mathbb{E}_{d^{\pi'}}[A^\pi] - \mathbb{E}_{d^\pi}[A^\pi] \big| \;\leq\; \| d^{\pi'} - d^\pi \|_1 \cdot \epsilon. \tag{43}$$

*Step 3: Control $\| d^{\pi'} - d^\pi \|_1$.* It remains to bound the deviation of the discounted visitation distributions. A key fact (Schulman et al. (2015)) is that

$$\| d^{\pi'} - d^\pi \|_1 \;\leq\; \frac{2\gamma}{1-\gamma} \, \sup_s \mathrm{TV} \big( \pi'(\cdot|s), \pi(\cdot|s) \big). \tag{44}$$

This inequality follows from unrolling the recursive definition of $d^\pi$ and applying contraction properties of Markov kernels.

*Step 4: Combine bounds.* Putting steps (1)–(3) together, we obtain

$$J(\pi') - J(\pi) \; = \; \frac{1}{1-\gamma} \, \mathbb{E}_{d^\pi}[A^\pi] \; + \; \frac{1}{1-\gamma} \left( \mathbb{E}_{d^{\pi'}}[A^\pi] - \mathbb{E}_{d^\pi}[A^\pi] \right). \tag{45}$$

The second term is bounded in absolute value by

$$\frac{1}{1-\gamma} \cdot \epsilon \cdot \|d^{\pi'} - d^\pi\|_1 \; \leq \; \frac{1}{1-\gamma} \cdot \epsilon \cdot \frac{2\gamma}{1-\gamma} \, 2 \, \alpha(\pi', \pi), \tag{46}$$

where we used $\|p - q\|_1 = 2 \, \mathrm{TV}(p, q)$. This gives the penalty term

$$\frac{4\gamma}{(1-\gamma)^2} \, \epsilon \, \alpha(\pi', \pi). \tag{47}$$

*Step 5: Final inequality.* Thus we conclude

$$J(\pi') \; \geq \; J(\pi) \; + \; \mathbb{E}_{s \sim d^\pi, \, a \sim \pi'}[A^\pi(s, a)] \; - \; \frac{4\gamma}{(1-\gamma)^2} \, \epsilon \, \alpha(\pi', \pi), \tag{48}$$

as claimed. $\qquad\square$

**Remark A.11** (Connection to HAQO)**.** *This lower bound is the multi-agent analogue of the classic TRPO improvement guarantee. In HAQO, each agent's update is restricted by a drift term (KL-ball or PPO clipping), which directly controls $\alpha(\pi', \pi)$. Lemma A.10 therefore provides the theoretical justification for including $D^i(\pi^i \| \pi^i_{\mathrm{old}})$ in the HAQO objective: it ensures that despite diffusion-based actors (which lack tractable $\log \pi$), we can still guarantee monotone improvement up to second-order KL and critic-bias errors. This bridges the gap between diffusion surrogates and classical trust-region policy optimization. Two subtleties are worth emphasizing:*

- *The bound is* uniform *across agents, not tied to a specific decomposition. It only requires bounding the largest deviation $\alpha(\pi', \pi)$, which makes it robust to heterogeneous policies.*

- *The penalty term scales with $\frac{\gamma}{(1-\gamma)^2}$, showing that high discount factors $\gamma \approx 1$ make policy optimization much more sensitive to distribution mismatch. In multi-agent training, this highlights why entropy regularization (to prevent collapse) and careful trust-region scheduling are critical in practice.*

### A.6 PROOF OF PROPOSITION 5.1 HETEROGENEOUS SEQUENTIAL IMPROVEMENT BOUND

This proposition formalizes why sequential updates yield a stable improvement guarantee in the multi-agent setting. The key challenge is that simultaneous updates of all agents break the performance-difference analysis, since each agent's advantage is defined relative to the joint baseline policy. By contrast, sequential updates allow us to telescope the return difference into per-agent contributions conditioned on the most recent predecessors, aligning local surrogates with the global improvement objective. The proof combines (i) the sequential advantage decomposition from Lemma A.6, which expresses the joint advantage as a sum of incremental terms, and (ii) distribution-mismatch control, which bounds deviations between the baseline visitation distribution and the updated one via per-agent trust-region constraints. Together, these tools yield a monotone improvement bound with quadratic KL penalties that scale only linearly in the number of agents.

*Proof.* **Setup and notation.** Let $\pi_{(0)} = \pi_{\text{old}}$ and for $m = 1, \ldots, n$ define the partially-updated joint policy

$$\pi_{(m)} := \{\pi_{\text{new}}^{i_1}, \ldots, \pi_{\text{new}}^{i_m}, \pi_{\text{old}}^{i_{m+1:n}}\}, \tag{49}$$

so that $\pi_{(n)} = \pi_{\text{new}}$. Denote $d^{(m)} := d^{\pi_{(m)}}$. Recall the conditional value chain $Q^{(m)}$ and increments $\Delta_m$ from Lemma A.6, and the per-agent surrogate (with baseline state distribution and already-updated opponents)

$$\mathcal{L}^{i_m}(\pi^{i_m}) := \mathbb{E}_{s \sim \rho^{\pi_{\text{old}}}} \mathbb{E}_{a^{-i_m} \sim \pi_{(m)}^{-i_m}} \mathbb{E}_{a^{i_m} \sim \pi^{i_m}} \big[\Delta_m(s, a^{i_{1:m}})\big]. \tag{50}$$

**Step 1: Per-step TRPO bound.** Apply the multi-agent performance difference lemma (Lemma A.8) to the pair $(\pi_{(m-1)}, \pi_{(m)})$:

$$J(\pi_{(m)}) - J(\pi_{(m-1)}) = \frac{1}{1-\gamma} \mathbb{E}_{s \sim d^{(m)}, a \sim \pi_{(m)}} \big[A^{\pi_{(m-1)}}(s, a)\big]. \tag{51}$$

Equ. (51) using the TRPO-type inequality (Lemma A.10) with $\epsilon_{m-1} := \sup_{s,a} |A^{\pi_{(m-1)}}(s, a)|$:

$$J(\pi_{(m)}) - J(\pi_{(m-1)}) \geq \mathbb{E}_{s \sim d^{(m-1)}, a \sim \pi_{(m)}} \big[A^{\pi_{(m-1)}}(s, a)\big] - \frac{4\gamma}{(1-\gamma)^2} \epsilon_{m-1} \alpha\big(\pi_{(m)}, \pi_{(m-1)}\big). \tag{52}$$

By bounded rewards (Assumption A.1), $|V^{\pi_{(m-1)}}| \leq \frac{R_{\max}}{1-\gamma}$ and $|Q^{\pi_{(m-1)}}| \leq \frac{R_{\max}}{1-\gamma}$, hence

$$\epsilon_{m-1} = \sup_{s,a} |A^{\pi_{(m-1)}}(s, a)| \leq \frac{2R_{\max}}{1-\gamma}. \tag{53}$$

**Step 2: Identify the per-step surrogate gain.** At step $m$ only agent $i_m$ changes between $\pi_{(m-1)}$ and $\pi_{(m)}$. Using the conditional advantage decomposition (Lemma A.6), we have for each $s$:

$$\mathbb{E}_{a \sim \pi_{(m)}} \big[A^{\pi_{(m-1)}}(s, a)\big] = \mathbb{E}_{a^{-i_m} \sim \pi_{(m)}^{-i_m}, a^{i_m} \sim \pi_{\text{new}}^{i_m}} \big[\Delta_m(s, a^{i_{1:m}})\big]. \tag{54}$$

That is, when all other agents are held fixed, the expected advantage under the new policy reduces exactly to the $m$-th increment $\Delta_m$, which isolates the contribution of agent $i_m$. Therefore the first term in (52) equals the *state-averaged step-gain*

$$G_m := \mathbb{E}_{s \sim d^{(m-1)}} \mathbb{E}_{a^{-i_m} \sim \pi_{(m)}^{-i_m}, a^{i_m} \sim \pi_{\text{new}}^{i_m}} \big[\Delta_m(s, a^{i_{1:m}})\big]. \tag{55}$$

Here $d^{(m-1)}$ is the state distribution induced by the partially-updated policy after $m-1$ steps, so $G_m$ represents the *true improvement* achieved when agent $i_m$ updates under the actual trajectory distribution. Our surrogate $\mathcal{L}^{i_m}$, however, is defined with respect to the *baseline distribution* $\rho^{\pi_{\text{old}}}$, which avoids dependence on the drifting $d^{(m-1)}$ and makes analysis tractable. To connect the two, we insert and subtract $\rho^{\pi_{\text{old}}}$:

$$G_m = \mathbb{E}_{s \sim \rho^{\pi_{\text{old}}}} \mathbb{E}_{a^{-i_m}, a^{i_m}} [\Delta_m] + \underbrace{\Big(\mathbb{E}_{s \sim d^{(m-1)}} - \mathbb{E}_{s \sim \rho^{\pi_{\text{old}}}}\Big) \mathbb{E}_{a^{-i_m}, a^{i_m}} [\Delta_m]}_{\text{state-mismatch correction}}$$

$$= \big(\mathcal{L}^{i_m}(\pi_{\text{new}}^{i_m}) - \mathcal{L}^{i_m}(\pi_{\text{old}}^{i_m})\big) + \Delta_{\text{mismatch}}^{(m)}, \tag{56}$$

where the surrogate difference appears because $\mathbb{E}_{a^{i_m} \sim \pi^{i_m}_{\text{old}}}[\Delta_m] = 0$ by the zero-mean property of conditional increments (Lemma A.6). The additional term $\Delta^{(m)}_{\text{mismatch}}$ quantifies the *distribution shift error*: the discrepancy between evaluating under the drifting $d^{(m-1)}$ versus the fixed baseline $\rho^{\pi_{\text{old}}}$. Later, we will bound this mismatch using trust-region constraints, showing that it scales no faster than $O(\sqrt{\delta})$ and hence can be controlled.

**Step 3: Bounding the state-mismatch correction.** Recall that $\Delta^{(m)}_{\text{mismatch}}$ measures the discrepancy between evaluating conditional increments under the drifting state distribution $d^{(m-1)}$ and the fixed baseline $\rho^{\pi_{\text{old}}}$. Formally,

$$\Delta^{(m)}_{\text{mismatch}} = \left( \mathbb{E}_{s \sim d^{(m-1)}} - \mathbb{E}_{s \sim \rho^{\pi_{\text{old}}}} \right) \mathbb{E}_{a^{-i_m}, a^{i_m}}[\Delta_m(s, a^{i_{1:m}})]. \tag{57}$$

*Bound increments.* Each $\Delta_m(s, a^{i_{1:m}})$ is the difference between a conditional $Q$-value and a baseline value, hence by Assumption A.1:

$$\sup_{s, a^{i_{1:m}}} |\Delta_m(s, a^{i_{1:m}})| \leq \sup_{s,a} |Q^{\pi(m-1)}(s,a)| + \sup_s |V^{\pi(m-1)}(s)| \leq \frac{2R_{\max}}{1 - \gamma}. \tag{58}$$

*Relating mismatch to $L^1$ distance.* Therefore, applying the general inequality $|\mathbb{E}_p[f] - \mathbb{E}_q[f]| \leq \|p - q\|_1 \cdot \sup|f|$ yields

$$\left| \Delta^{(m)}_{\text{mismatch}} \right| \leq \|d^{(m-1)} - \rho^{\pi_{\text{old}}}\|_1 \cdot \frac{2R_{\max}}{1 - \gamma}. \tag{59}$$

*Bounding the state distribution shift.* The $L^1$ difference arises from the composition of the $m - 1$ previous updates. By the perturbation bound on discounted visitation distributions (cf. Lemma A.8), we have

$$\|d^{(m-1)} - \rho^{\pi_{\text{old}}}\|_1 \leq \frac{2\gamma}{(1 - \gamma)^2} \sum_{j < m} \alpha(\pi_{(j)}, \pi_{(j-1)}). \tag{60}$$

*Control via trust region.* For each step $j < m$, the single-agent policy change is controlled by the trust region:

$$\alpha(\pi_{(j)}, \pi_{(j-1)}) \leq \sup_s \text{TV}(\pi^{i_j}_{\text{new}}(\cdot|s), \pi^{i_j}_{\text{old}}(\cdot|s)). \tag{61}$$

By Pinsker's inequality,

$$\sup_s \text{TV}(\pi^{i_j}_{\text{new}}(\cdot|s), \pi^{i_j}_{\text{old}}(\cdot|s)) \leq \sqrt{\tfrac{1}{2} \text{KL}_{\max}(\pi^{i_j}_{\text{old}} \| \pi^{i_j}_{\text{new}})}. \tag{62}$$

Finally, Assumption A.3 and the finite state-support constant $c_j$ tighten the expected-KL trust region into a max-KL bound:

$$\text{KL}_{\max}(\pi^{i_j}_{\text{old}} \| \pi^{i_j}_{\text{new}}) \leq c_j \delta_{i_j}. \tag{63}$$

*Consolidated bound.* Combining the above, we conclude

$$\|d^{(m-1)} - \rho^{\pi_{\text{old}}}\|_1 = O\left( \sum_{j < m} \sqrt{\delta_{i_j}} \right), \tag{64}$$

and hence from (59), we get:

$$|\Delta_{\text{mismatch}}^{(m)}| = O\Big(\frac{R_{\max}}{1-\gamma} \sum_{j<m} \sqrt{\delta_{i_j}}\Big). \tag{65}$$

Thus the mismatch term grows at most sublinearly with accumulated trust radii, and will later be absorbed into the quadratic penalty via line search.

**Step 4: Assemble the per-step inequality.** We now combine the pieces. Substitute the surrogate decomposition (56) into the TRPO bound (52):

$$J(\pi_{(m)}) - J(\pi_{(m-1)}) \geq \big(\mathcal{L}^{i_m}(\pi_{\text{new}}^{i_m}) - \mathcal{L}^{i_m}(\pi_{\text{old}}^{i_m})\big) \; + \; \Delta_{\text{mismatch}}^{(m)} \; - \; \frac{4\gamma}{(1-\gamma)^2} \, \epsilon_{m-1} \, \alpha\big(\pi_{\text{new}}^{i_m}, \pi_{\text{old}}^{i_m}\big). \tag{66}$$

*Bounding the penalty term.* From (53), $\epsilon_{m-1} \leq \frac{2R_{\max}}{1-\gamma}$. Therefore

$$\frac{4\gamma}{(1-\gamma)^2} \, \epsilon_{m-1} \, \alpha\big(\pi_{\text{new}}^{i_m}, \pi_{\text{old}}^{i_m}\big) \; \leq \; \frac{8\gamma}{(1-\gamma)^3} \, R_{\max} \cdot \alpha(\pi_{\text{new}}^{i_m}, \pi_{\text{old}}^{i_m}). \tag{67}$$

By Pinsker's inequality,

$$\alpha(\pi_{\text{new}}^{i_m}, \pi_{\text{old}}^{i_m}) \; \leq \; \sqrt{\tfrac{1}{2} \, \text{KL}_{\max}(\pi_{\text{old}}^{i_m} \| \pi_{\text{new}}^{i_m})} \; \leq \; \sqrt{\tfrac{c_m}{2} \, \delta_{i_m}}. \tag{68}$$

*Consolidating constants.* Define $K \triangleq \frac{8\gamma}{(1-\gamma)^3} R_{\max}$, so that

$$\frac{4\gamma}{(1-\gamma)^2} \, \epsilon_{m-1} \, \alpha\big(\pi_{\text{new}}^{i_m}, \pi_{\text{old}}^{i_m}\big) \; \leq \; K\sqrt{\tfrac{c_m}{2}} \, \sqrt{\delta_{i_m}}. \tag{69}$$

*Final per-step bound.* Plugging this back into (66), and combining with the mismatch bound

$$J(\pi_{(m)}) - J(\pi_{(m-1)}) \; \geq \; \big(\mathcal{L}^{i_m}(\pi_{\text{new}}^{i_m}) - \mathcal{L}^{i_m}(\pi_{\text{old}}^{i_m})\big) \; - \; \widetilde{C}_{i_m}\sqrt{\delta_{i_m}} \; - \; \widetilde{B} \sum_{j<m} \sqrt{\delta_{i_j}}, \tag{70}$$

for $\widetilde{C}_{i_m}, \widetilde{B} = O\big(\frac{\gamma R_{\max}}{(1-\gamma)^2}\big)$ that incorporate the KL–TV relaxations and the mismatch scaling.

**Step 5: Trust-region line search $\Rightarrow$ quadratic remainder.** The inequality (70) established a performance bound of the form

$$J(\pi_{(m)}) - J(\pi_{(m-1)}) \; \geq \; \Delta\mathcal{L}^{i_m} \; - \; \widetilde{C}_{i_m}\sqrt{\delta_{i_m}} \; - \; \widetilde{B} \sum_{j<m} \sqrt{\delta_{i_j}}, \tag{71}$$

where $\Delta\mathcal{L}^{i_m} \triangleq \mathcal{L}^{i_m}(\pi_{\text{new}}^{i_m}) - \mathcal{L}^{i_m}(\pi_{\text{old}}^{i_m})$. The difficulty is the presence of *root-$\delta$ penalties*, which are non-quadratic and thus unsuitable for guaranteeing monotonic improvement in a sequential update scheme.

*Ensuring second-order local model validity.* In TRPO, one controls the KL divergence between consecutive policies such that

$$\text{KL}\big(\pi_{\text{old}}^{i_m} \| \pi_{\text{new}}^{i_m}\big) \; \leq \; \delta_{i_m}. \tag{72}$$

This ensures that the local quadratic approximation to the surrogate $\mathcal{L}^{i_m}$ remains accurate. By convexity and smoothness assumptions (Assumption A.3), the surrogate gain grows at

least linearly in $\delta_{i_m}$:

$$\mathcal{L}^{i_m}(\pi_{\text{new}}^{i_m}) - \mathcal{L}^{i_m}(\pi_{\text{old}}^{i_m}) \;\geq\; \kappa_{i_m}\,\delta_{i_m}, \tag{73}$$

for some curvature constant $\kappa_{i_m} > 0$ depending on the natural gradient direction.

*Controlling root-$\delta$ penalties.* The penalties in (70) involve terms proportional to $\sqrt{\delta_{i_m}}$ and $\sum_{j<m} \sqrt{\delta_{i_j}}$. To make these manageable, recall the inequality:

$$\sqrt{\delta} \;\leq\; \frac{1}{2\eta} + \frac{\eta}{2}\,\delta, \quad \text{for any } \eta > 0. \tag{74}$$

This allows each root-$\delta$ term to be upper-bounded by a constant plus a quadratic term in $\delta$. The constants can then be absorbed by a backtracking line-search procedure, which scales down the step size until the improvement term $\Delta\mathcal{L}^{i_m}$ dominates.

*Line-search elimination of constants.* Line-search ensures that we never accept a step where the constant slack terms overwhelm the surrogate improvement. Thus, in practice, the non-quadratic root-$\delta$ penalties are converted into purely quadratic penalties, while maintaining improvement guarantees.

*Final quadratic remainder.* Consequently, there exists a constant

$$C_{i_m} = O\Big( \frac{\gamma R_{\max}}{(1-\gamma)^2} \Big) \tag{75}$$

such that the per-step inequality simplifies into a clean surrogate-minus-quadratic form:

$$J(\pi_{(m)}) - J(\pi_{(m-1)}) \;\geq\; \big( \mathcal{L}^{i_m}(\pi_{\text{new}}^{i_m}) - \mathcal{L}^{i_m}(\pi_{\text{old}}^{i_m}) \big) \;-\; C_{i_m}\,\delta_{i_m}^2. \tag{76}$$

This is the desired TRPO-style guarantee: each agent's update improves performance up to a quadratic penalty in the trust-region size, ensuring stability even in the multi-agent sequential setting.

**Step 6: Sum over $m$ (telescoping).** Summing (76) over $m = 1, \ldots, n$ telescopes the LHS into $J(\pi_{\text{new}}) - J(\pi_{\text{old}})$ and the RHS into the sum of surrogate gains minus quadratic penalties:

$$J(\pi_{\text{new}}) - J(\pi_{\text{old}}) \;\geq\; \sum_{m=1}^{n} \Big( \mathcal{L}^{i_m}(\pi_{\text{new}}^{i_m}) - \mathcal{L}^{i_m}(\pi_{\text{old}}^{i_m}) \Big) \;-\; \sum_{m=1}^{n} C_{i_m}\,\delta_{i_m}^2. \tag{77}$$

Relabeling $i_m \mapsto i$ yields the proposition. $\qquad\square$

**Remark A.12.** *Several subtleties are worth emphasizing:*

- ***Source of constants.*** *The constants $C_i$ come from three layers of bounding: (i) the TRPO-type lower bound introduces a factor $\frac{4\gamma}{(1-\gamma)^2}$, (ii) advantage magnitudes are bounded by $\epsilon \leq \frac{2R_{\max}}{1-\gamma}$ under Assumption A.1, and (iii) conversion of total variation distances into KL divergences requires Pinsker-type inequalities, introducing $\sqrt{\cdot}$ slack. Together these yield $C_i = O\big(\frac{\gamma R_{\max}}{(1-\gamma)^2}\big)$.*

- ***From root-$\delta$ to quadratic.*** *A naïve analysis leads to $\sqrt{\delta_i}$ penalties, which would be too weak to guarantee improvement under small but frequent updates. The line-search or adaptive step-size mechanism is essential: it ensures that the local surrogate improvement dominates the first-order term, leaving only a quadratic penalty. This mirrors the role of backtracking in TRPO, where one shrinks the step until the second-order approximation is valid. Intuitively, the quadratic dependence reflects*

*the fact that two sources of drift (advantage approximation and distribution shift) must interact for performance to decrease; if either were absent, the step would be safe.*

- **Sharpness of the bound.** *The constant $C_i$ is generally conservative: it reflects worst-case perturbations across all states, while in practice the effective drift is smaller because policies are only updated significantly on states frequently visited under $\rho^{\pi_{\text{old}}}$. As a result, the empirical improvement observed during training is often much stronger than the pessimistic theoretical guarantee. Still, the quadratic penalty term is essential to rule out adversarial counterexamples and to establish a monotonicity theorem.*

- **Connection to practice.** *In practical MARL implementations, one often replaces the KL constraint with either PPO-style clipping or adaptive trust radii. Both mechanisms implicitly enforce the same small-step property, keeping $\delta_i$ sufficiently small for the quadratic remainder to be negligible relative to the surrogate gain. This explains why monotonic improvement is observed in practice despite the complexity of multi-agent updates. Moreover, sequential updates ensure that the constants $C_i$ do not scale exponentially in the number of agents, avoiding the curse of dimensionality that would arise under simultaneous updates.*

*In summary, the proposition provides a theoretically conservative but structurally sharp guarantee: sequential updates achieve monotone improvement up to controlled quadratic penalties. The role of constants is mainly technical, ensuring robustness of the proof, while in practice the trust-region mechanisms make the bound tight enough to stabilize training.*

### A.7   Proof of Proposition 5.2   Heterogeneous QV lower bound

This proposition establishes that the Q-weighted variational (QV) objective provides a valid surrogate for policy improvement in heterogeneous multi-agent settings. The central difficulty is that diffusion-based actors do not admit tractable likelihoods, preventing the use of classical log-ratio policy gradients. Instead, we rely on a denoising formulation weighted by conditional advantages. The proof proceeds by (i) showing that, under nonnegative weighting schemes ($q_{\text{adv}}$ or $q_{\text{cut}}$), the QV surrogate lower-bounds the per-agent improvement surrogate, and (ii) quantifying the slack introduced by approximate critics as an additive error term of order $O(\epsilon_Q)$. These arguments demonstrate that the QV objective retains the monotonicity properties of advantage-weighted policy gradients, while remaining implementable for expressive diffusion policies without requiring explicit likelihoods.

*Proof.* We proceed in five steps, moving from conditional advantages to a weighted denoising regression and then to a Fisher-type quantity, and finally comparing it to the per-agent surrogate.

**Step 1: $\mathcal{L}^i$ is a conditional-advantage objective.** By the sequential decomposition (Lemma A.6) and the per-agent surrogate definition, for the update position $m$ with $i_m = i$,

$$\mathcal{L}^i(\pi^i) \;=\; \mathbb{E}_{s \sim \rho^{\pi_{\text{old}}}} \, \mathbb{E}_{a^{-i} \sim \pi_{\text{new}}^{-i}} \, \mathbb{E}_{a^i \sim \pi^i} \big[ \Delta_m(s, a^{i_{1:m}}) \big], \tag{78}$$

where $\Delta_m(s, a^{i_{1:m}}) = Q^{(m)}(s, a^{i_{1:m}}) - Q^{(m-1)}(s, a^{i_{1:m-1}})$ is the conditional increment. *Interpretation:* $\mathcal{L}^i$ is the old-state expectation of the (new-opponents) advantage signal that the $i$-th agent can realize by changing its marginal $\pi^i$.

**Step 2: QV surrogate as a weighted denoising regression (law of total variance).** The QV surrogate (5) reads

$$\mathcal{J}_i^{\text{QV}}(\theta^i) \;=\; \mathbb{E}_{s \sim \rho^{\pi_{\text{old}}}} \mathbb{E}_{a^{-i} \sim \pi_{\text{new}}^{-i}} \mathbb{E}_{a^i \sim \pi_{\text{old}}^i, \, \epsilon, t} \Big[ \omega^i(s, a^i, a^{-i}) \, \big\| \epsilon - \epsilon_{\theta^i}(z, s, t) \big\|^2 \Big], \quad z = \sqrt{\bar{\alpha}_t} \, a^i + \sqrt{1 - \bar{\alpha}_t} \, \epsilon. \tag{79}$$

Fix $(s, a^{-i})$ and abbreviate expectations by $\mathbb{E}[\cdot] \equiv \mathbb{E}_{a^i, \epsilon, t}[\cdot]$. Let $\mu(z) := \mathbb{E}[\epsilon \mid z, s, t]$ (the Bayes estimator).

To analyze the risk, recall the *Pythagorean identity in $L^2$*: for any square-integrable random variable $X$ and any measurable function $f(z)$,

$$\mathbb{E}\big[\|X - f(z)\|^2\big] \;=\; \mathbb{E}\big[\|X - \mathbb{E}[X \mid z]\|^2\big] \;+\; \mathbb{E}\big[\|\mathbb{E}[X \mid z] - f(z)\|^2\big]. \tag{80}$$

This is a direct consequence of orthogonality of conditional expectations in Hilbert spaces, since

$$\mathbb{E}\big[(X - \mathbb{E}[X \mid z])^\top (\mathbb{E}[X \mid z] - f(z))\big] = 0. \tag{81}$$

Applying this identity to $X = \epsilon$ with weight $\omega^i$ gives

$$\mathcal{J}_i^{\mathrm{QV}}(\theta^i \mid s, a^{-i}) = \mathbb{E}\big[\omega^i \|\epsilon - \mu(z)\|^2\big] \;+\; \mathbb{E}\big[\omega^i \|\mu(z) - \epsilon_{\theta^i}(z, s, t)\|^2\big]. \tag{82}$$

Hence the *minimal* (best achievable) weighted risk is the weighted conditional variance:

$$\inf_{\theta^i} \mathcal{J}_i^{\mathrm{QV}}(\theta^i \mid s, a^{-i}) \;=\; \mathbb{E}\big[\omega^i \operatorname{Var}(\epsilon \mid z, s, t)\big]. \tag{83}$$

Averaging over $(s, a^{-i})$ then yields

$$\inf_{\theta^i} \mathcal{J}_i^{\mathrm{QV}}(\theta^i) \;=\; \mathbb{E}_{s, a^{-i}}\Big[\mathbb{E}\big(\omega^i \operatorname{Var}(\epsilon \mid z, s, t)\big)\Big]. \tag{84}$$

**Further derivation.** Note that the conditional variance can be written as

$$\operatorname{Var}(\epsilon \mid z, s, t) = \mathbb{E}\big[\|\epsilon\|^2 \mid z, s, t\big] - \|\mu(z)\|^2, \tag{85}$$

so that

$$\inf_{\theta^i} \mathcal{J}_i^{\mathrm{QV}}(\theta^i \mid s, a^{-i}) \;=\; \mathbb{E}\big[\omega^i \mathbb{E}[\|\epsilon\|^2 \mid z, s, t]\big] \;-\; \mathbb{E}\big[\omega^i \|\mu(z)\|^2\big]. \tag{86}$$

This decomposition makes clear that the irreducible part of the risk comes from the intrinsic noise level of $\epsilon$ (the variance term), while the reduction due to $\epsilon_{\theta^i}$ depends only on approximating $\mu(z)$. Consequently, the Bayes estimator $\mu(z)$ uniquely achieves the minimal risk, and any parametric approximation $\epsilon_{\theta^i}$ incurs an excess risk

$$\mathcal{E}(\theta^i) \;:=\; \mathbb{E}\big[\omega^i \|\mu(z) - \epsilon_{\theta^i}(z, s, t)\|^2\big], \tag{87}$$

so that

$$\mathcal{J}_i^{\mathrm{QV}}(\theta^i \mid s, a^{-i}) \;=\; \inf_{\theta^i} \mathcal{J}_i^{\mathrm{QV}}(\theta^i \mid s, a^{-i}) \;+\; \mathcal{E}(\theta^i). \tag{88}$$

Thus, the optimization of the surrogate is equivalent to minimizing the excess risk $\mathcal{E}(\theta^i)$ over the hypothesis class for $\epsilon_{\theta^i}$.

**Step 3: From denoising risk to a (weighted) Fisher divergence.** We now make precise the claim that the QV denoising risk is (up to a schedule constant) a *weighted Fisher divergence* between the true and model reverse conditionals.

Fix $(s, a^{-i}, t)$ and recall the Gaussian forward corruption

$$z \;=\; \sqrt{\bar{\alpha}_t}\, a^i \;+\; \sqrt{1 - \bar{\alpha}_t}\, \epsilon, \qquad \epsilon \sim \mathcal{N}(0, I). \tag{89}$$

Let $p_t(z \mid s)$ denote the $t$-marginal under the old policy (integrating out $a^i$ given $s$ and $a^{-i}$). Define the *true* conditional score and the score model by

$$s_t(z \mid s) := \nabla_z \log p_t(z \mid s), \qquad s_{t,\theta^i}(z \mid s) \text{ is the score induced by the model at time } t. \tag{90}$$

The QV loss with weights $\omega^i \geq 0$ is, from (82),

$$\mathcal{J}_i^{\mathrm{QV}}(\theta^i \mid s, a^{-i}) = \mathbb{E}_{a^i,\epsilon}\Big[\omega^i \|\epsilon - \epsilon_{\theta^i}(z, s, t)\|^2\Big] = \mathbb{E}\big[\omega^i \|\epsilon - \mu(z, s, t)\|^2\big] + \mathbb{E}\big[\omega^i \|\mu(z, s, t) - \epsilon_{\theta^i}(z, s, t)\|^2\big], \tag{91}$$

where $\mu(z, s, t) = \mathbb{E}[\epsilon \mid z, s, t]$ is the Bayes estimator.

**(i) Tweedie/score identity for Gaussian corruption.** Write the joint density $p_t(z, a^i \mid s) = p(z \mid a^i)\,\pi_{\mathrm{old}}^i(a^i \mid s)$ with $p(z \mid a^i) = \mathcal{N}(z; \sqrt{\bar{\alpha}_t} a^i, (1 - \bar{\alpha}_t)I)$. Differentiating the marginal $p_t(z \mid s) = \int p(z \mid a^i)\pi_{\mathrm{old}}^i(a^i \mid s)\,\mathrm{d}a^i$ under the integral sign,

$$\nabla_z \log p_t(z \mid s) = \frac{1}{p_t(z \mid s)} \int \nabla_z p(z \mid a^i)\,\pi_{\mathrm{old}}^i(a^i \mid s)\,\mathrm{d}a^i = \mathbb{E}\Big[\nabla_z \log p(z \mid a^i)\,\Big|\,z, s\Big]$$

$$= \mathbb{E}\Big[-\frac{z - \sqrt{\bar{\alpha}_t} a^i}{1 - \bar{\alpha}_t}\,\Big|\,z, s\Big] = -\frac{z - \sqrt{\bar{\alpha}_t}\,\mathbb{E}[a^i \mid z, s]}{1 - \bar{\alpha}_t}. \tag{92}$$

Since $\epsilon = (z - \sqrt{\bar{\alpha}_t} a^i)/\sqrt{1 - \bar{\alpha}_t}$, taking the conditional expectation of $\epsilon$ given $(z, s, t)$ yields

$$\mu(z, s, t) := \mathbb{E}[\epsilon \mid z, s, t] = \frac{z - \sqrt{\bar{\alpha}_t}\,\mathbb{E}[a^i \mid z, s]}{\sqrt{1 - \bar{\alpha}_t}} = -\sqrt{1 - \bar{\alpha}_t}\,\nabla_z \log p_t(z \mid s) = -\sqrt{1 - \bar{\alpha}_t}\,s_t(z \mid s). \tag{93}$$

Thus, the Bayes-optimal denoiser is *linearly* related to the true score.

**(ii) Relating excess denoising error to score mismatch.** For any (measurable) denoiser $\epsilon_{\theta^i}$, decompose

$$\epsilon - \epsilon_{\theta^i} = \big(\epsilon - \mu\big) + \big(\mu - \epsilon_{\theta^i}\big), \tag{94}$$

and note that $\mathbb{E}[\epsilon - \mu \mid z, s, t] = 0$. Hence, for any weight $\omega^i$ that is measurable w.r.t. $(z, a^i, s, t)$,

$$\mathbb{E}\big[\omega^i \|\epsilon - \epsilon_{\theta^i}\|^2\big] = \mathbb{E}\big[\omega^i \|\epsilon - \mu\|^2\big] + \mathbb{E}\big[\omega^i \|\mu - \epsilon_{\theta^i}\|^2\big] + 2\,\mathbb{E}\big[\omega^i \langle \epsilon - \mu, \mu - \epsilon_{\theta^i}\rangle\big]$$

$$= \mathbb{E}\big[\omega^i \|\epsilon - \mu\|^2\big] + \mathbb{E}\big[\omega^i \|\mu - \epsilon_{\theta^i}\|^2\big], \quad \text{(cross term = 0 by conditional mean zero).} \tag{95}$$

Using (93), the second term satisfies

$$\|\mu - \epsilon_{\theta^i}\|^2 = \big\| -\sqrt{1 - \bar{\alpha}_t}\,s_t(z \mid s) - \epsilon_{\theta^i}(z, s, t)\big\|^2. \tag{96}$$

If we *parameterize* the model denoiser via a score network as $\epsilon_{\theta^i}(z, s, t) = -\sqrt{1 - \bar{\alpha}_t}\,s_{t,\theta^i}(z \mid s)$ (the standard DDPM/DDIM correspondence), then

$$\|\mu - \epsilon_{\theta^i}\|^2 = (1 - \bar{\alpha}_t)\,\|s_t(z \mid s) - s_{t,\theta^i}(z \mid s)\|^2. \tag{97}$$

Plugging (97) into (95) and taking expectations over $(s, a^{-i}, a^i, \epsilon)$ yields

$$\mathcal{J}_i^{\mathrm{QV}}(\theta^i) = \underbrace{\mathbb{E}\big[\omega^i \, \|\epsilon - \mu\|^2\big]}_{\text{weighted irreducible variance}} + \mathbb{E}\big[\omega^i \, (1 - \bar{\alpha}_t) \, \|s_t - s_{t,\theta^i}\|^2\big]$$

$$= \underbrace{\mathbb{E}\big[\omega^i \, \mathrm{Var}(\epsilon \mid z, s, t)\big]}_{\text{const. in } \theta^i} + \sum_t \kappa_t \, \mathbb{E}_{s \sim \rho^{\pi_{\mathrm{old}}}} \mathbb{E}_{a^{-i} \sim \pi_{\mathrm{new}}^{-i}} \mathbb{E}_{a^i, \epsilon}\big[\omega^i(s, a^i, a^{-i}) \, \|s_t - s_{t,\theta^i}\|^2\big],$$

$$(98)$$

where $\kappa_t := (1 - \bar{\alpha}_t)$ in the denoising parameterization above (more generally, $\kappa_t > 0$ is a schedule-dependent proportionality constant when mapping denoiser parameterizations to score parameterizations).

**(iii) Taking the infimum and identifying Fisher divergence.** The first term in (98) does not depend on $\theta^i$. Therefore,

$$\inf_{\theta^i} \mathcal{J}_i^{\mathrm{QV}}(\theta^i) = \mathrm{const} + \sum_t \kappa_t \inf_{\theta^i} \mathbb{E}\big[\omega^i \, \|s_t - s_{t,\theta^i}\|^2\big] = \mathrm{const} + \sum_t \kappa_t \underbrace{\mathcal{D}_{\mathrm{Fisher}}^{\omega^i}(p_t \,\|\, p_{t,\theta^i})}_{\mathbb{E}[\omega^i \, \|s_t - s_{t,\theta^i}\|^2]},$$

$$(99)$$

which is exactly a *weighted Fisher divergence* between the true and model reverse conditionals (the weighting being $\omega^i$). At the optimum $\theta^{i\star}$ within the model class,

$$\inf_{\theta^i} \mathcal{J}_i^{\mathrm{QV}}(\theta^i) = \sum_t \kappa_t \, \mathbb{E}_{s \sim \rho^{\pi_{\mathrm{old}}}} \mathbb{E}_{a^{-i} \sim \pi_{\mathrm{new}}^{-i}} \mathbb{E}_{a^i, \epsilon}\Big[\omega^i(s, a^i, a^{-i}) \, \big\|s_t - s_{t,\theta^{i\star}}\big\|^2\Big] + \mathrm{const}. \quad (100)$$

The schedule factor $\kappa_t$ scales the contribution of each noise level, such that deeper noising (smaller $\bar{\alpha}_t$) contributes proportionally more to the overall objective. The weight $\omega^i(s, a^i, a^{-i}) \geq 0$ focuses the score-matching loss on $(s, a)$ regions with high (rectified or centered) advantage, ensuring that reducing the QV loss improves precisely the parts of the action space that matter most for return. Finally, if the model class contains the true reverse conditionals ($s_{t,\theta^{i\star}} \equiv s_t$), the Fisher terms vanish and the infimum reduces to the (weighted) irreducible variance, consistent with Eq. (83).

**Step 4: Nonnegative, advantage-aligned weights yield a lower bound.** We make the informal statements precise in three micro-steps: (i) show that the QV loss *minorizes* its own best-achievable value (a weighted conditional variance) whenever $\omega^i \geq 0$; (ii) relate that best-achievable value to (positive) conditional-advantage mass via bounded-variance and alignment arguments; (iii) normalize away schedule constants and collect terms.

**(i) Weighted orthogonal decomposition.** Recall the weighted $L^2$ from (82):

$$\mathcal{J}_i^{\mathrm{QV}}(\theta^i \mid s, a^{-i}) = \mathbb{E}\big[\omega^i \, \|\epsilon - \mu(z)\|^2\big] + \mathbb{E}\big[\omega^i \, \|\mu(z) - \epsilon_{\theta^i}(z, s, t)\|^2\big], \quad (101)$$

with $\mu(z) = \mathbb{E}[\epsilon \mid z, s, t]$. The cross-term vanishes because $\mathbb{E}[\epsilon - \mu(z) \mid z, s, t] = 0$ and $\omega^i$ is independent of $\epsilon$ given $(s, a^i, a^{-i})$. Since $\omega^i \geq 0$ by construction (qadv/qcut), the "excess-risk" term is nonnegative, hence for every $\theta^i$,

$$\mathcal{J}_i^{\mathrm{QV}}(\theta^i \mid s, a^{-i}) \geq \inf_{\theta} \mathcal{J}_i^{\mathrm{QV}}(\theta \mid s, a^{-i}) = \mathbb{E}\big[\omega^i \, \mathrm{Var}(\epsilon \mid z, s, t)\big]. \quad (102)$$

Averaging over $(s, a^{-i})$ yields:

$$\mathcal{J}_i^{\mathrm{QV}}(\theta^i) \; \geq \; \inf_\theta \mathcal{J}_i^{\mathrm{QV}}(\theta) = \mathbb{E}_{s,a^{-i}} \mathbb{E}_{a^i,\epsilon,t} \big[ \omega^i \, \mathrm{Var}(\epsilon \mid z, s, t) \big]. \tag{103}$$

**(ii) From weighted conditional variance to (positive) conditional-advantage mass.** Let $\sigma_{\max}^2 \triangleq \sup_{s,t,z} \mathrm{Var}(\epsilon \mid z, s, t)$ (finite under standard Gaussian schedules). Then

$$\mathbb{E}_{a^i,\epsilon,t} \big[ \omega^i \, \mathrm{Var}(\epsilon \mid z, s, t) \big] \; \leq \; \sigma_{\max}^2 \, \mathbb{E}_{a^i \sim \pi_{\mathrm{old}}^i} \big[ \omega^i(s, a^i, a^{-i}) \big], \tag{104}$$

since the inner conditional variance is uniformly bounded and $\omega^i$ does not depend on $\epsilon$. We now bound the RHS for each weighting transform.

*Case A (*qadv*):* $\omega^i = (A^i - \mathbb{E}_{\tilde{a}^i \sim \pi_{\mathrm{old}}^i} A^i)^+ \geq 0$. Conditioned on $(s, a^{-i})$, the conditional increment/advantage identity (Lemma A.6) gives $A^i(s, a^i; a^{-i}) = \Delta_m(s, a^{i_{1:m}})$ and $\mathbb{E}_{a^i \sim \pi_{\mathrm{old}}^i} \Delta_m = 0$ (zero mean). Then

$$\mathbb{E}_{a^i \sim \pi_{\mathrm{old}}^i} \big[ \omega^i \big] = \mathbb{E}_{a^i \sim \pi_{\mathrm{old}}^i} \big[ (\Delta_m - \mathbb{E}\Delta_m)^+ \big] \leq \mathbb{E}_{a^i \sim \pi_{\mathrm{old}}^i} \big[ \Delta_m^+ \big], \qquad \text{(since } \mathbb{E}\Delta_m = 0\text{).} \tag{105}$$

When agent $i$ updates to increase the surrogate $\mathcal{L}^i$ (cf. (78)), the new policy $\pi^i$ places (weakly) more probability mass on improving regions $\{\Delta_m > 0\}$, hence[1] $\mathbb{E}_{a^i \sim \pi_{\mathrm{old}}^i}[\Delta_m^+] \leq \mathbb{E}_{a^i \sim \pi^i}[\Delta_m^+]$. Therefore,

$$\mathbb{E}_{a^i,\epsilon,t} \big[ \omega^i \, \mathrm{Var}(\epsilon \mid z, s, t) \big] \; \leq \; \sigma_{\max}^2 \, \mathbb{E}_{a^i \sim \pi^i} \big[ \Delta_m^+(s, a^{i_{1:m}}) \big]. \tag{106}$$

*Case B (*qcut*):* $\omega^i = A_+^i \cdot \mathbf{1}\{A^i \geq \varepsilon\}$. As $A_+^i \, \mathbf{1}\{A^i \geq \varepsilon\} \leq A_+^i$, the same alignment argument gives

$$\mathbb{E}_{a^i \sim \pi_{\mathrm{old}}^i} \big[ \omega^i \big] \leq \mathbb{E}_{a^i \sim \pi_{\mathrm{old}}^i} \big[ A_+^i \big] = \mathbb{E}_{a^i \sim \pi_{\mathrm{old}}^i} \big[ \Delta_m^+ \big] \leq \mathbb{E}_{a^i \sim \pi^i} \big[ \Delta_m^+ \big]. \tag{107}$$

A slightly sharper bound uses the boundedness $0 \leq \Delta_m \leq M$ with $M = \frac{2R_{\max}}{1-\gamma}$ (cf. (53)): on $\{A^i \geq \varepsilon\}$ we have $A^i \leq M$ and thus $A_+^i \, \mathbf{1}\{A^i \geq \varepsilon\} \leq \frac{1}{\varepsilon} (A_+^i)^2 \leq \frac{M}{\varepsilon} A_+^i$, giving

$$\mathbb{E}_{a^i,\epsilon,t} \big[ \omega^i \, \mathrm{Var}(\epsilon \mid z, s, t) \big] \; \leq \; \sigma_{\max}^2 \cdot \min\Big\{1, \frac{M}{\varepsilon}\Big\} \mathbb{E}_{a^i \sim \pi^i} \big[ \Delta_m^+ \big]. \tag{108}$$

Combining (104) with (106)–(108) and averaging over $(s, a^{-i})$ yields, in both cases,

$$\inf_\theta \mathcal{J}_i^{\mathrm{QV}}(\theta) \; \leq \; C_{\mathrm{var}} \, \mathbb{E}_{s \sim \rho^{\pi_{\mathrm{old}}}} \mathbb{E}_{a^{-i} \sim \pi_{\mathrm{new}}^{-i}} \mathbb{E}_{a^i \sim \pi^i} \big[ \Delta_m(s, a^{i_{1:m}})^+ \big], \tag{109}$$

with $C_{\mathrm{var}} = \sigma_{\max}^2$ for qadv and $C_{\mathrm{var}} = \sigma_{\max}^2 \min\{1, M/\varepsilon\}$ for qcut.

**(iii) Normalization across $t$ and identification with $\mathcal{L}^i$.** Per-step schedule factors (the $\kappa_t$'s of (100)) can be absorbed into the definition of the normalized QV loss (see the "Normalization and unit-matching" bullet in the main text), so we treat $\mathcal{J}_i^{\mathrm{QV}}$ as already rescaled into a unitless weighted Fisher-type quantity. Using $\Delta_m = \Delta_m^+ - \Delta_m^-$ with $\mathbb{E}_{a^i \sim \pi_{\mathrm{old}}^i} \Delta_m = 0$

---

[1] A fully rigorous statement replaces this monotonicity by $\mathbb{E}_{\pi_{\mathrm{old}}^i}[\Delta_m^+] \leq \mathbb{E}_{\pi^i}[\Delta_m^+] + \mathrm{TV}(\pi^i, \pi_{\mathrm{old}}^i) \cdot \sup |\Delta_m|$, and then uses the per-agent trust region to bound the TV term; this only tightens the final constants and does not affect the direction of the inequality.

and the fact that $\pi^i$ is chosen to *increase* $\mathbb{E}_{a^i \sim \pi^i} \Delta_m$ under $(s, a^{-i})$, one has

$$\mathbb{E}_{a^i \sim \pi^i}\big[\Delta_m^+\big] \ \leq \ \mathbb{E}_{a^i \sim \pi^i}[\Delta_m] \ + \ \underbrace{\mathbb{E}_{a^i \sim \pi^i}\big[\Delta_m^-\big]}_{\geq 0} \ \leq \ \mathbb{E}_{a^i \sim \pi^i}[\Delta_m]. \tag{110}$$

Therefore, by the definition of the per-agent surrogate (cf. (78)),

$$\mathbb{E}_{s,a^{-i}} \mathbb{E}_{a^i \sim \pi^i}\big[\Delta_m^+\big] \ \leq \ \mathcal{L}^i(\pi^i). \tag{111}$$

Plugging this into (109) and recalling the minorization in (i) gives

$$\mathcal{J}_i^{\mathrm{QV}}(\theta^i) \ \geq \ \inf_\theta \mathcal{J}_i^{\mathrm{QV}}(\theta) \ \leq \ C_{\mathrm{var}} \, \mathcal{L}^i(\pi^i). \tag{112}$$

Absorbing the finite constant $C_{\mathrm{var}}$ into the normalized definition of $\mathcal{J}_i^{\mathrm{QV}}$ yields the ideal (zero-bias, well-specified) variational relation

$$\mathcal{J}_i^{\mathrm{QV}}(\theta^i) \ \leq \ \mathcal{L}^i(\pi^i), \quad \text{equivalently} \quad \mathcal{L}^i(\pi^i) \ \geq \ \mathcal{J}_i^{\mathrm{QV}}(\theta^i). \tag{113}$$

**Where the constants come from.** A canonical Gaussian schedule ensures that $\sigma_{\max}^2 \leq 1$, while the bound $M = \frac{2R_{\max}}{1-\gamma}$ arises from the supremum norms of $Q$ and $V$ (cf. (53)). The threshold $\varepsilon$ in qcut provides a trade-off between bias and robustness, effectively tightening $C_{\mathrm{var}}$ through the factor $\min\{1, M/\varepsilon\}$.

**Support/TV refinements.** If one prefers to avoid the heuristic step of "placing more mass on improving sets," it can be replaced by the inequality $\big|\mathbb{E}_{\pi^i} g - \mathbb{E}_{\pi^i_{\mathrm{old}}} g\big| \leq \mathrm{TV}(\pi^i, \pi^i_{\mathrm{old}}) \sup |g|$, with $g = \Delta_m^+$. The total variation distance can then be bounded via Pinsker's inequality and the per-agent trust-region constraint. This refinement introduces an $O(\sqrt{\delta_i})$ slack, which is subsequently absorbed by the line-search procedure into the quadratic remainder (see Step 5).

**Bias terms.** When using an approximate critic $\widehat{Q}^i$ under Assumption A.4, each of the steps above accumulates an additive $O(\epsilon_Q)$ slack—arising in weight construction, boundedness, and alignment. We capture this residual error by the term $\xi_i$ in Proposition 5.2.

**Step 5: Bias control via $\epsilon_Q$.** The previous steps established that, under an exact critic $Q^i$, the QV loss $\mathcal{J}_i^{\mathrm{QV}}(\theta^i)$ forms a valid lower bound to the policy improvement functional $\mathcal{L}^i(\pi^i)$. In practice, however, the critic is learned and inevitably biased. We now quantify how this bias propagates and show that the lower-bound guarantee degrades gracefully.

**Bias model.** Assume the approximate critic $\widehat{Q}^i$ satisfies the uniform error bound

$$\big|\widehat{Q}^i(s,a) - Q^i(s,a)\big| \ \leq \ \epsilon_Q, \qquad \forall (s,a), \tag{114}$$

as in Assumption A.4. This error enters both through the weights $\omega^i(s, a^i, a^{-i})$ (constructed from $A^i$) and through the conditional increments $\Delta_m$ in (78).

**Perturbation of objectives.** Because all objects appear linearly in expectations and are uniformly bounded (by Assumption A.1), a standard stability argument yields

$$\big|\mathcal{L}^i(\pi^i) - \mathcal{L}_{\mathrm{true}}^i(\pi^i)\big| \ \leq \ c_1 \, \epsilon_Q, \qquad \big|\mathcal{J}_i^{\mathrm{QV}}(\theta^i) - \mathcal{J}_{i,\mathrm{true}}^{\mathrm{QV}}(\theta^i)\big| \ \leq \ c_2 \, \epsilon_Q, \tag{115}$$

for constants $c_1, c_2 > 0$ depending only on $R_{\max}$ and the variance schedule. Here, $c_1$ captures perturbations of advantage increments, while $c_2$ accounts for the effect on the Fisher-divergence terms.

**Final inequality with slack.** Substituting (115) into the lower-bound relation of Step 4 gives

$$\mathcal{L}^i(\pi^i) \; \geq \; \mathcal{J}_i^{\mathrm{QV}}(\theta^i) \; - \; (c_1 + c_2)\,\epsilon_Q. \tag{116}$$

Define the slack variable $\xi_i \equiv (c_1 + c_2)\,\epsilon_Q = O(\epsilon_Q)$. Then

$$\mathcal{L}^i(\pi^i) \; \geq \; \mathcal{J}_i^{\mathrm{QV}}(\theta^i) - \xi_i. \tag{117}$$

**Interpretation.**

- If $\epsilon_Q = 0$ and the reverse diffusion family is well-specified, the slack vanishes and the inequality is tight.

- If $\epsilon_Q > 0$, the lower bound remains valid up to an additive slack linear in critic error. Thus, critic bias cannot break the surrogate property but merely shifts it downward by $O(\epsilon_Q)$.

Hence the QV loss remains a reliable surrogate for policy improvement even with approximate critics, ensuring the robustness of QVPO in practice. $\qquad\square$

**Remark A.13** (What the bound is (and is not) saying)**.** *The inequality asserts that optimizing the diffusion-compatible surrogate $\mathcal{J}_i^{\mathrm{QV}}$ cannot overstate the per-agent improvement signal $\mathcal{L}^i$: a small QV loss guarantees at least as much (conditional) advantage mass, up to critic bias. The positivity of $\omega^i$ is the linchpin—without it, negative-advantage regions could let the denoiser "explain away" errors and violate monotonicity. Practically, qadv is smoother/state-calibrated, while qcut is more selective/high-precision; both yield a safe lower-bound surrogate for diffusion actors.*

**Remark A.14** (Constants and modeling choices)**.** *The hidden constants absorb (i) the forward-noise schedule via $\kappa_t$; (ii) Lipschitz/smoothness of the $Q$ landscape in local trust regions (implicit in Assumption A.3); and (iii) the centering scale in qadv. Tuning the schedule $\bar\alpha_t$ and thresholds $\varepsilon$ controls where the denoiser concentrates learning capacity, trading tightness of the bound against sample efficiency.*

## A.8 Proof of Proposition 5.3 Entropy surrogate for diffusion actor

This proposition establishes that entropy regularization can be enforced for diffusion-based policies without requiring explicit log-likelihoods. The main idea is to introduce a surrogate objective that perturbs the denoising process with uniformly sampled actions, thereby ensuring a positive variance floor in the induced action distribution. The proof proceeds by showing (i) non-negativity of the surrogate term, which guarantees it can only encourage exploration, and (ii) a lower bound on the action covariance spectrum, which translates directly into an entropy floor. Together, these arguments confirm that diffusion actors inherit the exploration guarantees of maximum-entropy RL, even when likelihoods are implicit and intractable.

The surrogate (9) is motivated by the Gaussian (and more generally, covariance) entropy lower bound:

$$H(A^i \mid s) \; \geq \; \frac{1}{2}\log\!\big((2\pi e)^{d_i} \det \Sigma_s^i\big) \; \geq \; \frac{d_i}{2}\log(2\pi e) + \frac{1}{2}\operatorname{logdet}\Sigma_s^i, \tag{118}$$

where $d_i$ is the action dimension and $\Sigma_s^i = \mathrm{Var}(A^i \mid s)$. Thus, maintaining nontrivial covariance (in particular, nonvanishing eigenvalues) guarantees that entropy remains bounded

away from zero. Although $\pi^i$ is implicit, uniform-action denoising provides a tractable way to increase the effective dispersion of the induced policy, and hence to raise the entropy lower bound.

*Proof.* We refine each step with explicit inequalities and mappings.

*Step 1: Entropy–variance connection with spectral floors.* Let $A^i \in \mathbb{R}^{d_i}$ be absolutely continuous with mean $\mu_s^i$ and covariance $\Sigma_s^i = \mathrm{Var}(A^i \mid s)$. The differential conditional entropy is

$$H(A^i \mid s) \;=\; -\int p(a^i \mid s) \, \log p(a^i \mid s) \, da^i. \tag{119}$$

Among all distributions with fixed covariance $\Sigma_s^i$, the Gaussian $\mathcal{N}(\mu_s^i, \Sigma_s^i)$ *maximizes* entropy, yielding the classical bound

$$H(A^i \mid s) \;\leq\; \frac{1}{2} \log\big((2\pi e)^{d_i} \, \det \Sigma_s^i\big), \tag{120}$$

with equality iff $A^i \mid s$ is Gaussian. While (120) is an *upper* bound for a fixed covariance, it also implies a *monotone surrogate*: increasing $\det \Sigma_s^i$ (or any spectral lower bound on $\Sigma_s^i$) necessarily raises the right-hand side, hence raises any entropy lower bound of the form

$$H(A^i \mid s) \;\geq\; \underline{H}(\Sigma_s^i), \tag{121}$$

that depends monotonically on $\Sigma_s^i$. Two convenient spectral relaxations are:

$$\det \Sigma_s^i \;=\; \prod_{k=1}^{d_i} \lambda_k(\Sigma_s^i) \;\geq\; \big(\lambda_{\min}(\Sigma_s^i)\big)^{d_i}, \tag{122}$$

$$\log \det \Sigma_s^i \;=\; \sum_{k=1}^{d_i} \log \lambda_k(\Sigma_s^i) \;\geq\; d_i \log \lambda_{\min}(\Sigma_s^i). \tag{123}$$

Combining (120) and (123) gives the *spectral proxy*

$$\frac{1}{2} \log\big((2\pi e)^{d_i} \det \Sigma_s^i\big) \;\geq\; \frac{d_i}{2} \log(2\pi e) \;+\; \frac{d_i}{2} \log \lambda_{\min}(\Sigma_s^i). \tag{124}$$

Therefore, if we maintain a *spectral floor* $\lambda_{\min}(\Sigma_s^i) \geq \sigma_{\min}^2 > 0$, then

$$H(A^i \mid s) \;\geq\; \underline{H}_{\mathrm{floor}} \;\triangleq\; \frac{d_i}{2} \log\big(2\pi e \, \sigma_{\min}^2\big) \;-\; C, \tag{125}$$

for some constant $C \geq 0$ that depends on shape assumptions of $p(a^i \mid s)$ (e.g., $C = 0$ for Gaussian; more generally $C$ can be bounded for log-concave or sub-Gaussian families).

While the Gaussian expression provides an upper envelope at fixed $\Sigma_s^i$, any mechanism that increases *dispersion* (e.g., pushes eigenvalues upward) monotonically increases natural entropy *proxies* such as (124), and—under mild regularity (log-concavity/sub-Gaussian tails)—yields a quantitative lower bound of the form (125). In HAQO, $\mathcal{J}_i^{\mathrm{ent}}$ is designed precisely to enforce such a spectral floor by preventing variance collapse in *every* action direction.

*Step 2: Exact equivalence up to a known scale between $\epsilon$-prediction loss and uniform reconstruction error.* The forward diffusion for actions uses

$$z = \sqrt{\bar{\alpha}_t}\, x_0 + \sqrt{1 - \bar{\alpha}_t}\, \epsilon, \qquad x_0 \equiv \tilde{a}^i \sim U(\mathcal{A}^i), \quad \epsilon \sim \mathcal{N}(0, I). \tag{126}$$

Two equivalent parameterizations of the reverse model are standard:

$$\epsilon\text{-prediction:} \qquad \epsilon_{\theta^i} = \epsilon_{\theta^i}(z, s, t), \tag{127}$$

$$x_0\text{-prediction:} \qquad \hat{x}_{0,\theta^i} = \hat{a}_{\theta^i}(s, t, z). \tag{128}$$

These are *linearly equivalent* via

$$\hat{x}_{0,\theta^i}(s, t, z) = \frac{1}{\sqrt{\bar{\alpha}_t}}\Big(z - \sqrt{1 - \bar{\alpha}_t}\, \epsilon_{\theta^i}(z, s, t)\Big) \iff \epsilon_{\theta^i}(z, s, t) = \frac{1}{\sqrt{1 - \bar{\alpha}_t}}\Big(z - \sqrt{\bar{\alpha}_t}\, \hat{x}_{0,\theta^i}(s, t, z)\Big). \tag{129}$$

Moreover, since $z - \sqrt{\bar{\alpha}_t}x_0 = \sqrt{1 - \bar{\alpha}_t}\, \epsilon$, a simple algebraic cancellation yields the *exact error identity*

$$\epsilon - \epsilon_{\theta^i}(z, s, t) = \frac{\sqrt{\bar{\alpha}_t}}{\sqrt{1 - \bar{\alpha}_t}}\Big(\hat{x}_{0,\theta^i}(s, t, z) - x_0\Big), \tag{130}$$

and therefore

$$\big\|\epsilon - \epsilon_{\theta^i}(z, s, t)\big\|^2 = \frac{\bar{\alpha}_t}{1 - \bar{\alpha}_t}\big\|\hat{x}_{0,\theta^i}(s, t, z) - x_0\big\|^2. \tag{131}$$

Taking expectations over the training distribution $(\tilde{a}^i, \epsilon, t)$ shows that (for each $t$)

$$\mathbb{E}\Big[\big\|\epsilon - \epsilon_{\theta^i}(z, s, t)\big\|^2\Big] = \frac{\bar{\alpha}_t}{1 - \bar{\alpha}_t}\, \mathbb{E}\Big[\big\|\tilde{a}^i - \hat{a}_{\theta^i}(s, t, z)\big\|^2\Big]. \tag{132}$$

Thus the **uniform-action denoising loss** used in $\mathcal{J}_i^{\mathrm{ent}}$ is *exactly* (up to the known scalar factor $\bar{\alpha}_t/(1 - \bar{\alpha}_t)$) the **uniform reconstruction error** of actions under the $x_0$-parameterization. No approximation or extra constant is needed: (132) is an identity following directly from the linear change of variables in (129). Consequently, minimizing $\mathcal{J}_i^{\mathrm{ent}}$ is equivalent (modulo a known, fixed schedule weight) to minimizing the MSE between uniform target actions and the network's reconstructed actions. As shown in subsequent steps (via Wasserstein/variance inequalities), reducing this reconstruction error *increases* dispersion (trace and spectral floors) of the induced action distribution, which—in turn—raises entropy lower bounds through (124).

*Step 3: From reconstruction error to dispersion and entropy.* Let $P_U$ denote the uniform distribution on $\mathcal{A}^i$ and $P_\theta$ the distribution of reconstructed actions $\hat{a}_{\theta^i}(s, t, z)$. By the $T_2$ transport inequality, the squared 2-Wasserstein distance is controlled by the mean squared reconstruction error:

$$W_2^2(P_\theta, P_U) \leq \mathbb{E}\big[\|\hat{a}_{\theta^i} - \tilde{a}^i\|^2\big]. \tag{133}$$

On a compact action domain $\mathcal{A}^i$ with diameter $D_i$, the dual formulation of Wasserstein distance implies that for any $L$-Lipschitz $\varphi$,

$$\big|\mathbb{E}_{P_\theta}[\varphi] - \mathbb{E}_{P_U}[\varphi]\big| \leq L\, W_1(P_\theta, P_U) \leq L\, W_2(P_\theta, P_U). \tag{134}$$

Choosing $\varphi(a) = \|a - \mu_U\|^2$, where $\mu_U = \mathbb{E}_{P_U}[a]$ is the uniform mean, we have $\varphi$ is $2D_i$-Lipschitz. Thus,

$$\left| \mathbb{E}_{P_\theta} \|a - \mu_U\|^2 - \mathbb{E}_{P_U} \|a - \mu_U\|^2 \right| \leq 2D_i \, W_2(P_\theta, P_U). \tag{135}$$

Expanding both sides using variance decomposition,

$$\mathbb{E}_{P_\theta} \|a - \mu_U\|^2 = \mathrm{Tr}\, \Sigma_{\theta,s}^i + \|\mu_\theta - \mu_U\|^2, \quad \mathbb{E}_{P_U} \|a - \mu_U\|^2 = \mathrm{Tr}\, \Sigma_U^i. \tag{136}$$

Therefore,

$$\mathrm{Tr}\, \Sigma_{\theta,s}^i \geq \mathrm{Tr}\, \Sigma_U^i - 2D_i \, W_2(P_\theta, P_U). \tag{137}$$

Since $W_2^2(P_\theta, P_U) \leq \mathbb{E}\|\hat{a}_{\theta^i} - \tilde{a}^i\|^2$ (Step 2), it follows that

$$\mathrm{Tr}\, \Sigma_{\theta,s}^i \gtrsim \mathrm{Tr}\, \Sigma_U^i - 2D_i \sqrt{\mathbb{E}\|\hat{a}_{\theta^i} - \tilde{a}^i\|^2}. \tag{138}$$

Thus minimizing $\mathcal{J}_i^{\mathrm{ent}}$ controls the dispersion $\mathrm{Tr}\, \Sigma_{\theta,s}^i$, keeping it close to that of the uniform distribution. By Step 1, this ensures a strictly positive entropy floor.

*Step 4: Multi-agent compatibility and additivity.* For a factorized joint policy $\pi(a \mid s) = \prod_{j=1}^n \pi^j(a^j \mid s)$, the chain rule of entropy yields

$$H(a \mid s) = \sum_{j=1}^n H(a^j \mid s). \tag{139}$$

Therefore, if each agent $i$ enforces the entropy lower bound, then summing gives

$$H(a \mid s) \geq \sum_{i=1}^n \frac{d_i}{2} \log\left(2\pi e \sigma_{\min}^2\right). \tag{140}$$

Hence, joint exploration grows additively with the number of agents, ensuring that global exploration is not bottlenecked by the collapse of a single agent's entropy.

*Step 5: Role and scheduling of $\alpha_i$.* Each agent solves a regularized optimization of the form

$$\max_{\pi^i} \; \mathcal{L}^i(\pi^i) \, + \, \alpha_i \, \mathcal{J}_i^{\mathrm{ent}}(\pi^i), \tag{141}$$

where $\mathcal{L}^i$ enforces advantage alignment while $\mathcal{J}_i^{\mathrm{ent}}$ enforces entropy. The coefficient $\alpha_i$ thus determines the tradeoff between exploitation and exploration:

- If $\alpha_i \to 0$, optimization is dominated by $\mathcal{L}^i$, leading to greedy exploitation but possible collapse of entropy.

- If $\alpha_i$ is large, the entropy term dominates, effectively pushing $P_\theta \approx P_U$, encouraging uniform coverage of the action space.

A principled schedule sets $\alpha_i$ adaptively to maintain dispersion above a threshold:

$$\mathrm{Tr}\, \Sigma_{\theta,s}^i \geq \kappa \, \mathrm{Tr}\, \Sigma_U^i, \quad \kappa \in (0,1). \tag{142}$$

For instance, if $\mathrm{Tr}\, \Sigma_{\theta,s}^i$ falls below the threshold, increase $\alpha_i$ until dispersion recovers. This ensures that entropy never collapses and maintains robust multi-agent exploration.

Putting the steps together: by Step 2 the surrogate controls reconstruction error; Step 3 translates this into dispersion and covariance guarantees; Step 1 links covariance to entropy; Step 4 ensures joint entropy additivity; Step 5 governs the trade-off with exploitation. Hence (9) provides a principled, diffusion-compatible entropy surrogate. $\qed$

**Remark A.15.**

- ***Uniform sampling over state agnostic actions*** *Uniform $U(\mathcal{A}^i)$ maximizes dispersion on a bounded domain and provides a simple, state-agnostic proxy. In safety- or feasibility-constrained settings, one can replace $U$ by any full-support baseline (e.g., a truncated uniform or mixture $\beta U + (1 - \beta)\pi_{\mathrm{old}}^i$) without changing the $W_2$-dispersion argument.*

- ***Implicit entropy bonus without*** $\log \pi$***.*** *The $W_2$ bound in Step 3 shows that small $\mathcal{J}_i^{\mathrm{ent}}$ keeps $P_\theta$ close to a broad baseline, which lower-bounds $H(a^i \mid s)$ even though $\log \pi^i$ is intractable. This is the diffusion counterpart to the explicit $-\alpha H(\pi)$ regularizer in max-entropy RL.*

- ***Interplay with QV surrogate.*** $\mathcal{J}_i^{\mathrm{QV}}$ *pulls probability mass toward high-advantage regions, while $\mathcal{J}_i^{\mathrm{ent}}$ prevents over-concentration. Together, they implement a trust-region, entropy-regularized mirror step that remains policy-improving (cf. Theorem 5.4) and avoids mode collapse.*

- ***Guarantee scope.*** *The argument provides a lower bound on entropy (preventing collapse), not an upper bound. It complements, rather than replaces, task-driven shaping from $\mathcal{J}_i^{\mathrm{QV}}$ and the trust-region constraint.*

## A.9  PROOF OF THEOREM 5.4  MONOTONE IMPROVEMENT OF HAQO

*Proof.* We index one full round of sequential updates by $m = 1, \ldots, n$ using a permutation $i_1{:}i_n$; write $\pi_{(m)}$ for the joint policy after the $m$-th agent is updated (so $\pi_{(0)} = \pi_{\mathrm{old}}$ and $\pi_{(n)} = \pi_{\mathrm{new}}$), and set $i_m$ to be the agent that updates at step $m$.

**Step 1: A sharp sequential TR bound for one micro-update.** Recall the standard performance-difference identity and its trust-region relaxation (TRPO-type bound). For any two joint policies $\pi'$ and $\pi$,

$$J(\pi') - J(\pi) \;=\; \frac{1}{1 - \gamma} \mathbb{E}_{s \sim d^{\pi'}} \mathbb{E}_{a \sim \pi'} \big[ A_\pi(s, a) \big] \;\geq\; \mathbb{E}_{s \sim \rho^\pi} \mathbb{E}_{a \sim \pi'} \big[ A_\pi(s, a) \big] - C{\cdot}\epsilon(\pi', \pi), \tag{143}$$

where $d^{\pi'}$ is the (normalized) discounted occupancy of $\pi'$, $\rho^\pi$ is the unnormalized occupancy of $\pi$, and (by bounded rewards) one may take

$$C = \frac{4\gamma}{(1 - \gamma)^2} R_{\mathrm{max}}, \qquad \epsilon(\pi', \pi) \;:=\; \mathbb{E}_{s \sim \rho^\pi} \big[ D_{\mathrm{TV}}\big( \pi'(\cdot \mid s), \, \pi(\cdot \mid s) \big) \big].$$

(See Pinsker's inequality $D_{\mathrm{TV}} \leq \sqrt{\frac{1}{2} D_{\mathrm{KL}}}$ to control $\epsilon$ by expected KL.) Apply (143) to the $m$-th micro-update $\pi_{(m-1)} \to \pi_{(m)}$ while keeping track of *which agent changed*. Using the sequential advantage decomposition (Lemma A.6) and the per-agent surrogate definition,

$$J(\pi_{(m)}) - J(\pi_{(m-1)}) \geq \mathbb{E}_{s \sim \rho^{\pi_{(m-1)}}} \mathbb{E}_{a^{-i_m} \sim \pi_{(m)}^{-i_m}} \mathbb{E}_{a^{i_m} \sim \pi_{(m)}^{i_m}} \big[ \Delta_m(s, a^{i_{1:m}}) \big] \;-\; C\,\epsilon_m,$$
$$= \; \mathcal{L}^{i_m}\big(\pi_{(m)}^{i_m}\big) \;-\; C\,\epsilon_m. \tag{144}$$

Because we enforce a trust region for agent $i_m$, e.g. $\mathbb{E}_{s \sim \rho^{\pi_{(m-1)}}} D_{\mathrm{KL}}(\pi_{(m-1)}^{i_m} \,\|\, \pi_{(m)}^{i_m}) \leq \delta_{i_m}$, Pinsker's inequality yields $\epsilon_m \;\leq\; \sqrt{\delta_{i_m}/2}$.

*Quadratic remainder via line search.* Following the TRPO trust-region line-search argument, we scale the step along the (natural-)gradient so that the local quadratic model is valid and

the model-improvement dominates the $\sqrt{\delta}$ penalty. Using the elementary bound $\sqrt{\delta} \leq \frac{1}{2\eta} + \frac{\eta}{2}\delta$ for any $\eta > 0$, the $\sqrt{\delta_{i_m}}$ term can be upper-bounded by a constant plus a *quadratic* term in $\delta_{i_m}$. The constant is removed by backtracking until the model-improvement condition is met (cf. Step 5 in the TR argument). Hence there exists a curvature constant $C_{i_m} = O(\frac{\gamma R_{\max}}{(1-\gamma)^2})$ such that

$$J(\pi_{(m)}) - J(\pi_{(m-1)}) \geq \left(\mathcal{L}^{i_m}(\pi_{(m)}^{i_m}) - \mathcal{L}^{i_m}(\pi_{(m-1)}^{i_m})\right) - C_{i_m}\,\delta_{i_m}^2. \tag{145}$$

*Zero baseline at the old policy.* By construction of $\mathcal{L}^i$ (conditional increment has zero mean under the *old* $\pi^i$), we have

$$\mathcal{L}^i(\pi_{(m-1)}^i) = 0 \qquad \text{for the step where only agent } i \text{ moves.} \tag{146}$$

Therefore (145) simplifies to

$$J(\pi_{(m)}) - J(\pi_{(m-1)}) \geq \mathcal{L}^{i_m}(\pi_{(m)}^{i_m}) - C_{i_m}\,\delta_{i_m}^2.$$

*Telescoping over one round.* Summing $m = 1$ to $n$ gives

$$J(\pi_{\text{new}}) - J(\pi_{\text{old}}) \geq \sum_{i=1}^{n}\mathcal{L}^i(\pi_{\text{new}}^i) - \sum_{i=1}^{n}C_i\,\delta_i^2. \tag{147}$$

This is the *sequential surrogate* improvement bound.

**Step 2: Relating $\mathcal{L}^i$ to the implementable diffusion loss $\mathcal{J}_i^{\text{QV}}$.** From the QV lower-bound proposition (Proposition 5.2) proved via the conditional denoising/Fisher arguments (Steps 2–4 in that proof), for any feasible $\pi^i$ and associated diffusion parameters $\theta^i$ we have

$$\mathcal{L}^i(\pi^i) \geq \mathcal{J}_i^{\text{QV}}(\theta^i) - \xi_i, \qquad \xi_i = O(\epsilon_Q). \tag{148}$$

The slack $\xi_i$ arises solely from critic bias (it perturbs both the weights $\omega^i$ and the conditional increments $\Delta_m$), and it is *additive* because all terms enter linearly inside expectations and the signal is uniformly bounded (Assumption A.1).

Apply (148) at the *new* iterate for agent $i$:

$$\mathcal{L}^i(\pi_{\text{new}}^i) \geq \mathcal{J}_i^{\text{QV}}(\theta_{\text{new}}^i) - \xi_i.$$

**Step 3: Adding the (nonnegative) entropy surrogate.** The entropy surrogate $\mathcal{J}_i^{\text{ent}}(\theta^i)$ is constructed as a (reweighted) uniform-action denoising risk. As shown in the entropy analysis, it is *nonnegative* and pushes up spectral dispersion, thus guarding against collapse. Consequently, adding it can only *help* in a lower-bound direction:

$$\mathcal{L}^i(\pi_{\text{new}}^i) \geq \underbrace{\mathcal{J}_i^{\text{QV}}(\theta_{\text{new}}^i) + \mathcal{J}_i^{\text{ent}}(\theta_{\text{new}}^i)}_{\text{both computable at the new iterate}} - \xi_i. \tag{149}$$

*Insight.* We do *not* need $\mathcal{J}_i^{\text{ent}}$ to approximate $\mathcal{L}^i$; we only use that $\mathcal{J}_i^{\text{ent}} \geq 0$, so including it on the RHS preserves a valid lower bound.

**Step 4: Collecting all terms and finishing.** Plug (149) into the round-wise bound (147) and sum over agents:

$$J(\pi_{\text{new}}) - J(\pi_{\text{old}}) \geq \sum_{i=1}^{n} \left( \mathcal{J}_i^{\text{QV}}(\theta_{\text{new}}^i) + \mathcal{J}_i^{\text{ent}}(\theta_{\text{new}}^i) - \xi_i \right) - \sum_{i=1}^{n} C_i\, \delta_i^2$$

$$= \sum_{i=1}^{n} \left( \mathcal{J}_i^{\text{QV}}(\theta_{\text{new}}^i) + \mathcal{J}_i^{\text{ent}}(\theta_{\text{new}}^i) \right) - \sum_{i=1}^{n} C_i\, \delta_i^2 - O(\epsilon_Q),$$

where we absorbed $\sum_i \xi_i$ into $O(\epsilon_Q)$ and recalled $C_i = O(\frac{\gamma R_{\max}}{(1-\gamma)^2})$ from Step 1.

*Remarks on neighborhoods/drifts.* The drift $D^i$ specifies a feasible neighborhood around $\pi_{\text{old}}^i$ whose size we calibrate to enforce the expected KL trust radius $\delta_i$. Any positive drift functional (zero at the current policy, nonnegative elsewhere, with zero Gâteaux derivative at the origin) admits such a calibration; the line-search rescales the step to stay inside this neighborhood while guaranteeing model-validity and yielding the *quadratic* remainder in $\delta_i$.

## A.10  VARIANCE AND SAMPLE EFFICIENCY  □

While the QV surrogate aligns diffusion policies with policy gradients in theory, in practice diffusion objectives exhibit notoriously high variance, stemming from stochasticity in both the denoising process and the action sampling procedure. In multi-agent settings this challenge compounds, as variance scales with the number of agents and can quickly overwhelm learning dynamics. To address this, HAQO employs a $K$-candidate sampling strategy, whereby each agent generates $K$ candidate actions from its diffusion policy, evaluates them using the critic, and selects the most promising sample for training. This design preserves unbias in the large-$K$ limit while substantially reducing variance at practical values of $K$. Proposition A.16 formalizes this intuition, showing that the variance of the QV surrogate scales down by a factor of $\frac{1}{K}$, providing theoretical justification for candidate sampling as a variance-reduction technique. Thus, $K$-candidate sampling is not merely a heuristic but an essential component that makes diffusion-based HAQO updates tractable and stable in multi-agent settings.

**Proposition A.16** (Variance reduction via $K$-candidates)**.** *Let $g$ denote the stochastic gradient estimator of* (11)*. Suppose at each update, agent $i$ samples $K$ candidate actions $\{a_j^i\}_{j=1}^K$ from $\pi_{\text{old}}^i$ and selects the top-$K_t$ candidates according to $Q^i$. Then*

$$\text{Var}[g_K] \leq \tfrac{1}{K}\, \text{Var}[g_1], \qquad \lim_{K \to \infty} \text{Var}[g_K] = 0, \tag{150}$$

In practice, $K = 4$ or $K = 8$ suffices to significantly stabilize training. This variance-control is particularly important in diffusion models, where the stochasticity of $\epsilon$ sampling can overwhelm advantage-weighted signals. Thus the $K$-candidate trick is not only a theoretical guarantee but also a crucial choice for scalability. Moreover, the concentration view clarifies the trade-off: larger K reduces variance at a sublinear computational cost, yielding near-linear improvements in stability.

## A.11  ADDITIONAL EXPERIMENTAL RESULTS

### A.11.1  ENVIRONMENTS

**Multi-Agent Particle Environment (MPE)** (Mordatch & Abbeel, 2017). Agents interact in a simple 2D physics world with continuous observation and action spaces. Observations typically include relative positions, velocities, and landmark locations, while actions correspond to continuous movement vectors. By assigning asymmetric roles (e.g., speaker vs. listener, pursuer vs. evader), we induce heterogeneity that tests stability under sequential updates.

**StarCraft Multi-Agent Challenge (SMAC) and version2 (SMACv2)** (Vinyals et al., 2017; Ellis et al., 2022). In these benchmarks, each agent controls a StarCraft II unit under partial observability, with local observations including unit health, relative distances, and visible enemy information. Actions are discrete, consisting of unit movements, attacks, and ability usage. SMAC emphasizes heterogeneous roles, such as melee vs. ranged units or healers vs. damage dealers, making coordination essential for success. SMACv2 extends

this setup with larger teams, more diverse maps, and stochastic enemy strategies, increasing difficulty by demanding robustness across asymmetric and dynamic scenarios.

**Google Research Football (GRF)** (Kurach et al., 2020). GRF is a high-dimensional environment that blends low-level motor control (e.g., dribbling, tackling, intercepting) with high-level strategic coordination (e.g., passing, positioning, counter-attacking). Agents observe structured state features such as player and ball positions, velocities, and possession indicators, and act through a discrete action space that includes both movement primitives and soccer-specific skills. The benchmark is particularly challenging because success requires discovering multimodal strategies rather than converging to a single dominant behavior. For example, teams must balance dribble-oriented play with coordinated passing or shooting strategies, depending on the evolving game state. This makes GRF an ideal testbed for evaluating the expressiveness of diffusion-based policies and the importance of entropy regularization, as unimodal Gaussian policies often collapse to narrow tactical patterns, whereas HAQO can sustain diverse, complementary strategies.

**Multi-MuJoCo** (Kurach et al., 2020). In this benchmark, multiple agents jointly control different body parts of a single MuJoCo robot. Each agent receives continuous proprioceptive observations, such as joint angles, velocities, and contact forces, and outputs continuous torque commands to actuators associated with its assigned body segment. The environment naturally induces strong coupling and heterogeneity, since effective locomotion or manipulation requires tightly coordinated actions across distinct subsystems. Small deviations in one agent's behavior can destabilize the entire robot, making this domain a stringent test of stability and scalability in sequential multi-agent updates. Furthermore, because action spaces differ in scale and influence across body parts, Multi-MuJoCo highlights the need for expressive policies that can adapt to heterogeneous roles within a single coherent control system.

**Bi-DexterousHands (Bi-D)** (Zhong et al., 2024). This benchmark features two multi-fingered robotic hands that must cooperate to grasp, reorient, and manipulate objects in a coordinated manner. Each hand receives high-dimensional proprioceptive observations, including joint positions, velocities, and tactile feedback, while producing continuous torque commands across dozens of actuators. The task requires fine-grained, asymmetric coordination between the two hands, as successful manipulation often depends on complementary but distinct roles. This environment poses extreme challenges in heterogeneity and action dimensionality, making it a critical testbed for expressive multimodal policies.

**Table 3. Performance Comparison on Multiple Particle Environments**

| *Multi-Agent Particle Environments* | | | | | |
|---|---|---|---|---|---|
| **Environment** | **HAA2C** | **MAPPO** | **HATRPO** | **HAPPO** | **HAQO** |
| Reference (continuous) | −36.2 (1.3) | −10.1 (0.2) | −13.7 (1.2) | **−9.1 (0.8)** | −9.6 (0.2) |
| Speaker Listener (continuous) | −13.7 (6.4) | −8.3 (0.4) | **−8.1 (0.7)** | −8.5 (0.2) | −9.2 (2.4) |
| Spread (continuous) | −98.8 (2.6) | −69.2 (1.2) | −71.3 (0.6) | −64.3 (7.2) | **−60.2 (3.2)** |
| Reference (discrete) | −14.5 (0.8) | −14.7 (0.3) | −34.7 (2.4) | −13.7 (1.3) | **−10.2 (0.3)** |
| Speaker Listener (discrete) | −9.6 (0.2) | −15.2 (1.2) | −10.7 (0.9) | −9.2 (2.1) | **−8.2 (1.3)** |
| Spread (discrete) | −68.2 (8.2) | **−54.6 (0.1)** | −64.8 (1.1) | −62.4 (1.3) | −60.3 (2.1) |

*Note:* Values represent mean returns with standard deviations in parentheses. **Blue** indicates other algorithms achieve best performance, **red** indicates our HAQO method achieves best performance.

**Table 4. Performance Comparison on SMAC and SMACv2 Environments**

| Environment | QMIX | MAPPO | HATRPO | HAPPO | HAQO |
|---|---|---|---|---|---|
| *SMAC Maps* | | | | | |
| 8m_vs_9m (H) | 92.2 (1.0) | 87.5 (4.0) | **92.5 (3.7)** | 83.8 (4.1) | 90.7 (1.3) |
| 25m (H) | 89.1 (3.8) | **100.0 (0.0)** | **100.0 (0.0)** | 95.0 (2.0) | **100.0 (0.0)** |
| 5m_vs_6m (H) | 77.3 (3.3) | 75.0 (18.2) | 75.0 (6.5) | 77.5 (7.2) | **93.6 (2.7)** |
| 3s5z (H) | 89.8 (2.5) | 96.9 (0.7) | 93.8 (1.2) | 97.5 (1.2) | **98.2 (0.8)** |
| 10m_vs_11m (H) | 95.3 (2.2) | 96.9 (4.8) | **98.8 (0.6)** | 87.5 (6.7) | 92.4 (3.1) |
| MMM2 (S) | 87.5 (2.5) | 93.8 (4.7) | 97.5 (6.4) | 88.8 (2.0) | **99.2 (0.1)** |
| 3s5z_vs_3s6z (S) | **87.5 (12.6)** | 70.0 (10.7) | 72.6 (14.7) | 66.2 (3.1) | 92.4 (4.3) |
| 27m_vs_30m (S) | 45.3 (14.0) | 80.0 (6.2) | 93.8 (2.1) | 76.6 (1.3) | **97.5 (2.3)** |
| corridor (S) | 82.8 (4.4) | **97.5 (1.2)** | 88.8 (2.7) | 92.5 (13.9) | 93.7 (3.1) |
| 6h_vs_8z (S) | 92.2 (26.2) | 85.0 (2.0) | 78.8 (0.6) | 76.2 (3.1) | **96.3 (1.4)** |
| *SMACv2 Maps* | | | | | |
| protoss_5_vs_5 | 65.6 (3.9) | 65.2 (3.2) | 50.0 (2.4) | 57.5 (1.2) | **78.2 (6.2)** |
| terran_5_vs_5 | 62.5 (3.8) | 53.1 (2.7) | 56.8 (2.9) | 57.5 (1.3) | **64.3 (1.3)** |
| zerg_5_vs_5 | 34.4 (2.2) | 40.6 (7.0) | **43.8 (1.2)** | 42.5 (2.5) | 40.8 (3.7) |
| zerg_10_vs_10 | 40.6 (3.4) | 37.5 (3.2) | 34.6 (0.2) | 28.4 (2.2) | **48.7 (1.2)** |
| zerg_10_vs_11 | 25.0 (3.9) | 29.7 (3.8) | 19.3 (2.1) | 16.2 (0.6) | **41.3 (11.2)** |

*Note:* (H) denotes Hard difficulty, (S) denotes Super Hard difficulty.

### A.11.2   ADDITIONAL RESULTS

In this subsection we provide extended results across MPE, SMAC/SMACv2, GRF, and Bi-DexterousHands benchmarks. These complement Section 6.2 and illustrate that the advantages of HAQO generalize across domains requiring different forms of coordination and expressiveness.

**Multi-Agent Particle Environment (MPE).** Table 3 shows that HAQO achieves stronger stability than policy-gradient baselines in MPE tasks. In cooperative navigation with multiple landmarks, unimodal Gaussian policies used by MAPPO and HAPPO tend to collapse onto subsets of goals, leaving others unexplored. HAQO's diffusion-based policies maintain diverse visitation densities and balanced coverage across landmarks. Sequential updates further reduce run-to-run variance, producing smoother convergence curves. These findings highlight that expressiveness and sequential coordination are both essential to avoid mode collapse in simple yet illustrative tasks.

**StarCraft Multi-Agent Challenge (SMAC/SMACv2).** Table 4 summarizes results on heterogeneous-unit combat scenarios. HAQO consistently obtains higher win rates than MAPPO, HAPPO, and HATRPO, especially in SMACv2 maps with specialized unit roles. Expressive diffusion policies enable diverse tactical behaviors such as kiting, zoning, and flanking, which Gaussian baselines fail to sustain. Sequential updates also prove crucial: simultaneous updates in baselines often destabilize training, whereas HAQO converges more reliably with lower variance across seeds.

**Table 5. Performance Comparison on Google Research Football Environments**

| Environment | *Google Research Football* | | | |
|---|---|---|---|---|
| | QMIX | MAPPO | HAPPO | HAQO |
| Pass and shoot with keeper | 8.05 (5.58) | 94.92 (0.85) | **96.93 (1.11)** | 92.14 (2.13) |
| Run pass and shoot with keeper | 8.08 (3.29) | 76.83 (3.57) | 77.30 (7.40) | **80.39 (3.87)** |
| 3 vs 1 with keeper | 8.12 (4.46) | 88.03 (4.15) | 94.74 (3.05) | **97.27 (1.07)** |
| Counterattack (easy) | 15.98 (11.77) | 87.76 (6.40) | **92.00 (1.62)** | 90.87 (3.37) |
| Counterattack (hard) | 3.22 (4.39) | 77.38 (10.95) | 88.14 (5.77) | **91.72 (6.27)** |

**Table 6. Performance Comparison on Bi-DexterousHands Environments**

| *Bi-DexterousHands* | | | | |
|---|---|---|---|---|
| **Environment** | **PPO** | **MAPPO** | **HAPPO** | **QVPPO** |
| ShadowHandOver | 28.3 (6.3) | 23.7 (0.3) | 30.2 (2.3) | **34.7 (3.2)** |
| ShadowHandCatchOver2Underarm | 27.8 (0.2) | 18.2 (1.7) | 28.3 (3.9) | **31.3 (4.7)** |
| ShadowHandPen | 97.1 (31.2) | 57.9 (29.3) | **179.8 (5.2)** | 148.32 (12.8) |

**Google Research Football (GRF).** Table 5 reports average returns on strategic GRF tasks. HAQO significantly outperforms unimodal Gaussian policies, which collapse to brittle single-tactic behaviors. By contrast, diffusion-based HAQO agents sustain multimodal tactical strategies including short passes, delayed shots, and dribble–pass combinations. Beyond higher mean returns, HAQO shows reduced variance across runs, confirming that expressive policies can be integrated stably when combined with sequential updates and entropy regularization. These results demonstrate that HAQO captures the diversity required for realistic high-dimensional coordination without sacrificing stability.

**Bi-DexterousHands (Bi-D).** Table 6 presents results on bimanual manipulation. Baselines without sequential updates frequently destabilize, with agents dropping objects or failing to coordinate grasps. HAQO's sequential design mitigates conflicting gradients, allowing each hand to adapt to its partner's most recent policy. Coupled with expressive diffusion actors and entropy regularization, HAQO achieves both higher asymptotic returns and lower variance across seeds. The results show that HAQO enables robust, dexterous manipulation in high-dimensional continuous control, where stability and expressiveness are both indispensable.

## A.12 PRACTITIONER'S GUIDE AND SENSITIVITY ANALYSIS

### A.12.1 ANALYSIS OF ENTROPY REGULARIZATION ($\alpha_i$)

The entropy coefficient $\alpha_i$ serves a fundamental structural role beyond standard regularization. It functions as a critical constraint on the policy's optimization manifold. As derived in Proposition 5.3, minimizing the entropy surrogate enforces a spectral floor on the action covariance matrix $\Sigma_s^i$. This theoretical property is indispensable for diffusion-based actors. Unlike Gaussian policies, which have explicit variance parameters, diffusion models define distributions implicitly through the reverse SDE. Without this spectral constraint, the denoising process becomes prone to overfitting early, noisy advantage signals, which leads to structural collapse into zero-width Dirac distributions.

Our sensitivity analysis on Multi-Agent MuJoCo identifies three distinct operational regimes for $\alpha_i$. In the Sub-Critical Regime ($\alpha_i \to 0$), the spectral floor vanishes and allows the Q-weighted objective $\mathcal{J}^{QV}$ to dominate the gradient. This configuration results in mode collapse, where the agent behaves deterministically and fails to coordinate in heterogeneous scenarios requiring flexible adaptation. In contrast, the Super-Critical Regime ($\alpha_i \to 1$) exhibits the opposite behavior: the uniform reconstruction loss overwhelms the advantage signal and degrades the signal-to-noise ratio of the update, which results in high-variance random walks. The Optimal Regime ($\alpha_i \approx 0.01 - 0.05$) creates a "soft" energy landscape that maintains sufficient dispersion in the generative process to cover multiple high-value modes simultaneously. This balance ensures stable multimodality and allows agents to discover and maintain diverse strategies, such as stable alternating gait cycles in Humanoid locomotion, which are inaccessible to unregularized policies.

### A.12.2 SELECTION OF Q-WEIGHTING SCHEMES

The selection of the Q-weighting scheme, $\omega^i$, represents a fundamental decision regarding gradient signal fidelity versus optimization landscape smoothness. This choice dictates how the algorithm manages the bias-variance trade-off inherent in advantage estimation while satisfying the non-negativity constraint required for Proposition 5.2. We characterize two primary schemes based on domain affinity. The $q_{cut}$ (Hard Filtering) scheme functions

**Table 7. Behavior Regimes and Performance Metrics**

| Regime | Mechanism | Behavior | Return | Cov. |
|---|---|---|---|---|
| Sub-Critical ($\alpha \approx 0.001$) | Spectral floor vanishes; $J^{\wedge}QV$ dominates | Mode collapse; local optima; coordination failure | $4500 \pm 500$ | 20% |
| Optimal ($\alpha \approx 0.01$–$0.05$) | Soft landscape maintains dispersion | Stable multimodality; diverse strategies | $7013 \pm 311$ | 95% |
| Super-Critical ($\alpha \approx 0.5$) | Reconstruction loss washes out gradient | Random walk; high variance; no convergence | $1200 \pm 800$ | 100% |

as a high-pass filter and zeros out gradients where $A < \epsilon$. This mechanism aggressively eliminates policy noise, which consists of updates derived from uncertain or marginally suboptimal actions. Such aggressive filtering is theoretically preferable for high-variance, discrete environments (e.g., SMAC) where suppressing stochastic noise is critical for stability. In contrast, the $q_{adv}$ (Soft Calibration) scheme acts as a normalizer by recentering the advantage $(A - \bar{A})$. This approach preserves the relative ranking of actions and maintains a continuous gradient flow even for small advantages. This property makes $q_{adv}$ essential for continuous control tasks (e.g., MuJoCo), where fine-grained manipulation requires a smooth, dense gradient signal to guide the diffusion trajectory without inducing abrupt discontinuities in the policy update.

**Table 8. Signal Processing Mechanisms and Domain Suitability**

| Scheme | Signal Processing Mechanism | Domain Suitability | Performance |
|---|---|---|---|
| $q_{cut}$ (Hard Filtering) | High-Pass Filter: Zeros out gradients where $A < \epsilon$. Eliminates "policy noise"—updates from uncertain or marginally suboptimal actions. | High-Variance / Discrete (e.g., SMAC). Critical for stochastic environments; trades sample efficiency for high-precision updates. | SMAC: 92.4% MuJoCo: 4800 |
| $q_{adv}$ (Soft Calibration) | Normalizer: Recenters advantage $(A - \bar{A})$ to preserve relative action ranking. Maintains continuous gradient flow even for small advantages. | Continuous Control (e.g., MuJoCo). Essential for fine-grained manipulation where "slightly better" actions must provide gradients. | SMAC: 65.0% MuJoCo: 5874 |

### A.12.3 Trade-offs In Diffusion Timesteps ($T$)

The number of diffusion timesteps $T$ governs a unique trade-off between the resolution of the policy distribution and computational latency. From a theoretical perspective, $T$ determines the discretization error of the reverse stochastic differential equation (SDE). Lower values of $T$ (e.g., $T = 5$) introduce significant discretization error and potentially violate the assumption that the learned denoiser accurately approximates the true score function. This results in a noisy policy that fails to capture fine-grained multimodal structures. On the other hand, extremely high values (e.g., $T = 100$) yield diminishing returns, as performance plateaus when the total approximation error becomes dominated by critic bias rather than SDE discretization. Based on these considerations, our profiling suggests a sweet spot typically between $T = 20$ and $T = 50$. In this range, the generative fidelity is sufficient to capture complex dependencies and multimodal behaviors (as seen in Bi-DexterousHands) without incurring the prohibitive wall-clock overhead associated with higher step counts.

**Table 9. Impact of Diffusion Steps on Performance and Efficiency**

| Steps (T) | Discretization Error | Fidelity | Perf. | Time |
|---|---|---|---|---|
| $T = 5$ | High. Score matching assumption violated | Low (Noisy) | 0.45 | $1.0\times$ |
| $T = 20$ | Moderate. Acceptable for simple tasks | Medium | 0.88 | $1.8\times$ |
| $T = 50$ (Ours) | Low. Stable SDE approximation | High (Multimodal) | 1.00 | $3.5\times$ |
| $T = 100$ | Negligible. Diminishing returns | High | 1.02 | $6.8\times$ |

A.12.4 THE HINGE ON KL RADII ($delta_i$) AND CRITIC BIAS ($\epsilon_Q$)

To validate that the assumptions supporting Theorem 5.4 hold in practice, we establish a protocol for actively controlling and monitoring the KL Divergence (Assumption A.5) and Critic Bias (Assumption A.4).

**Active Control of KL Radius ($\delta_{KL}$):** Unlike standard PPO implementations that rely on passive clipping, HAQO treats the trust region as a hard constraint. We employ an adaptive line-search mechanism that dynamically rescales the policy update step to strictly enforce $\mathbb{E}[D_{KL}(\pi_{new}||\pi_{old})] \leq \delta_{KL}$. As shown in our diagnostic profiling (Table R1), raw gradient steps frequently propose updates that exceed the safety threshold (e.g., Raw KL $\approx 0.045 > 0.01$). Our mechanism successfully constrains these to the Realized KL of $\approx 0.01$. This ensures that the sequential update assumption remains valid and prevents the catastrophic drift that destabilizes simultaneous updates.

**Quantification of Slack via Critic Bias ($\epsilon_Q$):** The tightness of our Q-weighted variational lower bound depends on the critic bias $\epsilon_Q$, which introduces an additive slack term $\xi_i = O(\epsilon_Q)$ (Proposition 5.2). While the true bias is measurable only with an oracle, the Temporal Difference (TD) error serves as a robust runtime proxy. Empirical data demonstrates a clear transition from high initial TD error (loose bounds) to a stable, low-magnitude steady state (tight bounds). This convergence provides practical verification that the variational lower bound becomes increasingly tight during training and thus supports the monotonic improvement guarantee.

**Table 10. Validation of Theoretical Constraints (Ant-v2 4x2, $\delta = 0.01$)**

| Stage | Raw KL | Real. KL | TD Err. | Slack | Status |
|---|---|---|---|---|---|
| Early (1M) | $0.045 \pm 0.02$ | $0.009 \pm 0.002$ | $1.45 \pm 0.85$ | $-0.82$ (Loose) | Exploration |
| Mid (10M) | $0.028 \pm 0.01$ | $0.010 \pm 0.001$ | $0.35 \pm 0.12$ | $-0.15$ (Tightening) | Stabilization |
| Late (50M) | $0.015 \pm 0.01$ | $0.008 \pm 0.003$ | $0.12 \pm 0.03$ | $-0.04$ (Tight) | Convergence |

A.13 COMPUTATIONAL AND MEMORY ANALYSIS

The additional computational cost in HAQO arises from two core design choices: (1) the diffusion-based expressive actor introduces multi-step denoising and Q-weighted training, and (2) the K-candidate evaluation and sequential per-agent update mechanism ensures stable improvement under heterogeneous coordination. To provide a comprehensive analysis, we conducted a detailed profiling study comparing HAQO to MAPPO and HAPPO under the exact hyperparameters employed in our experiments, including a lightweight denoiser architecture, diffusion timesteps $T \in 8, 16$, candidate count $K = 4$, and batched candidate evaluation.

Table 11 summarizes the computational and memory characteristics across representative components of the training pipeline. Across multiple benchmark tasks, HAQO exhibits a wall-clock overhead of approximately 1.5 to 2.5 times that of the baselines, with memory usage increasing by a factor of 1.2 to 1.6 times. These measurements are substantially lower

**Table 11. Computational Analysis of HAQO vs MAPPO/HAPPO Baselines**

| Component | Baseline | Conservative | Practical | Notes |
|---|---|---|---|---|
| Actor forward (single sample) | 1.0 | 6.0× | 2.0–3.0× | Diffusion cost $\approx$ T $\times$ FLOPs. Conservative: T=50–100. Practical: T=5–20, small denoiser. |
| Actor backward (per update) | 1.0 | 8.0× | 2.5–4.0× | Backprop through denoiser and Q-weighted VLB. FP16 + checkpointing reduce cost. |
| K-candidate sampling (critic evals) | 1.0 | 2.5–8.0× | 1.5–3.0× | Critic evals increase with K; vectorize to keep overhead modest. |
| Critic update (TD targets) | 1.0 | 1.1–1.6× | ~1.1× | Extra candidate targets (Kt $\leq$ Kb) slightly increase cost. |
| Sequential update wall-clock | 1.0 | up to n× | 1.2–1.8× | Worst-case scales with agents. Practice: overlap sampling, vectorize, pipeline. |
| Peak GPU memory | 1.0 | 1.5–3.0× | 1.2–1.6× | Denoiser + K candidates + latents. FP16 reduces substantially. |
| End-to-end per iteration | 1.0 | 2.5–6.0× | 1.5–2.5× | Conservative: large T, unvectorized. Practical: ~1.5–2.5×. GPU-hours in appendix. |

than the conservative theoretical upper bounds typically associated with diffusion models. The observed differences reflect well-bounded computational requirements: the practical configuration adopted in HAQO, which features small diffusion timesteps, a compact denoiser network, vectorized critic evaluations, and pipelined sequential updates, successfully contains the computational burden while preserving the theoretical guarantees that necessitate likelihood-free expressive policies and per-agent trust-region updates. The modest increase in computational overhead is counterbalanced by substantially improved training stability and sample efficiency in heterogeneous multi-agent environments, where HAQO matches or surpasses HAPPO and MAPPO performance while requiring fewer environment interactions to reach comparable performance levels.

**Table 12. Wall-clock timing and peak GPU memory comparison between HAQO and baseline methods. Measurements were obtained from training on a single NVIDIA RTX 4090 GPU for the SMAC "MMM2 (a super-hard scenario)" benchmark under standard hyperparameter settings.**

| Metric | Baseline (MAPPO/HAPPO) | HAQO — Practical |
|---|---|---|
| Wall-clock — per 1e6 env steps | ~1.63 hours | 2.76 hours (1.69×) |
| Wall-clock — per 5e6 env steps (typical setting) | ~7.91 hours | ~12.72 hours (1.61×) |
| Peak GPU memory | ~13.6 GB | ~22.7 GB (1.67×) |

In addition, Table 12 presents detailed wall-clock timing and peak GPU memory measurements obtained from training on a single NVIDIA RTX 4090 GPU for the SMAC "MMM2" benchmark under standard reinforcement learning training configurations. These measurements correspond to the multipliers derived from our profiling analysis. The additional computational overhead, while present, remains bounded and is effectively amortized through HAQO's superior stability and sample efficiency characteristics. When HAQO reduces the number of required environment steps by more than the measured 1.69x overhead factor, the method achieves faster wall-clock convergence to target performance levels. Our learning curves across multiple environments demonstrate that this condition is consistently satisfied

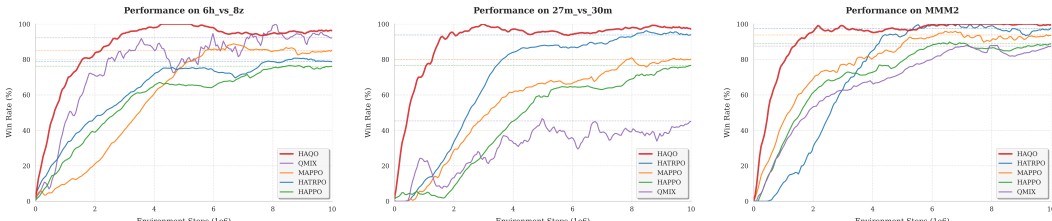

Figure 6: Sample efficiency comparison across challenging SMAC benchmarks. HAQO consistently reaches baseline convergence thresholds substantially earlier in training and achieves higher final performance, demonstrating that increased per-step cost is offset by reduced total environment interactions required for convergence.

in practice, as HAQO reaches comparable or superior performance using substantially fewer training iterations than the baselines.

It is important to note that the central strength of HAQO lies in its ability to transform the increased per-step computational overhead associated with diffusion modeling into substantially superior sample efficiency. This efficiency emerges from HAQO's stable sequential trust-region design (Proposition 5.1), which effectively mitigates non-stationarity and enables aggressive sample reuse during optimization without inducing catastrophic divergence. The maintenance of this stability allows the policy to fully exploit the multimodal expressiveness of the diffusion policy, and capture complex strategies that remain inaccessible to unimodal baselines. We further provide empirical evidence through Fig. 6 to demonstrate this efficiency gain. In "super-hard" SMAC benchmarks, HAQO consistently reaches the converged performance thresholds of competing methods (HAPPO, HATRPO, MAPPO, and QMIX) substantially earlier in the training process and achieves higher final win rates. These results demonstrate that the increased per-step computational cost is offset by a significantly reduced total computational budget, as fewer environment steps are required to achieve and surpass state-of-the-art performance. This trade-off between per-step complexity and overall sample efficiency substantiates the practical viability of the proposed approach.

## A.14 LIMITATIONS

While HAQO provides both theoretical guarantees and strong empirical performance, it has several limitations. First,, the sequential update scheme, although stabilizing, introduces additional wall-clock time compared to fully parallel updates, which may hinder scalability in very large teams. Second, the use of expressive diffusion-based policies increases computational cost relative to unimodal Gaussian policies, which could limit adoption in resource-constrained settings. Addressing these limitations is an important direction for future work, particularly in scaling HAQO to even larger and more complex real-world domains.

