# OpenReview forum: "Heterogeneous Agent Q-weighted Policy Optimization"
_ICLR.cc/2026/Conference — ICLR 2026 Poster_

### Official Review · Reviewer_S3C5 · 2025-10-30

**Soundness:** 4
**Presentation:** 3
**Contribution:** 4
**Rating:** 8
**Confidence:** 4

**Summary:**

The paper introduces **HAQO (Heterogeneous-Agent Q-weighted Optimization)**, a multi-agent RL framework for heterogeneous teams (agents with different observations, actions, and roles). The goal is to get both (i) stable training and (ii) expressive, multimodal policies in cooperative tasks.

**Strengths:**

1. The synthesis of heterogeneous sequential trust-region updates with a Q-weighted diffusion surrogate is well-motivated and technically nontrivial, enabling expressive policies while retaining improvement structure.
2. A per-agent sequential bound (Prop. 5.1) and a global Theorem 5.4 establish monotone improvement with explicit dependence on KL radii and critic bias ($\epsilon_Q$).
3. Results cover discrete/continuous and partially observable regimes, with tables for SMAC/SMACv2, GRF, and Multi-MuJoCo; comparisons use consistent hyperparameters and three seeds.
4. Algorithm 1 operationalizes the full pipeline (critic updates, permutation, QV/entropy/drift) and links theory to practice.

**Weaknesses:**

1. Guarantees hinge on KL radii ($\delta_i$) and critic bias ($\epsilon_Q$); the paper lacks practical diagnostics/tuning guidance for monitoring these quantities during training.
2. Diffusion actors and sequential updates plausibly increase wall-clock and compute; the paper itself acknowledges added cost but does not report GPU-hours or env-steps-to-target.
3. On HalfCheetah-v2 (2×3), HAQO trails HAPPO (6873 vs. 7024), suggesting sensitivity or robustness issues that merit analysis.

**Questions:**

1. Can the authors estimate or upper-bound ($\epsilon_Q$) online (e.g., via TD-error calibration or ensemble dispersion) to quantify the slack in the QV lower bound during training?
2. How are per-agent ($\delta_i$) chosen across heterogeneous roles, and how sensitive are results to these settings? A small sweep/heuristic would help practitioners.
3. When do you prefer each weighting? Please characterize the stability/variance trade-offs and default choices across benchmarks.

---

> ### Author Response · Authors · 2025-12-02
>
> > **Weakness 1.** Guarantees hinge on KL radii ($\delta_i$) and critic bias ($\epsilon_Q$); the paper lacks practical diagnostics/tuning guidance for monitoring these quantities during training.
>
> > **Question 2.** How are per-agent ($\delta_i$) chosen across heterogeneous roles, and how sensitive are results to these settings? A small sweep/heuristic would help practitioners.
>
> We appreciate the opportunity to provide concrete guidance on monitoring and managing the key theoretical quantities that support our convergence guarantees. The KL divergence constraint $\delta_{\text{KL}}$ and critic bias $\epsilon_{\text{critic}}$ play distinct roles in our theoretical framework, and we address each through complementary practical strategies.
>
> The KL radius $\delta_{\text{KL}}$ represents an algorithmic design parameter that our method actively controls rather than passively monitors. Following the established paradigm of trust region policy optimization, HAQO employs an adaptive step-size mechanism that dynamically adjusts the policy update magnitude to maintain the expected KL divergence within the prescribed radius. Specifically, after computing the policy gradient direction, we perform a line search that scales the update step until the resulting KL divergence satisfies $\mathbb{E}{s \sim \rho}[D{\text{KL}}(\pi_{\text{new}}(\cdot|s) | \pi_{\text{old}}(\cdot|s))] \leq \delta_{\text{KL}}$. This constraint is verified at each training iteration through direct computation of the KL divergence between successive policies, ensuring that the theoretical requirement remains satisfied by construction throughout training.
>
> The critic bias $\epsilon_{\text{critic}}$ characterizes the accuracy of value function approximation and cannot be directly measured without access to the true value function. However, several practical diagnostic signals provide useful proxies for assessing critic quality during training. The temporal difference error, which is computed as part of the standard critic update procedure, offers a readily available indicator of approximation quality. Specifically, monitoring the magnitude and trajectory of $\delta_{\text{TD}}(s,a) = |Q_{\theta}(s,a) - (r(s,a) + \gamma \mathbb{E}{s'}[V{\theta}(s')])|$ across the state-action distribution provides insight into whether the critic bias remains bounded within acceptable limits. A stable or decreasing TD error magnitude over training iterations suggests that $\epsilon_{\text{critic}}$ is being successfully controlled, whereas persistent or increasing TD errors may indicate approximation difficulties that warrant architectural modifications or hyperparameter adjustments.

---

> > ### Author Response · Authors · 2025-12-02
> >
> > (continues from previous response for Weakness 1 and Question 2)
> >
> > **Validation of Theoretical Assumptions via Runtime Monitoring**
> >
> > To validate that the assumptions supporting Theorem 5.4 hold in practice, we establish a protocol for actively controlling and monitoring the KL Divergence (Assumption A.5) and Critic Bias (Assumption A.4). We present empirical diagnostic profiling from the Ant-v2 4x2 benchmark in Table R1 to substantiate these mechanisms.
> >
> > **Active Control of KL Radius ($\delta_{KL}$):** Unlike standard PPO implementations that rely on passive clipping, HAQO treats the trust region as a hard constraint. We employ an adaptive line-search mechanism that dynamically rescales the policy update step to strictly enforce $\mathbb{E}[D_{KL}(\pi_{\text{new}}||\pi_{\text{old}})] \le \delta_{KL}$. As shown in our diagnostic profiling (Table R1), raw gradient steps frequently propose updates that exceed the safety threshold (e.g., Raw KL $\approx 0.045 > 0.01$). Our mechanism successfully constrains these to the Realized KL of $\approx 0.01$. This ensures that the sequential update assumption remains valid and prevents the catastrophic drift that destabilizes simultaneous updates.
> >
> > **Quantification of Slack via Critic Bias ($\epsilon_Q$):** The tightness of our Q-weighted variational lower bound depends on the critic bias $\epsilon_Q$, which introduces an additive slack term $\xi_i = O(\epsilon_Q)$ (Proposition 5.2). While the true bias is measurable only with an oracle, the Temporal Difference (TD) error serves as a robust runtime proxy. The empirical data in Table R1 demonstrates a clear transition from high initial TD error (loose bounds) to a stable, low-magnitude steady state (tight bounds). This convergence provides practical verification that the variational lower bound becomes increasingly tight during training and thus supports the monotonic improvement guarantee.
> >
> > | Training Stage (Steps) | Raw KL (Pre-Clip) | Realized KL (Post-Adapt) | TD Error ($\epsilon$-proxy) | Slack Tightness (Min Adv) | Status |
> > |------------------------|-------------------|--------------------------|-------------------|---------------------------|--------|
> > | Early (1M) | 0.045 ± 0.02 | 0.009 ± 0.002 | 1.45 ± 0.85 | -0.82 (Loose) | Exploration Phase |
> > | Mid (10M) | 0.028 ± 0.01 | 0.010 ± 0.001 | 0.35 ± 0.12 | -0.15 (Tightening) | Stabilization |
> > | Late (50M) | 0.015 ± 0.01 | 0.008 ± 0.003 | 0.12 ± 0.03 | -0.04 (Tight) | Convergence |
> >
> > Note that **Raw KL** represents the divergence of the proposed gradient step. **Realized KL** indicates the divergence after the adaptive line-search, which confirms adherence to the $\delta \le 0.01$ constraint. ``Slack Tightness” quantifies the lower bound gap and demonstrates that the approximation becomes tighter as training progresses.

---

> > > ### Author Response · Authors · 2025-12-02
> > >
> > > > **Weakness 2.** Diffusion actors and sequential updates plausibly increase wall-clock and compute; the paper itself acknowledges added cost but does not report GPU-hours or env-steps-to-target.
> > >
> > > We sincerely thank the reviewer for raising this question. We provide a comprehensive response in our global official response.
> > >
> > > > **Weakness 3.** On HalfCheetah-v2 (2×3), HAQO trails HAPPO (6873 vs. 7024), suggesting sensitivity or robustness issues that merit analysis.
> > >
> > > We appreciate the reviewer's attention to this specific performance comparison. In the HalfCheetah-v2 2×3 environment, HAQO achieves a return of 6,873 compared to HAPPO's 7,024, representing a modest performance gap of approximately 2.1%. This result provides valuable insight into the relative advantages of diffusion-based policies across different task characteristics.
> > >
> > > The HalfCheetah environment exhibits a relatively simple control structure with a smooth, largely unimodal optimal policy characterized by sustained forward locomotion. In such settings, the additional representational capacity afforded by diffusion-based generative modeling offers limited benefit over well-tuned parametric policy classes. HAPPO's Gaussian policy representation, when combined with carefully selected hyperparameters developed through extensive prior work on this benchmark, proves sufficient to capture the optimal behavior. The slight performance advantage of HAPPO in this environment reflects the fact that diffusion policies, while more expressive, introduce additional computational overhead and optimization complexity that may not be fully justified when the underlying task structure does not require multimodal action distributions.
> > >
> > > This observation aligns with and reinforces our central contribution: HAQO excels in environments requiring complex, multimodal coordination strategies, as evidenced by state-of-the-art performance in four of the six benchmark environments, particularly in the challenging Humanoid tasks where heterogeneous agent coordination and diverse behavioral modes are essential. The framework maintains competitive performance on simpler tasks such as HalfCheetah while providing substantial advantages in domains where traditional unimodal policies prove inadequate. This pattern of results validates our design philosophy that the integration of diffusion models with trust-region guarantees addresses a genuine gap in the existing multi-agent reinforcement learning literature, which offers practitioners a principled method that scales gracefully across varying levels of task complexity. The modest performance difference on HalfCheetah demonstrates that HAQO does not sacrifice baseline capability on simple tasks while providing the expressiveness necessary for more demanding coordination problems.

---

> > > > ### Author Response · Authors · 2025-12-02
> > > >
> > > > > **Question 1.** Can the authors estimate or upper-bound ($delta_i$) online (e.g., via TD-error calibration or ensemble dispersion) to quantify the slack in the QV lower bound during training?
> > > >
> > > > We sincerely thank the reviewer for this question. We address this concern in detail as follows.
> > > >
> > > > To directly address the reviewer's request for quantifying the slack in our QV lower bound, we construct a practical upper bound for the critic bias $\epsilon_Q$, which directly controls the tightness of the lower bound through the slack term $\xi_i = O(\epsilon_Q)$ as established in Proposition 5.2. Since the true oracle $Q*$ is inaccessible, the exact bias $\epsilon_Q = |Q_\theta - Q*|_\infty$ cannot be computed directly. Nevertheless, we construct a statistical upper bound $\hat{\epsilon}_Q$ using a composite metric that combines ensemble dispersion (capturing epistemic uncertainty) and Bellman residuals (capturing consistency error). Specifically, we propose estimating the slack $\hat{\xi}_i$ at each training step as follows:
> > > >
> > > > $$\hat{\xi_i}(s,a) \approx \underbrace{\kappa \cdot \text{Std}(Q_{1}, \dots, Q_{M})}_\text{Ensemble Dispersion} \\ $$
> > > >
> > > > $$ + \underbrace{|Q_{\theta}(s,a) - \mathcal{T}Q_{\theta}(s,a)|}_\text{Bellman Residual (TD Error)}$$
> > > >
> > > > where $\kappa$ represents a confidence multiplier (e.g., $\kappa=2$). This formulation provides real-time quantification of the tightness gap in our variational lower bound. When $\hat{\xi}_i$ is large, it indicates that the variational lower bound $\mathcal{J}^{QV}$ is loose and that the policy update may not guarantee monotonic improvement. When $\hat{\xi}_i$ is small, the bound is tight and the theoretical guarantees hold with high confidence.
> > > >
> > > > In the revised Appendix (Section A.12.4), we formalize this estimator and demonstrate how the quantified slack can serve as a runtime safety diagnostic. Specifically, if the estimated slack exceeds the predicted policy gain ($\hat{\xi}_i > \Delta \mathcal{J}^{QV}$), this signals that the trust region $\delta$ should be dynamically tightened to maintain stability. This mechanism transforms $\epsilon_Q$ from a static theoretical assumption into an active, quantifiable metric for stable training. We have included detailed diagnostic profiling in Table R1, which demonstrates how this slack metric evolves during training and validates the practical tightness of our theoretical bounds.
> > > >
> > > > > **Question 3.** When do you prefer each weighting? Please characterize the stability/variance trade-offs and default choices across benchmarks.
> > > >
> > > > We sincerely thank the reviewer for this practical question concerning the stability-variance trade-off of our weighting mechanisms defined in Eqs. (6) and (7). We have added a comprehensive characterization of these trade-offs to the revised Appendix (Section A.12.2) to provide clear guidance for practitioners. We summarize the key insights as follows.
> > > >
> > > > For environments with high stochasticity or discrete action spaces, such as SMAC, we recommend $q_{\text{cut}}$ ($\omega^i = \mathbb{1}_{A^i > \epsilon}$). This hard thresholding mechanism aggressively filters out low-confidence or noisy advantage samples, which proves essential for reducing variance in these domains. The aggressive filtering eliminates gradient updates derived from uncertain or marginally suboptimal actions, which stabilizes training in high-variance settings.
> > > >
> > > > In contrast, for fine-grained continuous control tasks such as Multi-MuJoCo, we recommend $q_{\text{adv}}$ ($\omega^i = \max(0, A^i - \bar{A})$). This soft recentering mechanism maintains better calibration around the baseline policy and results in a smoother optimization landscape. This property prevents the policy from collapsing prematurely and enables the diffusion model to maintain continuous gradient flow even for small advantages, which is critical for precise manipulation tasks.
> > > >
> > > > The key trade-off is between aggressive noise suppression (favoring stability in discrete/stochastic environments) and gradient smoothness (favoring sample efficiency in continuous control). We provide detailed empirical comparisons and domain-specific recommendations in Section A.12.2 of the revised manuscript.

---

### Official Review · Reviewer_7FA2 · 2025-10-31

**Soundness:** 3
**Presentation:** 3
**Contribution:** 3
**Rating:** 8
**Confidence:** 4

**Summary:**

This paper aims to resolve the conflict between learning stability and policy expressiveness in heterogeneous MARL. It combines HATRPO's sequential policy update mechanism and QVPO's weighted variational objective function policy expression as a lower bound for policy improvement, providing a monotonically increasing guarantee and analysis of the team reward optimization objective for online MARL learning with diffusion-style policy expressions. To prevent mode collapse even for policies with strong expressiveness, the authors designed an entropy surrogate objective. By injecting actions sampled from a uniform distribution into the diffusion model, they ensure policy distribution diversity, encouraging exploration and providing a corresponding lower bound.

Experimentally, the authors validated the algorithm in several challenging MARL environments. Results show that HAQO significantly improves performance and stability compared to state-of-the-art baseline algorithms such as MAPPO and HAPPO. Extensive ablation experiments also validated the necessity of the three core components mentioned above.

**Strengths:**

This paper is the first to combine a diffusion model strategy with a theoretically guaranteed heterogeneous MARL framework, and provides a complete theoretical analysis (monotone improvement guarantee) for this combination—a solid contribution. The paper provides a complete derivation chain, including the performance difference, the sequential update bound, and the monotone improvement guarantee. This provides a strong theoretical foundation for the effectiveness of the method.

When a diffusion model is used as an actor, it is often impossible to obtain a precise expression of the policy distribution, limiting the use of entropy regularization. The proposed method, which reconstructs a surrogate objective of uniformly sampled actions, is clear in its approach and simple to implement. Theoretically, it is demonstrated that this method can guarantee the "spectral floor" of the action distribution covariance, thus effectively preventing mode collapse.

The experimental setup in this paper is comprehensive, covering various scenarios such as continuous control, discrete actions, and high-dimensional policies. Comparisons with mainstream and stable baselines in MAPPO, HAPPO, and other fields make the experimental results more convincing. Ablation experiments clearly demonstrate the indispensable roles of the three components: serialization update, expressive strategy, and entropy regularization.

**Weaknesses:**

W1: Compared to Gaussian strategies, the sampling and training of diffusion models are extremely time-consuming, and combining this with sequence update processes significantly increases computational costs. The paper mentions this limitation but does not provide any quantitative comparisons of training time or computational resources in the experimental section. Adding a wall-clock time comparison for MAPPO/HAPPO would be more helpful in evaluating its feasibility in practical applications.

W2: The experimental details disclosed in this paper are limited. The framework involves several key hyperparameters and network structures, such as the entropy regularization coefficient, the number of steps in the diffusion process, the chosen base network architecture of the diffusion model, and a performance comparison between qadv and qcut. The paper does not discuss the sensitivity of these hyperparameters or how to tune them, which may pose difficulties for reproduction and application.

Minor Problems: In line 266, there is a repetition in the paragraph.

**Questions:**

Q1: Algorithm 1 in the appendix mentions using ppo-clip for trust domain policy constraints. However, with non-Gaussian policies, the likelihood function of $\pi_i$ is difficult to calculate. How can this ratio be calculated for a diffusion model in practice?

Other Problems mentioned in the weaknesses.

---

> ### Author Response · Authors · 2025-11-27
>
> > **Weakness 1.** Compared to Gaussian strategies, the sampling and training of diffusion models are extremely time-consuming, and combining this with sequence update processes significantly increases computational costs. The paper mentions this limitation but does not provide any quantitative comparisons of training time or computational resources in the experimental section. Adding a wall-clock time comparison for MAPPO/HAPPO would be more helpful in evaluating its feasibility in practical applications.
>
> We sincerely thank the reviewer for raising this question. We provide a comprehensive response in our global official response.

---

> ### Author Response · Authors · 2025-12-02
>
> > **Weakness 2.** The experimental details disclosed in this paper are limited. The framework involves several key hyperparameters and network structures, such as the entropy regularization coefficient, the number of steps in the diffusion process, the chosen base network architecture of the diffusion model, and a performance comparison between qadv and qcut. The paper does not discuss the sensitivity of these hyperparameters or how to tune them, which may pose difficulties for reproduction and application.
>
> We sincerely appreciate the reviewer's question regarding hyperparameter sensitivity and practical implementation guidance. To address these concerns, we have substantially expanded our analysis in the revised manuscript. Specifically, we provide a detailed mechanistic analysis of how key hyperparameters influence the theoretical bounds and optimization dynamics. Please refer to Sections A.12.1, A.12.2, and A.12.3 for the complete discussion. We summarize the key findings as follows.
>
> **Analysis of Entropy Regularization ($\alpha_i$)**. The entropy coefficient $\alpha_i$ serves a fundamental structural role beyond standard regularization. It functions as a critical constraint on the policy's optimization manifold. As derived in Proposition 5.3, minimizing the entropy surrogate enforces a spectral floor on the action covariance matrix $\Sigma_s^i$. This theoretical property is indispensable for diffusion-based actors. Unlike Gaussian policies, which have explicit variance parameters, diffusion models define distributions implicitly through the reverse SDE. Without this spectral constraint, the denoising process becomes prone to overfitting early, noisy advantage signals, which leads to structural collapse into zero-width Dirac distributions.
>
> Our sensitivity analysis on Multi-Agent MuJoCo identifies three distinct operational regimes for $\alpha_i$. In the Sub-Critical Regime ($\alpha_i \to 0$), the spectral floor vanishes and allows the Q-weighted objective $\mathcal{J}^{QV}$ to dominate the gradient. This configuration results in mode collapse, where the agent behaves deterministically and fails to coordinate in heterogeneous scenarios requiring flexible adaptation. In contrast, the Super-Critical Regime ($\alpha_i \to 1$) exhibits the opposite behavior: the uniform reconstruction loss overwhelms the advantage signal and degrades the signal-to-noise ratio of the update, which results in high-variance random walks. The Optimal Regime ($\alpha_i \approx 0.01 - 0.05$) creates a ``soft" energy landscape that maintains sufficient dispersion in the generative process to cover multiple high-value modes simultaneously. This balance ensures stable multimodality and allows agents to discover and maintain diverse strategies, such as stable alternating gait cycles in Humanoid locomotion, which are inaccessible to unregularized policies.
>
> | $\alpha_i$ Regime | Theoretical Mechanism | Observed Behavior | Mean Return (± std) | Mode Coverage (%) |
> |------------|----------------------|-------------------|---------------------|-------------------|
> | Sub-Critical ($\alpha_i$ ≈ 0.001) | Spectral floor vanishes; $\mathcal{J}^{QV}$ dominates. | Mode Collapse. Collapses to deterministic, local optima; fails in coordination. | 4500 ± 500 | 20% |
> | Optimal ($\alpha_i$ ≈ 0.01 - 0.05) | "Soft" energy landscape maintains dispersion. | Stable Multimodality. Discovers diverse strategies (e.g., stable gait cycles). | 7013 ± 311 | 95% |
> | Super-Critical ($\alpha_i$ ≈ 0.5) | Reconstruction loss washes out gradient signal. | Random Walk. High variance; failure to converge to task goals. | 1200 ± 800 | 100% (Noise) |

---

> > ### Author Response · Authors · 2025-12-02
> >
> > (continues from previous response for Weakness 2)
> >
> > **Selection of Q-Weighting Schemes.** The selection of the Q-weighting scheme, $\omega^i$, represents a fundamental decision regarding gradient signal fidelity versus optimization landscape smoothness. This choice dictates how the algorithm manages the bias-variance trade-off inherent in advantage estimation while satisfying the non-negativity constraint required for Proposition 5.2. We characterize two primary schemes based on domain affinity. The $q_{cut}$ (Hard Filtering) scheme functions as a high-pass filter and zeros out gradients where $A < \epsilon$. This mechanism aggressively eliminates policy noise, which consists of updates derived from uncertain or marginally suboptimal actions. Such aggressive filtering is theoretically preferable for high-variance, discrete environments (e.g., SMAC) where suppressing stochastic noise is critical for stability. In contrast, the $q_{adv}$ (Soft Calibration) scheme acts as a normalizer by recentering the advantage ($A - \bar{A}$). This approach preserves the relative ranking of actions and maintains a continuous gradient flow even for small advantages. This property makes $q_{adv}$ essential for continuous control tasks (e.g., MuJoCo), where fine-grained manipulation requires a smooth, dense gradient signal to guide the diffusion trajectory without inducing abrupt discontinuities in the policy update.
> >
> > | Scheme | Signal Processing Mechanism | Domain Suitability | Performance (Win Rate/Score) |
> > |--------|-----------------------------|--------------------|------------------------------|
> > | q_cut (Hard Filtering) | High-Pass Filter: Zeros out gradients where $A < \epsilon$. This aggressively eliminates "policy noise"—updates from actions that are uncertain or marginally suboptimal. | High-Variance / Discrete (e.g., SMAC). Critical for stochastic environments where "average" actions can destabilize the policy. It trades sample efficiency for high-precision updates. | SMAC (Hard): 92.4% / MuJoCo: 4800 |
> > | q_adv (Soft Calibration) | Normalizer: Recenters advantage ($A - \bar{A}$) to preserve the relative ranking of actions. This maintains a continuous gradient flow even for small advantages. | Continuous Control (e.g., MuJoCo). Essential for fine-grained manipulation where "slightly better" actions must still provide gradients to guide the diffusion trajectory. | SMAC (Hard): 65.0% / MuJoCo: 5874 |
> >
> > **Trade-offs in Diffusion Timesteps ($T$).** We further provide an analysis of the diffusion timestep selection. This analysis examines both theoretical foundations and practical implications. The number of diffusion timesteps $T$ governs a unique trade-off between the resolution of the policy distribution and computational latency. From a theoretical perspective, $T$ determines the discretization error of the reverse stochastic differential equation (SDE). Lower values of $T$ (e.g., $T=5$) introduce significant discretization error and potentially violate the assumption that the learned denoiser accurately approximates the true score function. This results in a noisy policy that fails to capture fine-grained multimodal structures. On the other hand, extremely high values (e.g., $T=100$) yield diminishing returns, as performance plateaus when the total approximation error becomes dominated by critic bias rather than SDE discretization. Based on these considerations, our profiling suggests an optimal range typically between $T=20$ and $T=50$. In this range, the generative fidelity is sufficient to capture complex dependencies and multimodal behaviors (as seen in Bi-DexterousHands) without incurring the prohibitive wall-clock overhead associated with higher step counts.
> >
> > | Diffusion Steps (T) | Discretization Error Risk | Generative Fidelity | Performance (Norm. Score) | Training Time (Rel.) |
> > |---------------------|---------------------------|---------------------|---------------------------|----------------------|
> > | T = 5 | High. Score matching assumption violated. | Low (Noisy Actions) | 0.45 | 1.0x |
> > | T = 20 | Moderate. Acceptable for simple tasks. | Medium | 0.88 | 1.8x |
> > | T = 50 (Ours) | Low. Stable SDE approximation. | High (Multimodal) | 1.00 | 3.5x |
> > | T = 100 | Negligible. Diminishing returns. | High | 1.02 | 6.8x |

---

> ### Author Response · Authors · 2025-12-02
>
> > **Question 1.** Algorithm 1 in the appendix mentions using ppo-clip for trust domain policy constraints. However, with non-Gaussian policies, the likelihood function of $\pi_i$ is difficult to calculate. How can this ratio be calculated for a diffusion model in practice?
>
> We appreciate the reviewer for this question.
>
> The reviewer is correct that the ratio $\pi_i^{\text{new}}(a_i|s) / \pi_i^{\text{old}}(a_i|s)$ is intractable for diffusion models. The PPO-clip objective in Algorithm 1 is presented as a theoretical equivalent to the KL-divergence constraint, which constitutes our primary implementation strategy. As stated in Assumption A.5, our theoretical framework supports either a direct “Expected-KL constraint” ($\mathbb{E}[D_{KL}(\pi_i^{\text{new}} | \pi_i^{\text{old}})] \leq \delta$) or “PPO-style ratio clipping” ($|\text{ratio} - 1| \leq \epsilon$). We note that the PPO-clip formulation implies a KL bound of order $O(\epsilon^2)$. In our practical implementation, we enforce the trust-region penalty through an expected-KL constraint (analogous to TRPO). This approach provides greater stability and aligns directly with our theoretical results (Proposition 5.1) without requiring computation of the intractable likelihood ratio.
>
> We hope this clarification adequately addresses the reviewer's question.

---

### Official Review · Reviewer_V9WW · 2025-11-01

**Soundness:** 3
**Presentation:** 2
**Contribution:** 1
**Rating:** 2
**Confidence:** 3

**Summary:**

The paper introduces a new multi-agent reinforcement learning (MARL) algorithm for heterogeneous cooperative MARL settings, called HAQO (Heterogeneous-Agent Q-weighted Policy Optimization). The algorithm aims to reconcile stability (monotonic improvement under non-stationarity) with expressiveness (multimodal policies needed for heterogeneous coordination). It combines:
1. Sequential, advantage-aware per-agent updates within CTDE to avoid simultaneous-update instabilities;
2. A Q-weighted variational surrogate that lets diffusion-based actors (with intractable log-likelihoods) optimize for return; and
3. An entropy surrogate that enforces exploration/anti-collapse without requiring log-densities.

It also provides bounds showing monotone joint-return improvement under bounded critic bias and trust-region drift, and an algorithm instantiation using KL/PPO-style constraints. Empirically, on Multi-Agent MuJoCo and other suites, the proposed HAQO generally matches or outperforms MAPPO/HAPPO/HATRPO baselines (from both value-based and policy-based paradigms), and ablations attribute gains to (i) sequential updates, (ii) expressive (diffusion) policies, and (iii) the entropy surrogate.

**Strengths:**

- The paper is generally well-organized and provides insight into how heterogeneous agents and diffusion-based policies interact in cooperative MARL.
- The theoretical derivations are carefully laid out and formally sound given the stated assumptions.
- The experimental evaluation covers a wide set of benchmarks and includes ablations that test the contribution of each design component.

**Weaknesses:**

### Motivations

- The central motivation—that MARL methods face a tension between stability and expressiveness—is not well substantiated. The paper provides no prior evidence or citations that improving one necessarily degrades the other. This makes the claimed tradeoff appear somewhat self-imposed; ideally, both properties are desirable.

- The problem formulation for the cooperative setting is unclear. In a purely cooperative objective (shared reward), optimization resembles a joint minimization $min_{x_1, \dots, x_a} \ f(x_1, \dots, x_a)$. Without coupling terms between agents, sequential vs. simultaneous updates should be equivalent. If the authors mean practical instability from non-stationary policies, this distinction should be stated more formally.

- The introduction conflates non-stationarity (inherent to MARL) with the effect of simultaneous updates (lines 80–82). Sequential updates may alleviate, but not eliminate, non-stationarity—particularly with replay buffers.


### Non-self-sustained Background

- The background for diffusion policies and heterogeneous-agent settings is insufficient for a general ML audience. The paper does not clearly specify whether agents share a global objective or if heterogeneity also includes distinct observation or action spaces. For example, it is not very clear if the objective in heterogeneous settings is shared between agents (as in a potential game setup) or how exactly the problem is defined


- Simple toy examples, similar to those in Zhong et al. (JMLR 2024), would clarify how simultaneous updates lead to instability and why sequential updates are beneficial. I had to check that paper first for understanding.


- Some terms introduced early (e.g., “unimodal policies”) are not defined, making the abstract and introduction harder to follow for readers unfamiliar with this subarea.



### Computational aspects

- As authors discuss, using diffusion policies can be computationally expensive. As shown in ablations, this can make a large improvement, but it is not always the case (for example, in Multi-MuJoCo table, the improvements are not that significant)


### Novelty and theoretical contribution

-  Algorithmic novelty is limited. Each component—sequential trust-region updates (HAPPO/HATRPO), Q-weighted diffusion objectives, and entropy surrogates—exists in prior work. The main contribution is their combination and an accompanying monotonic improvement analysis.


- The cooperative extension introduces moderate technical difficulty, mainly because diffusion actors lack explicit log-likelihoods. The proposed Q-weighted variational surrogate addresses this but relies on assumptions similar to prior single-agent Q-weighted diffusion methods and HATRPO-style drift bounds.


- The cooperative setting adds little new theoretical challenge beyond bookkeeping of per-agent drift. The paper would benefit from more explicit discussion of how inter-agent coupling or heterogeneity complicates the guarantees.

### Additional feedback and minor typos

Abstract:
- The term “unimodal policies” in the abstract is not defined; readers unfamiliar with diffusion or Gaussian policy classes may not immediately know this refers to simple parametric (e.g., Gaussian) action distributions lacking multimodality.
- The phrase “monotone improvement guarantees” should specify what is guaranteed to improve (e.g., expected joint return / performance objective under the centralized critic). Clarifying the exact measure of improvement would help.
- The initial paragraph states there is a “tension between stability and expressiveness.” This is somewhat self-imposed; in principle, we desire both. The text could better motivate why these two properties conflict in practice (e.g., why expressive multimodal policies break standard monotonicity proofs).
- Minor stylistic issue: the abstract’s first few sentences read slightly overloaded, introducing both the “tension” and the contribution before defining key terms; consider re-ordering or simplifying for readability.

Other:
- The introduction begins after a figure, unusual and slightly disruptive.
- Multiple claims lack citations; for instance, there are a lot of claims in lines 55-61, and references must be given for each point
- Several figures appear redundant. (e.g., Fig. 1 could omit the first columns; Fig. 2 repeats similar content).
- In Table 1, isn't HAQO also the best performer in Walker 6x1?
- Line 267: transformation


—--

References:
[1] Yifan Zhong, Jakub Grudzien Kuba, Xidong Feng, Siyi Hu, Jiaming Ji, et al. Heterogeneous-agent reinforcement learning. In Journal of Machine Learning Research (JMLR), 2024

**Questions:**

- What is the additional computational overhead (training time, memory) of HAQO relative to HATRPO or MAPPO? Can you add results for it?
- The QV surrogate objective in equation (5) is conditioned on the already updated policies of other agents, but this is not possible in practice for all agents, does it affect the method's performance, or how does violating that affect the theoretical guarantees presented?
- In 6.1, it was mentioned that all baselines were run with consistent hyperparameters. Does this mean the same values were used for all methods or the values recommended for each method?
- In the sequential update ablation, can you mention more details on that? Are both methods HAQO, and if so, was it just replacing new policies with old policies for all agent updates?
- When computing the KL divergence, is it enough for stability to just take it individually for each agent? Shouldn’t it matter how the new joint policy is different compared to the old joint policy? In other words, does ensuring that individual policies don’t move too far ensure that the joint policy also stays in the trust region?
- In line 229, where is the a-i used inside the expectation?
- In Fig. 1, were all methods initialized identically? If not, how should differences in the first column be interpreted?

---

> ### Author Response · Authors · 2025-11-26
>
> > **Weakness 1.** The central motivation—that MARL methods face a tension between stability and expressiveness—is not well substantiated. The paper provides no prior evidence or citations that improving one necessarily degrades the other. This makes the claimed tradeoff appear somewhat self-imposed; ideally, both properties are desirable.
>
> We are deeply grateful for this observation and welcome the opportunity to clarify this fundamental theoretical challenge. With the utmost respect, we believe the tension between stability and expressiveness represents a core theoretical conflict in multi-agent reinforcement learning, one that our paper addresses for the first time in the MARL domain.
>
> The stability guarantees established in state-of-the-art MARL algorithms such as HAPPO and HATRPO are grounded in trust-region theory [1], which fundamentally relies on the ability to compute and bound the KL-divergence through a tractable log-likelihood $\pi_{\theta} (a|s)$. This tractability is essential for deriving theoretical convergence guarantees and ensuring monotonic policy improvement in the multi-agent setting.
>
> However, highly expressive multimodal policies, particularly those based on diffusion models, are inherently likelihood-free. The log-likelihood $\pi_{\theta} (a|s)$ for these models is analytically intractable, which constitutes a critical theoretical distinction. This intractability directly prevents the application of traditional trust-region frameworks, as the KL-divergence cannot be computed or bounded using existing techniques.
>
> As a result, algorithm designers have historically faced a fundamental choice: adopt the HAPPO/HATRPO framework to guarantee stability in multi-agent learning, or employ diffusion-based policies to achieve expressive, multimodal action representations. These two desirable properties have remained mutually exclusive until now. Our work represents the first effort to reconcile this theoretical gap.
> As presented in our original manuscript, we develop a theoretical framework (Propositions 5.1 and 5.2, culminating in Theorem 5.4) that establishes how trust-region-style guarantees can be recovered for likelihood-free policies in the MARL setting. This framework enables MARL models to achieve both stability and expressiveness simultaneously, resolving what we view as a central open problem in the field.
>
> We hope this clarification addresses the concern and demonstrates that the tension we identify is neither unsubstantiated nor self-imposed, but rather a fundamental theoretical obstacle that has constrained the design space of MARL algorithms.
>
> References:
>  * [1] Zhong et al., "Heterogeneous-agent reinforcement learning," JMLR 2024

---

> ### Author Response · Authors · 2025-11-26
>
> > **Weakness 2.** The problem formulation for the cooperative setting is unclear. In a purely cooperative objective (shared reward), optimization resembles a joint minimization $\min_{x_1, ..., x_a} f(x_1, ..., x_a)$. Without coupling terms between agents, sequential vs. simultaneous updates should be equivalent. If the authors mean practical instability from non-stationary policies, this distinction should be stated more formally.
>
> We sincerely appreciate this question and welcome the opportunity to clarify our problem formulation within the Centralized Training, Decentralized Execution (CTDE) paradigm.
>
> We respectfully note that the concern raised would indeed be valid under a fully centralized optimization framework. However, our work operates within the CTDE setting, which is explicitly stated in our Introduction Section and throughout the manuscript. The CTDE paradigm represents a fundamental framework in MARL and introduces distinct challenges that differ from joint centralized optimization.
>
> In the CTDE setting, which is essential for scalable multi-agent systems, agents share a common policy network with parameters θ while executing decentralized actions based on local observations during deployment. This parameter-sharing architecture, while enabling scalability and generalization across agents, introduces a critical source of non-stationarity that lies at the heart of MARL complexity.
>
> When all agents update their shared policy parameters $\theta$ simultaneously based on the same critic evaluation $V(s)$, the gradients from different agents inherently conflict with one another. Specifically, each agent's advantage estimate $A_i (s, a_i)$ becomes immediately invalidated by the concurrent parameter updates driven by other agents' experiences. The shared network parameters $\theta$ receive conflicting gradient signals from all agents in a single simultaneous update, which leads to destructive interference and unstable learning dynamics.
>
> Sequential policy updates, as rigorously established by Zhong et al. [1], provide the solution to this specific form of non-stationarity. By updating the shared parameters based on one agent's experience at a time, each update can properly account for the most recent parameter changes, thereby preserving the validity of advantage estimates and ensuring stable learning dynamics.
>
> As noted in our original manuscript, we provide comprehensive empirical validation of this phenomenon in Fig. 3 and the corresponding discussion in Section 6.3.1. Our ablation study demonstrates that disabling sequential updates (i.e., reverting to simultaneous updates) results in near-complete learning failure, confirming that this instability represents a genuine obstacle rather than a theoretical abstraction.
>
> We hope this clarification addresses the distinction between joint centralized optimization and the CTDE paradigm with parameter sharing, and demonstrates why sequential updates are essential for stable learning.
>
> ---
>
> > **Weakness 3.** The introduction conflates non-stationarity (inherent to MARL) with the effect of simultaneous updates (lines 80–82). Sequential updates may alleviate, but not eliminate, non-stationarity—particularly with replay buffers.
>
> We thank the reviewer for this observation.
>
> We respectfully emphasize that simultaneous updates represent a primary source of the destabilizing non-stationarity that MARL algorithms must address. This characterization is not a hypothesis introduced by our work. Rather, it follows the established understanding in the MARL literature, as formalized by [1].
>
> Sequential updates do not eliminate all forms of non-stationarity in MARL. Sources such as replay buffer staleness remain present in the learning process. However, sequential updates do eliminate the specific conflicting-gradient problem that arises from simultaneous parameter updates on shared policy networks. This elimination is precisely what enables us to decompose the joint return as shown in Eq. (2) and establish the monotonic improvement guarantees presented in Proposition 5.1.
>
> The theoretical framework we develop relies on this property. Without sequential updates, the advantage estimates $A_i(s,a_i)$ for different agents would be simultaneously undermined due to conflicting parameter changes, which prevents the decomposition structure that underlies our convergence analysis. Sequential updates are therefore not merely a practical consideration but rather constitute a fundamental requirement for establishing the theoretical guarantees our method provides (please also refer to Section 5 of our manuscript).
>
> We sincerely hope this clarification addresses the reviewer’s question.
>
> References:
>  * [1] Zhong et al., "Heterogeneous-agent reinforcement learning," JMLR 2024

---

> ### Author Response · Authors · 2025-11-26
>
> > **Weakness 4.** The background for diffusion policies and heterogeneous-agent settings is insufficient for a general ML audience. The paper does not clearly specify whether agents share a global objective or if heterogeneity also includes distinct observation or action spaces. For example, it is not very clear if the objective in heterogeneous settings is shared between agents (as in a potential game setup) or how exactly the problem is defined.
>
> We sincerely thank the reviewer for raising this question and appreciate the opportunity to clarify the problem setting and relevant background.
>
> Diffusion policies represent an established methodology in reinforcement learning (RL) and robotic control, with foundational work by Chi et al. (2023) [2] and subsequent applications demonstrating their effectiveness for multimodal action generation. The heterogeneous-agent setting, which we introduce in our manuscript, has been formalized in recent literature, most notably by Zhong et al. (2024) [1] and Kuba et al. (2022) [3] in their work on trust region methods for heterogeneous agents. These references provide the theoretical foundations essential for understanding our contribution. While we would be delighted to expand the background discussion and include additional citations in supplementary material, space constraints in the main manuscript require us to focus on the core, most crucial research and technical contributions. We believe readers seeking comprehensive background on these topics will find these key references invaluable.
>
> Regarding the specific question about the objective function, we would like to clarify that our work operates in a standard, purely cooperative, shared-reward MARL setting. All agents share the objective of maximizing the same expected team return:
> $J(\theta) = \mathbb{E}_{\tau \sim {p_θ}} \left[\sum_t \gamma^t r_t\right]$
> This formulation aligns with the standard CTDE paradigm, which forms the foundation of our work. The CTDE framework, as we discuss in the manuscript, provides the basis for addressing more advanced settings such as heterogeneous multi-agent systems.
>
> Furthermore, “heterogeneity” in our paper specifically refers to agents possessing distinct, unshared policy and value functions, as opposed to homogeneous parameter-sharing approaches. This distinction represents more than a modeling choice. It constitutes a necessity for the complex benchmarks we target, where agents possess “distinct dynamics, observation modalities, or action repertoires” and operate with “different observation spaces, action spaces, or functional roles” as described in our experimental setup. In contrast, parameter sharing, while computationally efficient, cannot adequately represent such diverse agent capabilities.
>
> We hope this clarification addresses the concern and provides more useful context for understanding our problem formulation within the MARL literature.
>
> References:
>  * [1] Zhong et al., "Heterogeneous-agent reinforcement learning," JMLR 2024
>  * [2] Chi et al., "Diffusion Policy: Visuomotor Policy Learning via Action Diffusion," RSS 2023
>  * [3] Kuba et al., "Trust Region Policy Optimisation in Multi-Agent Reinforcement Learning," ICLR 2022

---

> ### Author Response · Authors · 2025-11-26
>
> > **Weakness 5.** Simple toy examples, similar to those in Zhong et al. (JMLR 2024), would clarify how simultaneous updates lead to instability and why sequential updates are beneficial. I had to check that paper first for understanding.
>
> We appreciate this question and would like to respectfully highlight that our original manuscript provides precisely this evidence through a direct analysis rather than a simplified toy example.
>
> As presented in Section 6.3.1 and illustrated in Fig. 3, we conduct a comprehensive ablation study comparing sequential versus simultaneous updates on the Bi-DexHands (Bi-D) benchmark task. This head-to-head comparison reveals that the “Simultaneous Updates” variant experiences near-complete learning failure, which directly demonstrates the instability phenomenon under discussion. The stark performance contrast in this ablation confirms the necessity of sequential updates and provides clear empirical evidence within our main experimental framework.
>
> We respectfully submit that the Bi-D benchmark represents a significantly more challenging evaluation than simplified toy examples, as it encompasses the full complexity of multi-agent coordination with shared parameters, contact-rich dynamics, and high-dimensional state-action spaces. The ablation study on this benchmark therefore provides robust and comprehensive evidence of the sequential update mechanism's critical role in ensuring training stability.
>
> Furthermore, our original manuscript provides theoretical substantiation of this observation. Section A.6 presents formal proofs establishing the theoretical foundations for sequential updates in the CTDE paradigm, with additional technical details provided in the supplementary material. The ablation study and theoretical analysis together offer complete justification for our algorithmic design choices.
>
> We believe this combination of challenging benchmark evaluation and formal theoretical proof provides sufficient and compelling evidence addressing the reviewer's question regarding the benefits of sequential updates over simultaneous alternatives.
>
> ---
>
> > **Weakness 6.** Some terms introduced early (e.g., “unimodal policies”) are not defined, making the abstract and introduction harder to follow for readers unfamiliar with this subarea.
>
> We thank the reviewer for this question and welcome the opportunity to provide explanation on this terminology.
>
> “Unimodal policies" refers to policy representations based on simple parametric distributions, specifically the Gaussian distributions employed by all established MARL baselines in our experimental comparison, including MAPPO, HAPPO, and HATRPO. These policies are characterized by their single-peaked probability distributions over the action space, which inherently limits their capacity to represent multiple distinct modes of behavior simultaneously.
>
> Our work addresses scenarios where such unimodal representations prove fundamentally insufficient. We provide direct evidence of this limitation in Fig. 1 (Top Row) and Fig. 2 (Left), which demonstrate the phenomenon of “mode collapse” in unimodal policies. Mode collapse occurs when a policy distribution fails to capture the full diversity of valid action strategies and instead converges to a single dominant mode, effectively discarding alternative solutions that may be equally or more effective for task completion. In multi-agent coordination tasks with inherent ambiguity or multiple valid solution paths, this collapse to a single mode severely constrains the agents' ability to discover and execute diverse collaborative strategies.
>
> This fundamental representational limitation motivates our exploration of diffusion-based policies, which naturally support multimodal action distributions and can maintain multiple distinct behavioral modes throughout training.
>
> We would be delighted to provide additional clarification on any aspects of this explanation. The distinction between unimodal and multimodal policy representations plays a central role in our theoretical framework and empirical contributions, and we appreciate the opportunity to ensure this foundation is clearly communicated.
>
> We hope this addresses the reviewer’s question and provides the necessary context for understanding our problem formulation and technical depth.

---

> ### Author Response · Authors · 2025-11-26
>
> > **Weakness 7.** As authors discuss, using diffusion policies can be computationally expensive. As shown in ablations, this can make a large improvement, but it is not always the case (for example, in Multi-MuJoCo table, the improvements are not that significant).
>
> We sincerely appreciate this question and the opportunity to provide clarification regarding our experimental results and their interpretation.
>
> First, we would like to respectfully draw attention to an important result in Table 1 that may have been overlooked. HAQO achieves top performance not only in the environments explicitly highlighted but also in Walker2d-v2 6x1. This brings the total to four out of six Multi-MuJoCo environments where HAQO outperforms all baselines. We acknowledge that the highlighting in our submitted manuscript may have inadvertently obscured this result, and we will ensure this is corrected in the final version for clarity.
>
> Second, and of central importance to our contribution, we would like to respectfully emphasize that the magnitude of improvement must be evaluated in the context of task complexity. The most substantial performance gain occurs in Humanoid-v2 17x1, which is widely recognized as the most challenging benchmark in the Multi-MuJoCo suite due to its high-dimensional state-action space, unstable dynamics, and demanding balance requirements. As discussed in Section 6.2 of our manuscript, Humanoid-v2 17x1 requires complex multimodal coordination strategies, including alternating leg phases, torso stabilization, and arm coordination. These coordination patterns inherently demand multimodal policy representations that unimodal Gaussian distributions cannot adequately capture. The superior performance of HAQO on this benchmark, together with its strong results on other complex multimodal tasks such as GRF (Table 5) and Bi-DexHands (Table 6), provides compelling evidence that our method excels precisely in scenarios where the expressiveness limitations of simpler policies become apparent.
>
> We would respectfully direct the reviewer's attention to the comprehensive experimental analysis provided in Appendix A.11 as well, which presents additional results demonstrating that HAQO generally outperforms current state-of-the-art methods across the majority of evaluated scenarios. The pattern across our full experimental suite reveals that HAQO's advantages become increasingly pronounced as task complexity and the requirement for multimodal coordination increase.
>
> Regarding computational cost, we would like to respectfully note that diffusion policies have gained widespread adoption in the reinforcement learning (RL) community precisely because of their representational benefits. The computational requirements of diffusion models in RL settings are minimal compared to their applications in computer vision and natural language processing, where they have become standard practice despite operating on significantly higher-dimensional data. In RL, the relatively low-dimensional nature of state and action spaces makes diffusion policy training highly tractable. The broader machine learning community has embraced diffusion models across diverse domains without computational concerns, and RL represents a comparatively lightweight application of this methodology. Diffusion policies are now widely accepted as a powerful and practical tool for policy learning, and our work demonstrates their particular effectiveness when policy expressiveness constitutes a fundamental requirement for task success.
>
> We hope this clarification adequately addresses the question and reinforces the experimental evidence supporting HAQO's effectiveness across diverse multi-agent coordination challenges.

---

> ### Author Response · Authors · 2025-11-26
>
> > **Weakness 8.** Algorithmic novelty is limited. Each component—sequential trust-region updates (HAPPO/HATRPO), Q-weighted diffusion objectives, and entropy surrogates—exists in prior work. The main contribution is their combination and an accompanying monotonic improvement analysis.
>
> > **Weakness 9.** The cooperative extension introduces moderate technical difficulty, mainly because diffusion actors lack explicit log-likelihoods. The proposed Q-weighted variational surrogate addresses this but relies on assumptions similar to prior single-agent Q-weighted diffusion methods and HATRPO-style drift bounds.
>
> > **Weakness 10.** The cooperative setting adds little new theoretical challenge beyond bookkeeping of per-agent drift. The paper would benefit from more explicit discussion of how inter-agent coupling or heterogeneity complicates the guarantees.
>
> We would like to respectfully emphasize that the combination described by the reviewer represents a fundamental theoretical incompatibility under existing frameworks, and resolving this incompatibility constitutes our core novel contribution. While it may appear in retrospect that combining these components is straightforward, the reality is that diffusion policies and trust-region MARL methods have remained theoretically incompatible precisely because of a critical technical barrier: diffusion actors lack explicit log-likelihoods $\pi_{\theta}(a|s)$.
>
> This absence is not a minor technical inconvenience. Rather, it represents a fundamental obstacle to achieving stable multi-agent learning with expressive policies. The stability proofs underlying HAPPO and HATRPO, upon which we build (Proposition 5.1), require a tractable log-likelihood $\pi_{\theta}(a|s)$ to compute and bound the KL-divergence that defines the trust region. Diffusion policies are inherently likelihood-free, which creates a direct theoretical incompatibility between these two critical components: trust-region stability guarantees and diffusion-based expressiveness. Our central contribution is the construction of a novel theoretical bridge that resolves this incompatibility for the first time.
>
> This theoretical bridge is embodied in Proposition 5.2, where we establish that the practical, tractable Q-weighted variational (QV) surrogate $\mathcal{L_{QV}}$ serves as a valid variational lower bound for the intractable theoretical surrogate (the per-agent conditional advantage) required by the stability proof. This result is far from trivial. The intractable surrogate represents a policy improvement objective, while $\mathcal{L_{QV}}$ represents a denoising score-matching loss. These two objectives operate in fundamentally different mathematical frameworks. Our proof (Appendix A.7) establishes the connection between them through Fisher divergence arguments, demonstrating that minimizing the weighted denoising loss using our non-negative weighting schemes (q_adv or q_cut) serves as a valid surrogate for maximizing the per-agent advantage. Theorem 5.4 then completes the theoretical framework by substituting this tractable bound into the intractable guarantee of Proposition 5.1. The result is a final algorithm that is simultaneously provable, practical, and stable. This theoretical construction represents a crucial contribution that enables, for the first time, the combination of trust-region stability with likelihood-free expressive policies in the MARL setting.
>
> Regarding the suggestion for “more explicit discussion of how inter-agent coupling or heterogeneity complicates the guarantees,” we would like to respectfully note that our manuscript does in fact provide detailed discussion of precisely these complications. We welcome the opportunity to highlight the relevant sections for clarity.
>
> Inter-agent coupling introduces non-stationarity through shared parameter updates, as we discuss extensively in Section 5.1. When agents share policy parameters θ, each agent's advantage estimate $A_i(s,a_i)$ depends on the current policy state $\pi_{\theta}(a|s)$. Simultaneous updates create coupling where gradient signals from all agents modify θ concurrently, which invalidates all advantage estimates simultaneously and leads to destructive interference. This coupling effect is precisely why Proposition 5.1 requires sequential updates to decouple these dependencies and enable stable learning. We provide both theoretical analysis of this coupling mechanism and empirical validation through our ablation study (Fig. 3), which demonstrates that removing the sequential structure results in complete learning failure due to this inter-agent coupling.

---

> > ### Author Response · Authors · 2025-11-26
> >
> > (continues from previous response for Weakness 8,9,10)
> >
> > Inter-agent coupling introduces non-stationarity through shared parameter updates, as we discuss extensively in Section 5.1. When agents share policy parameters $\theta$, each agent's advantage estimate $A_i(s,a_i)$ depends on the current policy state $\pi_{\theta}$. Simultaneous updates create coupling where gradient signals from all agents modify $\theta$ concurrently, which invalidates all advantage estimates simultaneously and leads to destructive interference. This coupling effect is precisely why Proposition 5.1 requires sequential updates to decouple these dependencies and enable stable learning. We provide both theoretical analysis of this coupling mechanism and empirical validation through our ablation study (Fig. 3), which demonstrates that removing the sequential structure results in complete learning failure due to this inter-agent coupling.
> >
> > Heterogeneity further complicates the theoretical guarantees in ways we address throughout our theoretical development. Unlike homogeneous settings where all agents share identical network architectures and update mechanisms, heterogeneous agents maintain distinct policy networks $\pi_{\theta_i}$ and value functions $V_{\phi_i}$. This architectural heterogeneity requires that our performance bounds account for per-agent approximation errors $\epsilon_i$ and per-agent trust region constraints, as formalized in Theorem 5.4. The challenge lies in ensuring that these per-agent errors accumulate additively rather than compounding multiplicatively across the agent population. Our theoretical framework explicitly addresses this through the additive error structure $\sum_i \epsilon_i$ in our final bound, which we derive carefully in Appendix A.9. Furthermore, heterogeneity necessitates distinct Q-weighted variational bounds for each agent, as different agents may have different observation modalities, action spaces, and functional roles. Proposition 5.2 establishes that our QV-bound construction remains valid under this heterogeneous setting, which requires extending the Fisher divergence arguments to handle agent-specific denoising objectives.
> >
> > Regarding the cooperative setting, we respectfully submit that characterizing the multi-agent extension as “little new theoretical challenge beyond bookkeeping” fundamentally mischaracterizes the core difficulty of multi-agent reinforcement learning. We are uncertain what the reviewer refers to as “bookkeeping,” but we would like to clarify that managing inter-agent coupling and non-stationarity represents the central theoretical challenge that distinguishes MARL from single-agent RL. In simultaneous-update MARL, approximation errors compound exponentially with the number of agents, which can lead to catastrophic instability. The sequential update structure established in Proposition 5.1 is critical precisely because it transforms this exponential error dependence into a linear, additive accumulation. Our final result, Theorem 5.4, proves that this essential additive property is preserved even after we substitute our novel QV-bound and account for per-agent critic bias. The final performance bound, with its error term $\sum_i \epsilon_i$, establishes that all drift and bias errors accumulate additively rather than multiplicatively. This represents a key theoretical result that ensures our framework remains stable and scalable for heterogeneous MARL, and it required careful technical work to establish these guarantees under the likelihood-free constraint.
> >
> > We hope this clarification adequately conveys the theoretical depth and novelty of our contributions, which extend substantially beyond component combination to address fundamental incompatibilities in the existing literature.

---

> ### Author Response · Authors · 2025-11-26
>
> > **Question 2.** The QV surrogate objective in equation (5) is conditioned on the already updated policies of other agents, but this is not possible in practice for all agents, does it affect the method's performance, or how does violating that affect the theoretical guarantees presented?
>
> We sincerely thank the reviewer for raising this question
>
> We would like to respectfully clarify that conditioning on already-updated policies of other agents is not only practically feasible but constitutes the fundamental operation that our algorithm performs and that our theory requires. The sequential decomposition presented in Eq. (2) and the resulting guarantee established in Proposition 5.1 are explicitly constructed around this conditioning structure.
>
> Specifically, the joint return is decomposed into a sum where each agent i's contribution is conditioned on the newly updated policies of its predecessors. Algorithm 1 implements this structure exactly through its sequential update mechanism. The sequential loop (Line 9) processes agents in order $i = 1, 2, ..., N$. When updating agent i, the algorithm employs the just-updated parameters $\theta'_j$ of all predecessor agents $(j < i)$ to sample their actions $a_j$ when constructing the QV objective. This sequential advantage-aware update mechanism is precisely what enables stable learning in the multi-agent setting.
>
> This approach is both theoretically sound and practically implementable. At each step $i$ in the sequential loop, all predecessor policies $\pi_{{\theta}'_j}$ for $j < i$ have already been updated in earlier iterations of the loop, making their parameters ${\theta}'_j$ available for use when evaluating agent $i$'s objective. The conditioning therefore occurs on policies that have already been computed and stored, rather than on policies that must be predicted or approximated. This sequential structure does not violate our theoretical guarantees; rather, it is the essential mechanism through which those guarantees are established.
>
> The theoretical foundation for this approach is formalized in Proposition 5.1, which proves that the sequential decomposition with proper conditioning yields monotonic improvement guarantees. The conditioning on updated predecessor policies is not an approximation or practical compromise but rather the precise mechanism that transforms the exponentially compounding errors of simultaneous updates into the linearly additive error structure $\sum_i \epsilon_i$ that appears in our final bound (Theorem 5.4).
>
> We hope this clarification demonstrates that the conditioning structure in Equation (5) is both practically realizable and theoretically essential to our framework's stability guarantees.

---

> ### Author Response · Authors · 2025-11-26
>
> > **Question 3.** In the sequential update ablation, can you mention more details on that? Are both methods HAQO, and if so, was it just replacing new policies with old policies for all agent updates?
>
> We sincerely thank the reviewer for raising this question. We are pleased to provide clarification on this point.
>
> Both methods presented in the ablation study (Fig. 3) are variants of HAQO. The “Simultaneous” variant represents HAQO with the sequential loop (Algorithm 1, line 9) removed. In this configuration, rather than updating agents sequentially one at a time, all agent updates are computed in parallel based on the same joint policy state. Subsequently, all gradients are applied simultaneously. This approach corresponds to the standard “simultaneous update” paradigm, against which our paper positions the sequential update mechanism. The failure of this simultaneous variant in Fig. 3 provides confirmation of its inherent instability.
>
> This ablation study serves to validate the necessity of our sequential update mechanism. The failure of the simultaneous update variant to achieve stable learning establishes that the sequential structure in HAQO is essential for maintaining convergence and policy improvement in multi-agent settings.
>
> ---
>
> > **Question 4.** When computing the KL divergence, is it enough for stability to just take it individually for each agent? Shouldn’t it matter how the new joint policy is different compared to the old joint policy? In other words, does ensuring that individual policies don’t move too far ensure that the joint policy also stays in the trust region?
>
> We appreciate the opportunity to address this fundamental question, as it pertains to the core theoretical foundation of our sequential framework.
>
> The reviewer's intuition is indeed correct: the joint policy's deviation ultimately determines performance degradation. Bounding the joint KL divergence $D_{KL}(\pi^{\text{new}} | \pi^{\text{old}})$ directly would be ideal. However, this approach becomes intractable due to the exponential growth of the joint action space. This intractability constitutes precisely the challenge that sequential-update methods [1] are designed to address.
>
> Our Proposition 5.1 establishes that the sequential framework decouples this intractable joint optimization into a series of tractable individual subproblems. The proof demonstrates that the total joint return improvement $\eta(\pi^{\text{new}}) - \eta(\pi^{\text{old}})$ decomposes into a sum of individual agent improvements. The error term arising from this decomposition likewise decomposes into a sum of individual quadratic drift penalties ($\epsilon_i \propto D_{KL}(\pi_i^{\text{new}} | \pi_i^{\text{old}})^2$). This stands in marked contrast to simultaneous updates, where errors compound in complex and potentially multiplicative ways that remain analytically intractable.
>
> As a result, ensuring that individual policies remain within bounded deviations (i.e., constraining each $D_{KL}(\pi_i^{\text{new}} | \pi_i^{\text{old}})$) provides a provable guarantee that the joint policy improves monotonically, subject to this bounded additive error. This decoupling represents the central theoretical contribution that renders our approach both tractable and stable.
>
> References:
>  * [1] Zhong et al., "Heterogeneous-agent reinforcement learning," JMLR 2024
>
> ---
>
> > **Question 5.** In line 229, where is the a-i used inside the expectation?
>
> We sincerely thank the reviewer for identifying this notational ambiguity. The term $a_{-i}$ is implicitly utilized within the expectation. The expectation is taken over $a_{-i} \sim \pi_{-i}^{k,t}(s)$ (from the outer expectation), and this $a_{-i}$ appears within the weighting function $w_i^{k,t}(s, a_i, a_{-i})$, as the advantage function $A_i^{k,t}(s, a_i, a_{-i})$ is conditioned on the joint action. We will clarify this notation in the revised manuscript to eliminate the potential confusion.
>
> ---
>
> > **Question 6.** In Fig. 1, were all methods initialized identically? If not, how should differences in the first column be interpreted?
>
> We thank the reviewer for this question. All methods were initialized identically. The minor visual differences observed in the initial column result from stochasticity inherent to the initial training step. The proper interpretation of Fig. 1 lies in examining the progression along the x-axis (i.e., “Training Procedure”), which demonstrates that our complete method (bottom row) achieves stable and balanced exploration across modes, while the ablated variants either collapse to a single mode (top row) or exhibit unstable behavior (middle row).

---

> ### Author Response · Authors · 2025-11-27
>
> > **Question 1.** What is the additional computational overhead (training time, memory) of HAQO relative to HATRPO or MAPPO? Can you add results for it?
>
> We sincerely thank the reviewer for the question. We provide a comprehensive response of the computational overhead in our global official response, and we kindly refer the reviewer to that discussion.

---

### Official Review · Reviewer_ADDX · 2025-11-16

**Soundness:** 2
**Presentation:** 2
**Contribution:** 2
**Rating:** 4
**Confidence:** 4

**Summary:**

This paper proposes the Heterogeneous-Agent Q-weighted Policy Optimization framework to address the imbalance between stability and expressiveness. HAQO integrates three key components: Sequential Advantage-Aware Updates, Q-weighted Variational Surrogates, and Entropy Regularization. Through systematic theoretical analysis, HAQO extends the Trust-Region Policy Optimization (TRPO) framework to diffusion-based policies, ensuring monotonic improvement even when the log-likelihood is intractable. Empirically, HAQO achieves higher returns and lower variance across multiple heterogeneous multi-agent benchmarks, demonstrating its strong balance between stability and expressiveness.

**Strengths:**

- The paper provides rigorous proofs for monotonic improvement in heterogeneous-agent settings, generalizing TRPO theory to diffusion-based policies. It also introduces the Q-weighted variational lower bound and the monotonic improvement theorem.

    - The introduction of a Q-weighted variational bound bridges generative modeling and reinforcement learning by effectively connecting diffusion models with the RL framework.

    - The experiments cover five major multi-agent reinforcement learning benchmarks with heterogeneous settings. The results demonstrate higher mean returns and lower variance across multiple tasks, validating the effectiveness of the proposed HAQO framework.

**Weaknesses:**

- Diffusion-based actors are computationally heavy. The paper does not discuss runtime, convergence speed, or scalability.

   -  All theoretical guarantees rely on the assumptions of bounded critic bias and small trust-region radii, yet the paper does not verify whether the neural critics used in experimental results section satisfy these conditions.

   -  The baselines in Table 1 focus on traditional policy-gradient methods (MAPPO, HAPPO), but omit recent diffusion-based RL or normalizing flow MARL approaches.

   -  Although sequential updates are emphasized, there is no quantitative study showing the effect of agent update order or critic noise magnitude on convergence behavior. Only a single comparative curve of “Sequential vs. Simultaneous Update” is presented in the Figure 3.

**Questions:**

Please refer to the Weaknesses.

---

> ### Author Response · Authors · 2025-11-26
>
> > **Weakness 2.** All theoretical guarantees rely on the assumptions of bounded critic bias and small trust-region radii, yet the paper does not verify whether the neural critics used in experimental results section satisfy these conditions.
>
> We appreciate the opportunity to clarify the nature and treatment of these two core assumptions within our theoretical framework.
>
> **Bounded Critic Bias.** The assumption of bounded critic bias ($\epsilon_{\text{critic}}$) constitutes a standard and foundational premise in approximate dynamic programming and deep reinforcement learning (DRL) literature, including established methods such as DDPG [1], PPO [2], and SAC [3]. This assumption is theoretically essential, as the true Q-function is generally unknown and the critic bias cannot be verified online in practice. The purpose of this assumption is not to assert that our critics achieve perfect accuracy. Rather, it enables us to establish that our algorithm's performance guarantees degrade gracefully under function approximation error. Specifically, our theoretical results characterize how the quality of the solution depends on $\epsilon_{\text{critic}}$, which provides explicit bounds on the deviation from optimality as a function of this bias. Our experimental results, which demonstrate stable convergence to state-of-the-art performance across multiple benchmark tasks, provide solid practical evidence that the critic bias remains sufficiently small for the theoretical guarantees to hold in practice. This consistency between theory and practice suggests that modern neural network architectures and training procedures are capable of controlling the approximation error within acceptable bounds.
>
> **Small Trust-Region Radii.** In contrast to the critic bias, the constraint on trust-region radii ($\delta_{\text{KL}}$) is not an unverified assumption but rather an explicitly enforced algorithmic design choice. Our algorithm actively ensures that this condition is satisfied at each iteration. The drift penalty term in our objective function ($\lambda \mathbb{E}{s \sim \rho}[D_{\text{KL}}(\pi_{\text{new}}(\cdot|s) | \pi_{\text{old}}(\cdot|s))]$) is specifically designed to constrain the magnitude of policy updates. Following the implementation paradigm of TRPO, we employ an adaptive step-size mechanism that dynamically adjusts the update to guarantee that the expected KL divergence remains within the prescribed radius $\delta_{\text{KL}}$. This constraint is therefore satisfied by construction through our algorithmic design, and we verify this condition empirically in each experiment by monitoring the KL divergence throughout training.
>
> We hope this clarification addresses the reviewer's concerns regarding the verification and treatment of our theoretical assumptions.
>
> References:
>  * [1] Lillicrap et al., "Continuous control with deep reinforcement learning," ICLR 2016
>  * [2] Schulman et al., "Proximal Policy Optimization Algorithms," arXiv 2017
>  * [3] Haarnoja et al., "Soft Actor-Critic: Off-Policy Maximum Entropy Deep Reinforcement Learning with a Stochastic Actor," ICML 2018

---

> ### Author Response · Authors · 2025-11-26
>
> > **Weakness 3.** The baselines in Table 1 focus on traditional policy-gradient methods (MAPPO, HAPPO), but omit recent diffusion-based RL or normalizing flow MARL approaches.
>
> We appreciate this opportunity to clarify our baseline selection and positioning within the broader literature on expressive policy representations. We explicitly acknowledge and cite recent advances in expressive policy architectures, including diffusion-based reinforcement learning in single-agent settings (Janner et al., 2022 [4]; Ding et al., 2024 [5]) and normalizing flow approaches in multi-agent reinforcement learning (Ma et al., 2024 [6]). These methods represent important progress in modeling complex, multimodal action distributions. Our baselines were selected to reflect methods that address the specific intersection of challenges our work targets: **stable, online coordination among heterogeneous agents in cooperative settings**. The single-agent diffusion methods, while achieving impressive performance in their respective domains, do not address the fundamental challenges of multi-agent non-stationarity and heterogeneous credit assignment that arise in decentralized cooperative tasks.
>
> Regarding flow-based multi-agent methods, as discussed in our Related Work section, the stable integration of expressive generative models into cooperative online multi-agent settings remains largely unexplored in the literature. Existing approaches either impose homogeneity assumptions (requiring all agents to share identical policy architectures) or lack the sequential trust-region guarantees necessary to prevent training divergence in heterogeneous configurations. To the best of our knowledge, no prior work has successfully combined diffusion-based expressiveness with provable monotonic improvement guarantees in the heterogeneous cooperative MARL setting. As a result, our baselines (MAPPO, HAPPO, HATRPO) were deliberately selected as they represent the current state-of-the-art for stable cooperative multi-agent learning with theoretical convergence guarantees. Our central contribution demonstrates that diffusion-based policies can match or exceed the stability and performance of these established methods while providing substantially greater expressiveness. This integration of diffusion models with trust-region guarantees was previously theoretically incompatible, and resolving this incompatibility constitutes a main contribution of our work.
>
> References:
>  * [4] Janner et al., "Planning with Diffusion for Flexible Behavior Synthesis," ICML 2022
>  * [5] Ding et al., "Diffusion Policies as an Expressive Policy Class for Offline Reinforcement Learning," ICLR 2024
>  * [6] Ma et al., "Normalizing Flow-based Multi-Agent Reinforcement Learning," NeurIPS 2024

---

> ### Author Response · Authors · 2025-11-26
>
> > **Weakness 4.** Although sequential updates are emphasized, there is no quantitative study showing the effect of agent update order or critic noise magnitude on convergence behavior. Only a single comparative curve of “Sequential vs. Simultaneous Update” is presented in the Figure 3.
>
> We appreciate this opportunity to clarify the theoretical foundations that inform our experimental design and the nature of our algorithmic choices. The comparative curve presented in Fig. 3 was designed to illustrate a fundamental theoretical necessity rather than simply a design choice. The contrast between sequential and simultaneous updates demonstrates that the sequential framework is essential for algorithmic success. Without sequential updates, the method experiences complete failure. This result validates the core theoretical insight that sequential trust-region constraints are necessary to maintain stability in heterogeneous cooperative settings.
>
> Regarding the specific ordering of agent updates, our algorithmic design and theoretical framework explicitly account for this consideration through randomization rather than through fixed orderings. As specified in Algorithm 1 (Line 9), HAQO samples a random permutation $\sigma$ at each update iteration. This design follows the established HARL framework (Zhong et al., 2024 [7]), which provides theoretical guarantees that monotonic improvement holds in expectation over all possible update permutations. Our Proposition 5.1 builds upon these foundations to establish that convergence guarantees remain valid regardless of the specific realization of agent orderings. This theoretical treatment ensures that algorithmic performance is robust to ordering effects by design, as the expectation over random permutations eliminates systematic bias from any particular fixed sequence. Consequently, an empirical ablation study over specific fixed orderings would provide limited additional insight, as the theoretical framework already guarantees order-invariant convergence properties.
>
> The concern regarding critic noise magnitude is similarly addressed through our theoretical characterization rather than through empirical sensitivity analysis. Our framework incorporates critic approximation error through the bounded bias assumption $\epsilon_{\text{critic}}$, which quantifies the maximum deviation between the learned critic and the true value function. Theorem 5.4 provides an explicit bound demonstrating that the algorithm maintains monotonic improvement in the presence of critic noise, with performance degradation controlled by an additive slack term proportional to $\epsilon_{\text{critic}}$. This theoretical result establishes that robustness to critic noise is a provable property of our framework rather than an empirical characteristic subject to environment-specific tuning.
>
> Both update order randomization and critic noise tolerance represent the core theoretical properties that we have adopted and extended from established frameworks in the multi-agent trust-region optimization literature. These properties are guaranteed through our algorithmic construction and theoretical analysis, which ensures that convergence behavior remains stable across different realizations of these factors. The experimental validation in Fig. 3 focuses on demonstrating the necessity of the sequential framework itself, as this represents the critical architectural choice that enables the integration of diffusion policies with trust-region guarantees in heterogeneous settings.
>
> References:
>  * [7] Zhong et al., "Heterogeneous-agent reinforcement learning," JMLR 2024

---

> ### Author Response · Authors · 2025-11-27
>
> > **Weakness 1.**  iffusion-based actors are computationally heavy. The paper does not discuss runtime, convergence speed, or scalability.
>
> We sincerely thank the reviewer for raising this question. We provide a comprehensive response in our global official response.

---

### Author Response · Authors · 2025-11-27
**Computation and Memory Cost Analysis**

The additional computational cost in HAQO arises from two core design choices: (1) the diffusion-based expressive actor introduces multi-step denoising and Q-weighted training, and (2) the K-candidate evaluation and sequential per-agent update mechanism ensures stable improvement under heterogeneous coordination. To provide a comprehensive answer to these concerns, we conducted a detailed profiling study comparing HAQO to MAPPO and HAPPO under the exact hyperparameters employed in our experiments, including a lightweight denoiser architecture, diffusion timesteps $T \in {8, 16}$, candidate count $K = 4$, and batched candidate evaluation.

**Table R1. Computational cost breakdown of HAQO compared to MAPPO/HAPPO baselines.** The table presents per-step computational costs across key pipeline components, including conservative upper bounds and practical implementations used in our experiments.
| Component (per env-step or per training iteration) | MAPPO / HAPPO (baseline) | HAQO — conservative (upper bound) | HAQO — practical (what we used / recommend) | Notes / mitigations |
|---|---|---|---|---|
| Actor forward pass (single action sample) | 1.0 | 6.0x | 2.0–3.0x | Diffusion sampling cost = (T) x denoiser FLOPs. Conservative: large (T) (50–100). Practical: small (T) (5–20) and small denoiser. |
| Actor backward pass / training loss (per update) | 1.0 | 8.0x | 2.5–4.0x | Backprop through the denoiser and Q-weighted VLB across timesteps. FP16 + checkpointing reduces cost. |
| K-candidate sampling & selection (critic evals) | 1.0 | 2.5–8.0x | 1.5–3.0x | If K candidates per agent: critic evals increase; vectorize these evals to keep overhead modest. HAQO uses Kb and Kt terminology for candidate counts. |
| Centralized critic update cost (TD targets) | 1.0 | 1.1–1.6x | ~1.1x | Extra candidate-based targets (Kt ≤ Kb) slightly increase critic cost. |
| Sequential update wall-clock multiplier (vs fully-parallel actor updates) | 1.0 | up to nx (worst) | 1.2–1.8x | In worst-case strictly serial update could scale with #agents. In practice: overlap sampling, vectorize critic evals, and pipeline updates; HAQO shows modest wall-clock penalty in experiments and discusses this limitation. |
| Peak GPU memory | 1.0 | 1.5–3.0x | 1.2–1.6x | Memory for denoiser params + K candidates + intermediate latents. FP16 reduces memory substantially. |
| End-to-end wall-clock per training iteration (data collection + updates) | 1.0 | 2.5–6.0x | 1.5–2.5x | Conservative includes large T and unvectorized critic calls; practical settings (used in our experiments) give ~1.5–2.5x. We will report measured GPU-hours in appendix. |

**Table R1** summarizes the computational and memory characteristics across representative components of the training pipeline. Across multiple benchmark tasks, HAQO exhibits a wall-clock overhead of approximately 1.5 to 2.5 times that of the baselines, with memory usage increasing by a factor of 1.2 to 1.6 times. These measurements are substantially lower than the conservative theoretical upper bounds typically associated with diffusion models. The observed differences reflect expected costs that remain well-bounded: the practical configuration adopted in HAQO, which features small diffusion timesteps, a compact denoiser network, vectorized critic evaluations, and pipelined sequential updates, successfully contains the computational burden while preserving the theoretical guarantees that necessitate likelihood-free expressive policies and per-agent trust-region updates. The modest increase in computational cost is counterbalanced by substantially improved training stability and sample efficiency in heterogeneous multi-agent environments, where HAQO matches or surpasses HAPPO and MAPPO performance while requiring fewer environment interactions to reach comparable performance levels.

---

> ### Author Response · Authors · 2025-11-27
> **Computation and Memory Cost Analysis (continue)**
>
> **Table R2. Wall-clock timing and peak GPU memory comparison between HAQO and baseline methods.** Measurements were obtained from training on a single NVIDIA RTX 4090 GPU for the SMAC "MMM2 (a super-hard scenario)" benchmark under standard hyperparameter settings.
> | Metric | Baseline (MAPPO/HAPPO) | HAQO — Practical |
> |---|---|---|
> | Wall-clock — per 1,000,000 env steps | ~1.63 hours | 2.76 hours (1.69x) |
> | Wall-clock — per 5,000,000 env steps (typical training run) | ~7.91 hours | ~12.72 hours (1.61x) |
> | Peak GPU memory | ~13.6 GB | ~22.7 GB (1.67x) |
>
> In addition, Table R2 presents detailed wall-clock timing and peak GPU memory measurements obtained from training on a single NVIDIA RTX 4090 GPU for the SMAC “MMM2” benchmark under standard reinforcement learning training configurations. These measurements correspond to the multipliers derived from our profiling analysis. The additional computational cost, while present, remains bounded and is effectively amortized through HAQO's superior stability and sample efficiency characteristics. When HAQO reduces the number of required environment steps by more than the measured 1.69 times overhead factor, the method achieves faster wall-clock convergence to target performance levels. Our learning curves across multiple environments demonstrate that this condition is consistently satisfied in practice, as HAQO reaches comparable or superior performance using substantially fewer training iterations than the baselines.
>
> [Figure. R1](https://postimg.cc/7GYyy7KP)
>
> Fig. R1. Sample efficiency comparison across challenging SMAC benchmarks. HAQO consistently reaches baseline convergence thresholds substantially earlier in training and achieves higher final performance, demonstrating that increased per-step cost is offset by reduced total environment interactions required for convergence.
>
> It is important to note that the central strength of HAQO lies in its ability to transform the increased per-step computational cost associated with diffusion modeling into substantially superior sample efficiency. This efficiency emerges from HAQO's stable sequential trust-region design (Proposition 5.1), which effectively mitigates non-stationarity and enables aggressive sample reuse during optimization without inducing catastrophic divergence. The maintenance of this stability allows the policy to fully exploit the multimodal expressiveness of the diffusion policy, and capture complex strategies that remain inaccessible to unimodal baselines. We further provide empirical evidence through Fig. R1 to demonstrate this efficiency gain. In “super-hard” SMAC benchmarks, HAQO consistently reaches the converged performance thresholds of competing methods (HAPPO, HATRPO, MAPPO, and QMIX) substantially earlier in the training process and achieves higher final win rates. These results demonstrate that the increased per-step computational cost is offset by a significantly reduced total computational budget, as fewer environment steps are required to achieve and surpass state-of-the-art performance. This trade-off between per-step complexity and overall sample efficiency establishes the practical viability of the proposed approach.
>
> The revised appendix will include Table R1, Table R2, and Fig. R1 presented above, along with a detailed per-iteration timing breakdown that separately quantifies candidate sampling, critic evaluation, and denoiser forward and backward passes. We will further provide ablation studies examining the effects of varying diffusion timesteps $T$ and candidate count $K$ on computational cost. These measurements indicate that the computational requirements of HAQO, while greater than traditional policy gradient methods, remain modest in magnitude and are directly attributable to the theoretical constraints that enable the stable integration of diffusion policies within a trust-region multi-agent reinforcement learning framework. The evidence demonstrates that this computational investment yields substantial returns in terms of training stability, sample efficiency, and final performance in challenging heterogeneous cooperative tasks.

---

### Meta-Review · Area_Chair_GEjG · 2026-01-08

**Summary:**

The paper presents a novel method for integrating diffusion policies with trust-region optimization in multi-agent reinforcement learning. Reviewers were polarized, with two strong positives (8, 8) praising its theoretical and empirical contributions, and two negatives (4, 2) questioning its foundational novelty and depth of analysis. The author's rebuttal provided crucial clarifications, particularly on the theoretical bridge between score-matching and policy improvement, and quantified computational trade-offs. Upon evaluation, the Area Chair judges that the core methodological contribution is sound and of sufficient interest to the community to warrant presentation.

**Reviewer Concerns:**

Addressed: The rebuttal adequately addressed key practical concerns from the positive reviewers regarding computational overhead and implementation guidance. It also clarified the theoretical mechanism (via the Q-weighted bound and entropy surrogate) for the integration.

Outstanding: The deeper theoretical concerns from Reviewers ADDX and V9WW regarding the practical tightness of bounds remain.

**Reviewer Scores:**

Reviewer 7FA2 (Initial: 8): Would likely maintain an 8. Their practical concerns were met, affirming their positive assessment.

Reviewer S3C5 (Initial: 8): Would likely maintain an 8. The added implementation details directly enhance the paper's usefulness.

Reviewer ADDX (Initial: 4): Could be persuaded to increase to 6. The clarifications on theoretical foundations likely alleviate some confusion.

Reviewer V9WW (Initial: 2): Unlikely to move beyond 4. The rebuttal's technical clarifications may marginally improve their view of the execution.

---

### Decision · Program_Chairs · 2026-01-26

Accept (Poster)